# OFD1 inhibition induces BRCAness to create a therapeutic vulnerability to PARP inhibition in pancreatic cancer

Peng Li[1,4], Junjie Ye [1,4], Qian Yang[1,4], Ni Wang[1], Chaoyi Li[1], Xiaoxiao Zou [1], Hanyan Luo[1], Yi Pan[1], Lingxi Jiang [2,3], Baiyong Shen [2,3] ✉, Zaiming Tang [1] ✉ & Qing Zhong [1] ✉

BRCAness is a homologous recombination repair (HRR) deficiency phenotype mimicking *BRCA1/2* loss, leading to PARP inhibitor sensitivity in BRCA-associated cancers including pancreatic cancer[1–7]. However, how to induce BRCAness in BRCA-proficient tumors remains unclear. We identify OFD1 as a positive regulator of BRCA1 in human pancreatic cancer cells and specimens, with its overexpression correlating with poor prognosis. OFD1 depletion impairs HRR and confers synthetic lethality with PARP inhibitors. Mechanistically, OFD1 interacts with E2F4 in the cytosol to prevent assembly of the transcriptional repressor DREAM complex at the *BRCA1* promoter. Targeting OFD1 or disrupting its interaction with E2F4 promotes E2F4 nuclear translocation and DREAM complex formation, suppressing *BRCA1* expression. OFD1 inhibition synergizes with olaparib in pancreatic cancer xenograft, spontaneous, and patient-derived xenograft models, and in other BRCA-associated cancer models. These findings reveal a mechanism of *BRCA1* transcriptional regulation and highlight OFD1 as a therapeutic target to induce BRCAness in BRCA-proficient pancreatic cancer.

Pancreatic ductal adenocarcinoma (PDAC) remains a lethal cancer globally, with a five-year survival rate of less than 20%[8]. Chemotherapy and surgery are currently the primary therapeutic options for PDAC, however, the low rate of early diagnosis and the high metastasis rate result in life-threatening recurrence in 80% of patients with PDAC who undergo surgical resection[9,10]. The most common genetic mutations associated with pancreatic cancer include *KRAS*, *TP53*, *SMAD4*, and *CDKN2A*[11]. Furthermore, whole-genome sequencing has identified inherited *BRCA1/2* mutations as genetic risk factors for pancreatic cancer[12]. *BRCA1* and *BRCA2* play a crucial role in the DNA repair process through the HRR pathway[13–16]. PARP inhibitor (PARPi) was initially developed to block DNA base excision repair, which leads to the conversion of single-strand nicks into double-strand breaks. Cells harboring *BRCA1/2* mutations are unable to repair these DNA double-strand breaks, ultimately resulting in programmed cell death[5,17]. More recently, PARPi have been shown to trap PARP1/2 proteins on DNA, thereby impairing DNA transcription and replication. In such cases, HRR is required to resolve the trapped PARP-DNA complexes. Cells with deficient HRR activity fail to repair these complexes, leading to replication fork degradation and apoptosis[2]. As the first drug developed based on the principle of synthetic lethality, PARPi has provided clinical benefits to cancer patients with *BRCA1/2* mutations[18]. Despite these significant breakthroughs, the clinical utility of PARPi remains limited to a small subset of PDAC patients, ~5–7%[6,19]. Therefore,

[1]Institute for Translational Medicine on Cell Fate and Disease, Shanghai Ninth People's Hospital, Key Laboratory of Cell Differentiation and Apoptosis of Chinese Ministry of Education, Department of Pathophysiology, Shanghai Jiao Tong University School of Medicine (SJTU-SM), Shanghai, China. [2]Shanghai Key Laboratory of Pancreatic Neoplasms Translational Medicine, Research Institute of Pancreatic Diseases, Shanghai Jiao Tong University School of Medicine, Shanghai, China. [3]State Key Laboratory of Systems Medicine for Cancer, Shanghai, China. [4]These authors contributed equally: Peng Li, Junjie Ye, Qian Yang. ✉e-mail: shenby@shsmu.edu.cn; zaimingtang2017@shsmu.edu.cn; qingzhong@shsmu.edu.cn

expanding the therapeutic applications of PARPi may have profound clinical significance.

The concept of BRCAness has recently been introduced to expand the applicability of PARPi. The canonical definition of BRCAness refers to defects in HRR that mimic *BRCA1/2* loss[20]. Efforts have been made to elucidate the mechanisms that underlie BRCAness. Inhibition of cyclin-dependent kinases (CDKs), which are critical regulators of the DNA damage response, enhances the efficacy of PARPi by impairing HRR in BRCA-proficient cancers. Cdk1 inhibition has been reported to sensitize BRCA-proficient cancers to PARP inhibition by impairing BRCA1 function[21]. Similarly, targeting CDK12/CDK13 induces a BRCAness phenotype in HRR-competent triple-negative breast cancer (TNBC) tumors, leading to a synergistic effect with PARPi[22]. In addition, ALK directly phosphorylates CDK9 at Tyr19, a modification that promotes HRR and confers resistance to PARPi[23]. However, given the critical roles of key kinases in cell cycle regulation, the potential toxicity and side effects of CDK inhibitors must be carefully considered. Identifying and managing these toxicities and side effects are crucial factors to be addressed before clinical application. BRCAness is observed in tumors with specific genetic alterations, characterized by abnormal gene expression that modulates the DNA Damage response or HRR process. For instance, CXorf67 is specifically upregulated in Posterior Fossa group A (PFA) ependymomas. CXorf67 inhibits HRR-mediated DNA repair, and PFA ependymomas with elevated CXorf67 expression can be effectively treated with PARPi[24]. Moreover, LMO2 expression is associated with improved survival in Diffuse Large B-Cell Lymphoma (DLBCL), possibly due to its inhibitory effect on HRR activity in DLBCLs, thus providing a rationale for the therapeutic use of PARPi in LMO2-positive DLBCLs[25]. Therefore, further investigation of key regulators of BRCAness with broad applicability is of clinical significance.

OFD1 (Oral-Facial-Digital Syndrome 1 protein) is a centrosomal protein associated with ciliopathies, a group of genetic disorders caused by abnormalities in ciliogenesis[26–28]. Our research has shown that autophagy promotes ciliogenesis by degrading OFD1 at centriolar satellites[29]. Notably, our recent studies have indicated a potential role of OFD1 in cancer. OFD1 is highly expressed in various cancer types, and knockdown of OFD1 suppresses the growth of multiple cancer cell lines in mouse xenograft models[30]. However, the therapeutic potential and underlying mechanisms of targeting OFD1 remain to be fully understood.

In this study, we identify a strong positive correlation between *OFD1* and *BRCA1* expression in pancreatic cancer. Depletion of OFD1 leads to downregulation of *BRCA1* mRNA transcription in a DREAM complex-dependent manner. Furthermore, depletion of OFD1 sensitizes pancreatic cancer cells and tumor models to PARPi. Thus, OFD1 depletion mimics *BRCA1* deficiency and may serve as a potential therapeutic strategy to sensitize HRR-proficient cancers to PARP inhibition.

## Results

### Overexpression of OFD1 in PDAC is correlated with poor prognosis

In our recent study, we identified a potential role for OFD1 in cancer, particularly in pancreatic cancer, where its knockdown significantly inhibited subcutaneous tumor growth[30]. To further investigate the role of OFD1 in pancreatic cancer progression, we analyzed *OFD1* expression in the publicly available GEO database. Analysis of multiple published PDAC datasets showed that *OFD1* was significantly overexpressed in pancreatic cancer tissues compared to normal pancreatic tissues[31–37] (Fig. 1a and Supplementary Fig. 1a). In addition, we confirmed the upregulation of OFD1 in a tissue microarray consisting of a cohort of 90 PDAC patients (Fig. 1b, c and Supplementary Data 1). Correlating tumor OFD1 expression with survival data showed that patients with higher OFD1 expression in their tumors had poorer median survival (Fig. 1d). To explore the functional role of OFD1 in

PDAC progression, we generated an *LSL-Kras*[G12D]; *p53*[172H/+]; *Pdx1-cre*; (KPC) mouse model in which Ofd1 was genetically depleted via Cre-mediated recombination in the pancreatic duct epithelial lineage, hereafter referred to as KPCO mice. Pancreatic *Ofd1* (*Pdx1-Cre*) null mice exhibited normal postnatal development (Supplementary Fig. 1b–d). Notably, *Ofd1* deficiency significantly prolonged survival of KPCO mice compared to that of KPC mice (Fig. 1e–g). These results suggest that OFD1 plays an important role in the progression of PDAC.

### OFD1 sustains the survival of pancreatic cancer cells to PARPi

To further evaluate the role of OFD1 in PDAC tumorigenesis, we investigated the inhibitory effects of OFD1 knockdown in a panel of pancreatic cancer cell lines. We generated lentivirus-delivered Tet-On inducible OFD1 knockdown cell lines as previously reported[30], and doxycycline (DOX) treatment efficiently reduced OFD1 protein levels (Fig. 2a). Interestingly, we observed a heterogeneous response to OFD1 inhibition in different pancreatic cancer cell lines ex vivo (Fig. 2a). Although some pancreatic cancer cell lines were sensitive to OFD1 inhibition, as we previously reported, others such as MIA PaCa-2, SW1990, HPAC, PATU8988T, BXPC3 and PANC1005 were considerably less sensitive (Fig. 2a). We hypothesized that although OFD1 depletion alone is insufficient to kill these pancreatic cancer cells, it may sensitize them to other anticancer therapies. To explore this, we conducted a chemical screening to identify compounds that could synergize with OFD1 inhibition to kill pancreatic cancer cells. A cellular viability screen using an FDA-approved drug library (Supplementary Data 2) was conducted on MIA PaCa-2, a pancreatic cancer cell line moderately sensitive to OFD1 knockdown (Fig. 2b). The top hits from the screening were identified as PARPi, including olaparib, veliparib and rucaparib (Fig. 2c). Furthermore, colony formation assays confirmed the synergistic effects of OFD1 knockdown and olaparib treatment in multiple pancreatic cancer cell lines, including MIA PaCa-2, SW1990, PATU8988T, and PANC1005 (Fig. 2d). Subsequently, we conducted in vivo functional validation. Xenograft tumors were established from SW1990 cells with OFD1 knockdown and re-expression of OFD1 RNAi-resistant form. These tumors were resistant to OFD1 knockdown. SW1990-shOFD1 tumors treated with olaparib showed a significant reduction in tumor growth rate, tumor mass, and tumor proliferation compared to tumors with either OFD1 knockdown or olaparib treatment alone. This synthetic lethality effect was abolished in SW1990 tumors with re-expressed OFD1 (Fig. 2e–g and Supplementary Fig. 2a, b). Immunohistochemistry (IHC) analysis revealed that the combination treatment reduced proliferation (as indicated by Ki67-positive cells) and increased γH2AX protein levels compared to tumors treated with either OFD1 knockdown or olaparib alone (Fig. 2h, i). These data suggest that targeting OFD1 sensitizes pancreatic cancer cells to PARPi, potentially acting as a synthetic lethal partner.

### OFD1 is a determining factor for BRCA1 expression

To investigate how OFD1 knockdown sensitizes pancreatic cancer cells to olaparib, we conducted RNA sequencing (RNA-seq) on the pancreatic cancer cell lines PANC1, MIA PaCa-2, PATU8988T following OFD1 knockdown. *BRCA1* emerged as one of the most significantly altered genes involved in homology-directed repair (HDR) in response to OFD1 knockdown (Fig. 3a–c and Supplementary Data 3). Gene Set Enrichment Analysis (GSEA) of the KEGG pathways and Gene Ontology biological processes revealed that OFD1 knockdown significantly downregulated genes involved in DNA HRR, DNA double-strand break repair, base excision repair, and DNA replication (Fig. 3d–g and Supplementary Fig. 3a). We further confirmed the effect of OFD1 knockdown on BRCA1 downregulation in several other pancreatic cancer cell lines (Fig. 3h–j and Supplementary Fig. 3b, c). Notably, the downregulation of BRCA1 caused by OFD1 knockdown could be rescued in a dose-dependent manner by reintroducing an exogenous optimized variant of OFD1 that is resistant to the siRNA (Fig. 3k). In addition, we

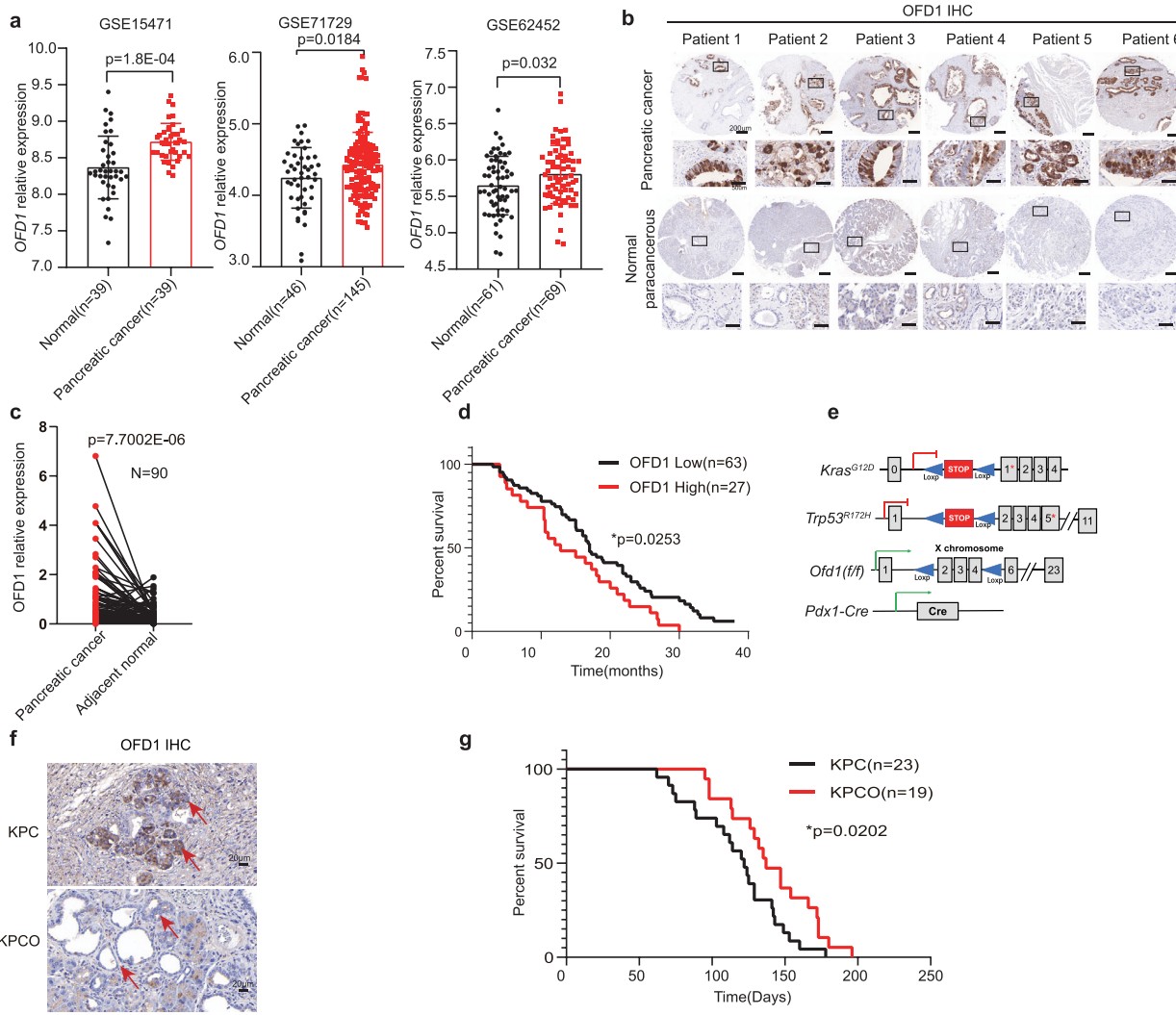

**Fig. 1 | Overexpression of OFD1 in pancreatic cancer correlates with poor prognosis. a** Upregulation of *OFD1* expression in human pancreatic cancer tissues. Gene expression levels of *OFD1* were compared between tumor (T) and normal (N) tissues using publicly available GEO datasets GSE15471[31], GSE71729[34] and GSE62452[35]. Statistical significance was assessed by a paired two-tailed *t*test for GSE15471, and unpaired two-tailed *t* test for GSE71729 and GSE62452. Error bars represent mean values ± SD. **b, c** Increased OFD1 protein levels in human pancreatic cancer. IHC analysis of OFD1 expression in human pancreas cancer tissue microarray samples (*N* = 90 patients) using an anti-OFD1 antibody. **b** Representative results from six patient samples are shown. Scale bars: 200 μm for full tissue sections, 50 μm for magnified images. **c** the statistical analysis of the difference in OFD1 expression between normal adjacent tissue and pancreatic cancer tissue was

performed using a two tailed paired *t* test. Error bars represent mean values ± SD. **d** Overall survival of patients with resected PDAC (PDAC microarray) according to OFD1 expression. Patient overall survival was examined using Kaplan-Meier survival analysis, followed by a Log-rank (Mantel-Cox) test (OFD1 low: *n* = 63 patients; OFD1 high: *n* = 27 patients). **e** Schematic of the *Ofd1*-deficient KPC mouse pancreatic tumor model. **f** Representative IHC staining for OFD1 (anti-OFD1 antibody) in pancreatic tumors from KPC (*LSL-Kras^G12D/+; Trp53^R172H/+; Pdx-1-Cre*) and KPCO (*Pdx1-cre; LSL-Kras^G12D/+; Trp53^R172H/+; Ofd1*-null allele) genotypes, Scale bar = 20 μm. **g** *Ofd1* knockout extended the lifespan of PDAC mice. Kaplan-Meier survival curves for mice of KPC (*n* = 23 mice) and KPCO (*n* = 19 mice) genotypes were analyzed by the Log-rank (Mantel-Cox) test. Source data are provided as a Source Data file.

used stable cell lines expressing siRNA-resistant GFP-OFD1 and Flag-OFD1 to demonstrate that reintroduction of OFD1 rescued BRCA1 expression and restored sensitivity to olaparib (Fig. 3l, m and Supplementary Fig. 3d, e).

Next, we investigated the correlation between *OFD1* and *BRCA1* expressions under various stresses. If OFD1 is a key determinant of BRCA1 expression, we would expect variations in OFD1 levels to correspond to changes in BRCA1 levels, and the expression patterns of *OFD1* and *BRCA1* should be tightly correlated under stress. Indeed, a significant positive correlation between *OFD1* and *BRCA1* expression was observed following treatment with a series of DNA damage-inducing agents and some small-molecule kinase inhibitors (Supplementary Fig. 3f and Supplementary Data 4). Furthermore, analysis of the CTRP database revealed that *OFD1* mRNA expression was

negatively correlated with drug sensitivity to several chemotherapeutic agents, including olaparib (Supplementary Fig. 3g, k). These data suggested that OFD1 plays a regulatory role in the expression of BRCA1 under various stress conditions.

## OFD1 deficient pancreatic cancer cells display compromised HRR

BRCA1 is well established as a crucial regulator in DNA HRR[38,39]. To determine whether OFD1 modulates the HRR or non-homologous end-joining (NHEJ) repair pathway, we employed the traffic light reporter (TLR) system, which provides a flow-cytometric readout for repair at I-SceI induced DNA double-strand breaks (DSBs)[40]. BRCA1 knockdown via siRNA was used as a positive control. Flow cytometry analysis revealed a significant reduction in HRR events in OFD1 knockdown

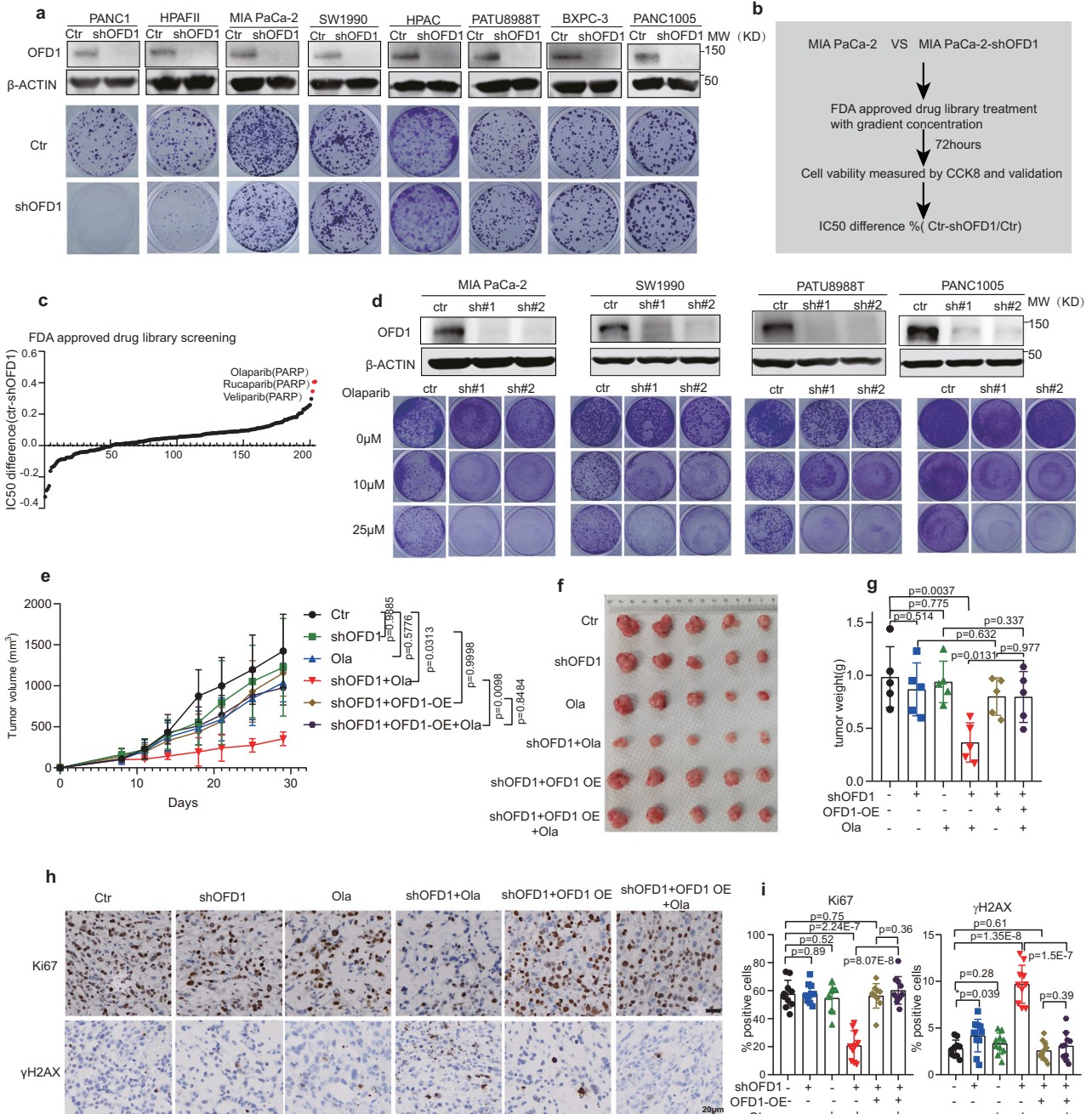

**Fig. 2 | OFD1 depletion sensitizes PDACs to PARPi treatment. a** Colony formation assay and immunoblot assays (upper panel) of the indicated pancreatic cell lines showing the heterogeneous killing effects following stable OFD1 knockdown. Representative images are from three independent biological replicates.
**b** Flowchart illustrating the synergistic drug screening strategy in OFD1 knockdown cells. **c** Comparison of drug sensitivity between MIA PaCa-2 and MIA PaCa-2-shOFD1 cells. Percentage changes in IC$_{50}$ values are shown. Common drug targets among the top hits are indicated. **d** OFD1 knockdown sensitizes pancreatic cancer cells to olaparib. Two independent shRNAs targeting OFD1 were used to knockdown OFD1 in MIA PaCa-2, SW1990, PATU8988T, and PANC1005 cells, with an empty vector (sh-Ctr) as the control. OFD1 knockdown efficiency was confirmed by western blotting, with β-Actin used as a loading control. Representative images are from three independent biological replicates. **e**–**g** SW1990 xenograft tumor mouse

model. **e** Tumor volume (mm³) in SW1990 xenografts treated with vehicle or olaparib. Tumor volume differences between groups were analyzed using two-way ANOVA followed by Tukey's post hoc test (*n* = 5 mice). **f** Representative images of tumors from each group at the end of the treatment period (*n* = 5 tumors). **g** Tumor weight was measured at the endpoint of the experiment (*n* = 5 tumors). Error bars, mean values ± SD, unpaired two-tailed student's *t* test. **h** IHC staining of Ki67 and γH2AX in tumor tissues. Representative images of Ki67 and γH2AX IHC staining in tumors from each treatment group. Scale bar = 50 μm. **i** Quantification of Ki67 and γH2AX IHC staining. Results are presented as the mean values from a representative experiment, including two randomly selected fields of view per mice (*n* = 5 tumors). Error bars, mean values ± SD, unpaired two-tailed student's *t* test. Source data are provided as a Source Data file.

cells, which was rescued by the overexpression of Flag-tagged OFD1 in MIA PaCa-2 (Fig. 4a, b and Supplementary Fig. 4a, b), This HRR defect was accompanied by a notable increase in chromosomal abnormalities, as observed in metaphase chromosome spread from OFD1

knockdown cells compared to that of wide-type MIA PaCa-2 cells (Fig. 4c). To further evaluate whether OFD1 knockdown affects DSB repair pathways, we examined the accumulation of γH2AX and BRCA1 after ionizing irradiation (IR). While both OFD1-deficient and wide-type

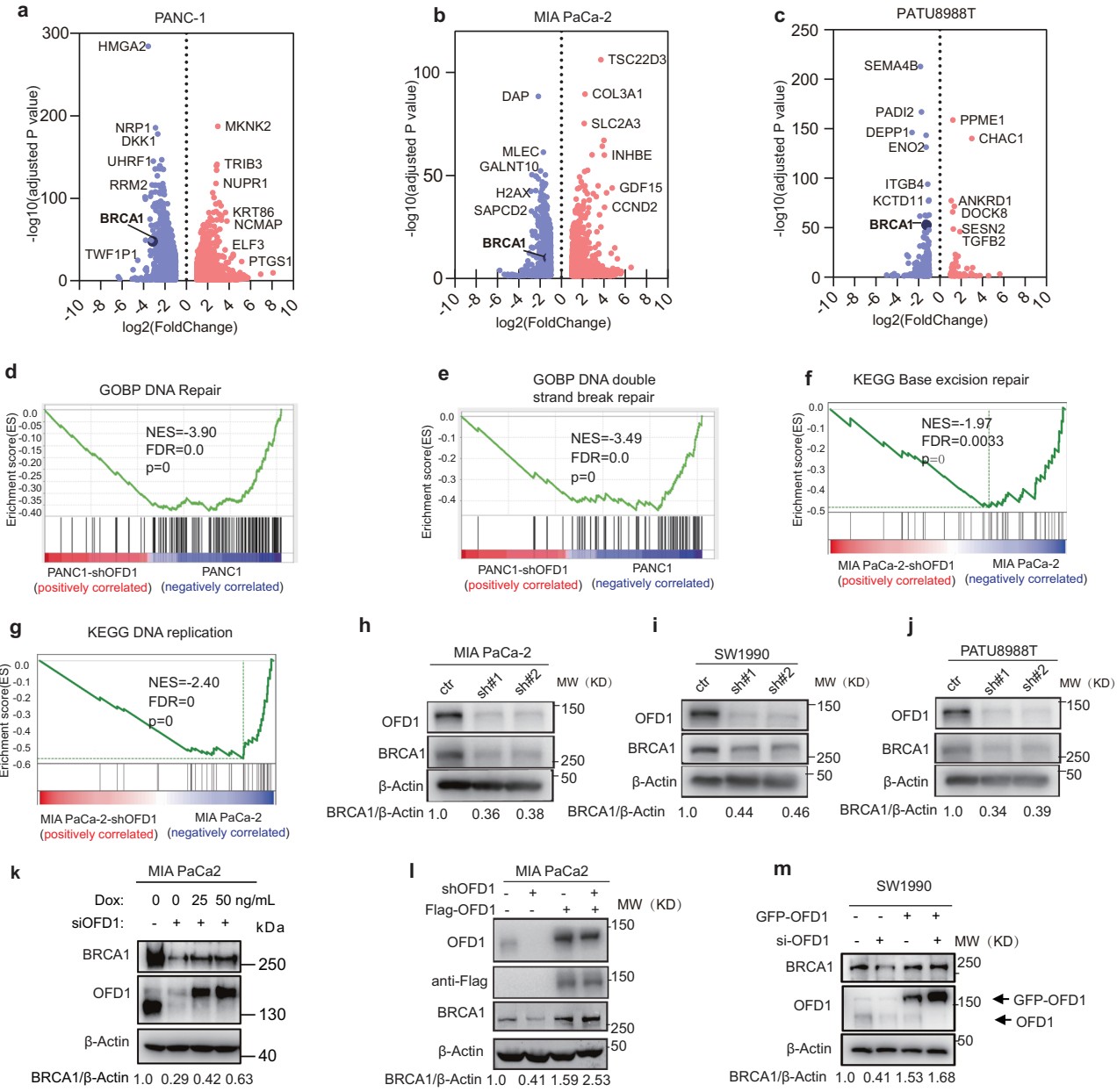

**Fig. 3 | Depletion of OFD1 downregulates BRCA1 expression. a–c** Volcano plots illustrating differential gene expression between control versus OFD1 knockdown pancreatic cell lines. Differential gene expression was assessed between control and OFD1 knockdown in PANC1 (**a**), MIA PaCa-2 (**b**), and PATU 8988T (**c**) cell lines. Two-sided Wald tests with Benjamini-Hochberg correction were used to assess differential gene expression. **d–g** Gene Set Enrichment Analysis (GSEA) of DNA damage response pathways upon OFD1 knockdown: (**d**) GOBP DNA repair pathway is downregulated upon OFD1 knockdown in the PANC1 cells (**e**), GOBP DNA double-strand break repair pathway is downregulated upon OFD1 knockdown in PANC1 cells. **f** KEGG base excision repair pathway is downregulated upon OFD1 knockdown in the MIA PaCa-2 cell line. **g** KEGG DNA replication pathway is downregulated upon OFD1 knockdown in the MIA PaCa-2 cell line. Normalized enrichment scores, nominal *P*-values (permutation test), and FDRs (Benjamini-Hochberg correction) are shown. Two-sided tests were applied. **h–j** Validation of BRCA1 downregulation by western blotting. Western blot analysis of BRCA1 protein levels in control versus shOFD1-transfected MIA PaCa-2 (**h**), SW1990 (**i**), and PATU8988T (**j**) pancreatic cancer cell lines. Representative images are from three independent biological replicates. Two independent shRNAs targeting OFD1 were used. **k** Rescue of BRCA1 expression in MIA PaCa-2 Cells by Tet-on inducible GFP-OFD1. Western blot showing that the Tet-on inducible system re-expressed GFP-OFD1 (resistant to siOFD1) restored BRCA1 protein levels reduced by siOFD1 in MIA PaCa-2 cells. MIA PaCa-2-Tet-on-GFP-OFD1 cells were transfected with siOFD1 and treated with varying concentrations of doxycycline (0, 25, 50 ng/mL), OFD1 and BRCA1 protein levels were assessed by western blot. Representative images are from three independent biological replicates. **l** Rescue of BRCA1 Expression in MIA PaCa-2 cells by stable re-expression of Flag-OFD1. as shown by Western blot using siOFD1-resistant construct. Representative images are from three independent biological replicates. **m** Rescue of BRCA1 expression in SW1990 cells by stable re-expression of GFP-OFD1. Western blot showing that stable re-expression of GFP-OFD1 (siOFD1-resistant) restored BRCA1 expression in SW1990 cells following siOFD1 knockdown. Representative images are from three independent biological replicates. Source data are provided as a Source Data file.

cells exhibited comparable levels of DNA damage upon irradiation, as indicated by similar γH2AX levels, OFD1 knockdown cells displayed a marked reduction in BRCA1 immunostaining and a defect in BRCA1 damage foci formation following IR exposure, and this defect was restored upon re-expression of OFD1 (Fig. 4e–g and Supplementary Fig. 4g–i). Furthermore, the combination of OFD1 knockdown and PARPi resulted in increased γH2AX foci formation compared to PARP inhibition alone (Supplementary Fig. 4c, d). To directly assess DNA

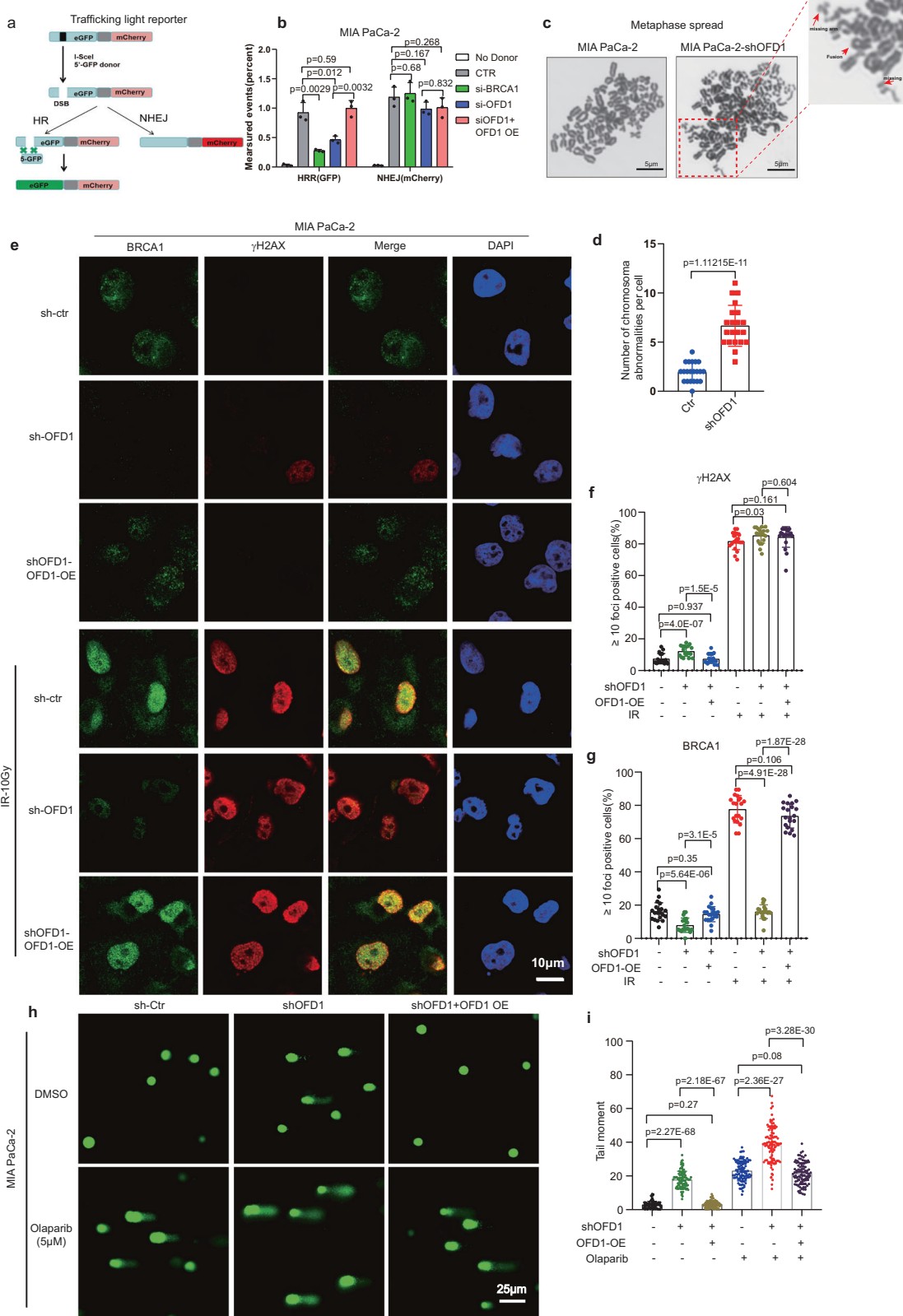

damage, we employed a neutral comet assay to determine whether OFD1 inhibition exacerbates PARPi-induced DNA damage. OFD1 knockdown modestly increased the tail moment, whereas the combination of OFD1 knockdown and PARPi treatment resulted in a synthetic increase in the tail moment, indicative of increased DNA damage. The tail moment was restored to levels that comparable to those of the vehicle treatment group upon re-expression of OFD (Fig. 4h, i and

Supplementary Fig. 4e–h). Taken together, these results suggest that OFD1 deficiency compromises HRR by downregulating BRCA1 expression.

## OFD1 regulates BRCA1 expression through the DREAM complex

To determine whether OFD1 affects *BRCA1* mRNA stability or transcription, we examined the stability of *BRCA1* mRNA. MIA PaCa-2 cells

**Fig. 4 | OFD1 depletion impairs BRCA1-mediated DNA damage response and repair efficiency. a** Schematic of the HR-GFP reporter for HRR efficiency. **b** Effect of OFD1 knockdown on HRR efficiency. GFP-positive cells were quantified by FACS 48 h after transfection. BRCA1 knockdown was used as a positive control. Data: mean values ± SD ($n = 3$ biologically independent experiments); unpaired two-tailed Student's $t$ test. **c** Impact of OFD1 knockdown on metaphase chromosome spread in the MIA PaCa-2 cells. Arrows indicate chromosomes with missing arms or chromosomal fusions. Scale bar = 50 µm. **d** Quantification of chromosomal abnormalities in the indicated groups ($n > 20$ metaphase spreads). Error bars represent mean values ± SD, unpaired two-tailed student's $t$ test. $p = 1.11215E-11$ Ctr

vs shOFD1. **e–g** Representative Confocal images (**e**) and quantification of γH2AX(**f**) and BRCA1 (**g**) foci in MIA PaCa-2 cells with OFD1 knockdown or re-expression, ± 10 Gy IR. Scale bar, 10 µm. ≥ 20 fields per group analyzed. Data are mean ± SD; unpaired two-tailed Student's $t$ test. **h, i** Neutral comet assay in sh-Ctr, shOFD1, shOFD1 + OFD1 OE MIA PaCa-2 cells after treatment with 0 or 5 µM olaparib for 48 h. **h** Representative images of the comet assay. Scale bar = 25 µm. The tail moment was quantified using Image J (Open-Comet plugin) software. **i** Quantification of tail moment. Data are presented as mean values ± SD ($n = 100$ cells per group); unpaired two-tailed $t$ test. The experiments were performed in triplicate. Source data are provided as a Source Data file.

were treated with the transcriptional inhibitor actinomycin D, and *BRCA1* mRNA levels were monitored over time. The half-life of *BRCA1* mRNA showed no significant change in OFD1 knockdown cells compared to that of controls (Supplementary Fig. 5a, b). However, luciferase reporter assays revealed that *BRCA1* promoter activity was significantly reduced following OFD1 knockdown in MIA PaCa-2 cells (Fig. 5a), suggesting that OFD1 primarily regulates *BRCA1* expression through transcriptional regulation.

Next, we investigated the mechanism by which OFD1 regulates *BRCA1* transcription, GSEA molecular signatures analysis in OFD1 knockdown pancreatic cancer cells revealed significant enrichment of genes regulated by the DREAM complex in both PANC1 and MIA PaCa-2 cells upon OFD1 knockdown (Fig. 5b, c). The DREAM transcriptional repressor complex plays a pivotal role in regulating the expression of HR repair proteins, including BRCA1[41–43]. We hypothesized that the DREAM complex mediates the downregulation of *BRCA1* induced by OFD1 depletion (Fig. 5d). Previous studies have shown that the assembly of the DREAM complex requires phosphorylation of LIN52 at Ser28 by DYRK1A[44]. Harmine, a DYRK1A inhibitor, has been reported to disrupt DREAM complex formation[45]. We found that treatment with harmine rescued BRCA1 expression in OFD1 knockdown PANC1 and MIA PaCa-2 cells in a dose-dependent manner (Fig. 5e, f). Disrupting DREAM complex by CRISPR-Cas9 mediated targeting E2F4, a major component of DREAM complex, rescued OFD1 knockdown induced BRCA1 downregulation (Fig. 5h). Moreover, dual siRNA knockdown of *LIN37* and *LIN52*, two subunits of the DREAM complex, disrupted the DREAM complex and rescued BRCA1 downregulation caused by OFD1 knockdown (Fig. 5h and Supplementary Fig. 5c–f). In addition, co-immunoprecipitation assays demonstrated that OFD1 knockdown enhanced the interaction between DREAM components, specifically E2F4 and RBL2, as well as E2F4 and RBBP4 (Fig. 5i, j). Treatment with the CDK4/6 inhibitor palbociclib was used as a positive control to inhibit DREAM complex disassembly[46]. To further investigate the assembly of the DREAM complex on the *BRCA1* promoter, chromatin immunoprecipitation (ChIP) experiments revealed that OFD1 knockdown increased the recruitment of E2F4, RBL2, RBBP4, and LIN54 to the *BRCA1* promoter in both PANC1 and MIA PaCa-2 pancreatic cancer cells (Fig. 5k, l). Together, these findings indicate that OFD1 inhibits *BRCA1* transcription by promoting assembly of the transcriptional repressive DREAM complex on the *BRCA1* promoter.

## OFD1 prevents E2F4 nuclear entry to regulate BRCA1 expression

E2F4, a subunit of the DREAM complex, has been implicated in the transcriptional regulation of *BRCA1*[47,48]. Since E2F4-bound gene sets were significantly enriched in the downregulated gene sets following OFD1 knockdown (Fig. 5b, c), we investigated whether E2F4 is involved in the regulation of *BRCA1* by OFD1. Genetic ablation of endogenous E2F4 rescued *BRCA1* downregulation induced by OFD1 knockdown (Fig. 5g).

E2F4 plays a pivotal role in cell cycle progression, and its translocation from the cytoplasm to the nucleus is a crucial step in the transcriptional regulation of target genes by the repressive DREAM complex[49–52]. We hypothesized that OFD1 regulates *BRCA1* expression

by influencing the nuclear translocation of E2F4. Immunofluorescence and nuclear-cytoplasmic fractionation assays confirmed a significant increase in nuclear entry of E2F4 following OFD1 knockdown (Fig. 6a–d).

Given that OFD1 is predominantly localized in the cytoplasm of pancreatic cancer cells, we hypothesized that OFD1 regulates E2F4 translocation through interaction in the cytoplasm. The interaction between OFD1 and E2F4 was analyzed by co-localization and proximity ligation assay (PLA). The Super-resolution microscope showed that fluorescence signals from endogenous E2F4 colocalized with FLAG-tagged OFD1 in the cytoplasm (Supplementary Fig. 6b), and the association between endogenous E2F4 and GFP-tagged OFD1 was further detected in the PLA Assay using individual antibodies. PLA signals were observed in wild-type E2F4-expressing cells but not in E2F4 knockdown cells, confirming the specificity of this interaction (red signals in Supplementary Fig. 6a and Supplementary Fig. 6c–h). The interaction between recombinant OFD1 and E2F4 was further validated by endogenous immunoprecipitation (IP) in three pancreatic cancer cell lines (Fig. 6e) and an in vitro pulldown assay (Supplementary Fig. 6i, j). Overexpression of siRNA-resistant OFD1 reduced E2F4 nuclear entry caused by OFD1 knockdown (Fig. 6f, g), suggesting OFD1 sequestered E2F4 in cytosol to prevent the formation of DREAM complex.

E2F4 contains an N-terminal DNA-binding region (amino acid residues 1–85), a dimerization domain for TFDP1/2 binding (amino acid residues 86–181) and a C-terminal transactivation domain (amino acids 337–413) (Supplementary Fig. 7a). Interestingly, immunoprecipitation assays showed that OFD1 binds to E2F4 with an undefined region (amino-acid residues 182–336) (Supplementary Fig. 7b), Furthermore, mapping experiments identified the region corresponding to amino acids 32–41 in OFD1 mediates its interaction with E2F4. Deletion of this region compromised the interaction between OFD1 and E2F4 in a co-immunoprecipitation experiment (Supplementary Fig. 7c–e). To identify the specific binding site of E2F4 to OFD1, we divided the 181–337 region into eight segments and synthesized the corresponding peptides. A peptide competition assay revealed that peptide-spanning residues (283–302) disrupted the interaction between E2F4 and OFD1 (Supplementary Fig. 8a–c).

To determine whether disrupting the OFD1-E2F4 interaction in the cytoplasm could mimic the effects of OFD1 knockdown, specifically promoting E2F4 nuclear translocation and suppressing BRCA1 expression, we treated cells with a TAT cell-penetrating peptide conjugated to peptide (283–302). Immunofluorescence analysis demonstrated that the TAT-conjugated peptide (283–302) significantly promoted the nuclear translocation of E2F4 compared to that of the control peptide (Supplementary Fig. 8d–f). Moreover, treating pancreatic cancer cell lines MIA PaCa-2 and SW1990 with 25 µM of TAT-conjugated peptide (283–302) reduced BRCA1 expression and enhanced their sensitivity to olaparib compared to that of cells treated with a scrambled control peptide (Supplementary Fig. 8g, h). These findings indicate that OFD1 interacts with E2F4 in the cytoplasm to inhibit its nuclear entry and DREAM complex formation, thereby regulating *BRCA1* expression.

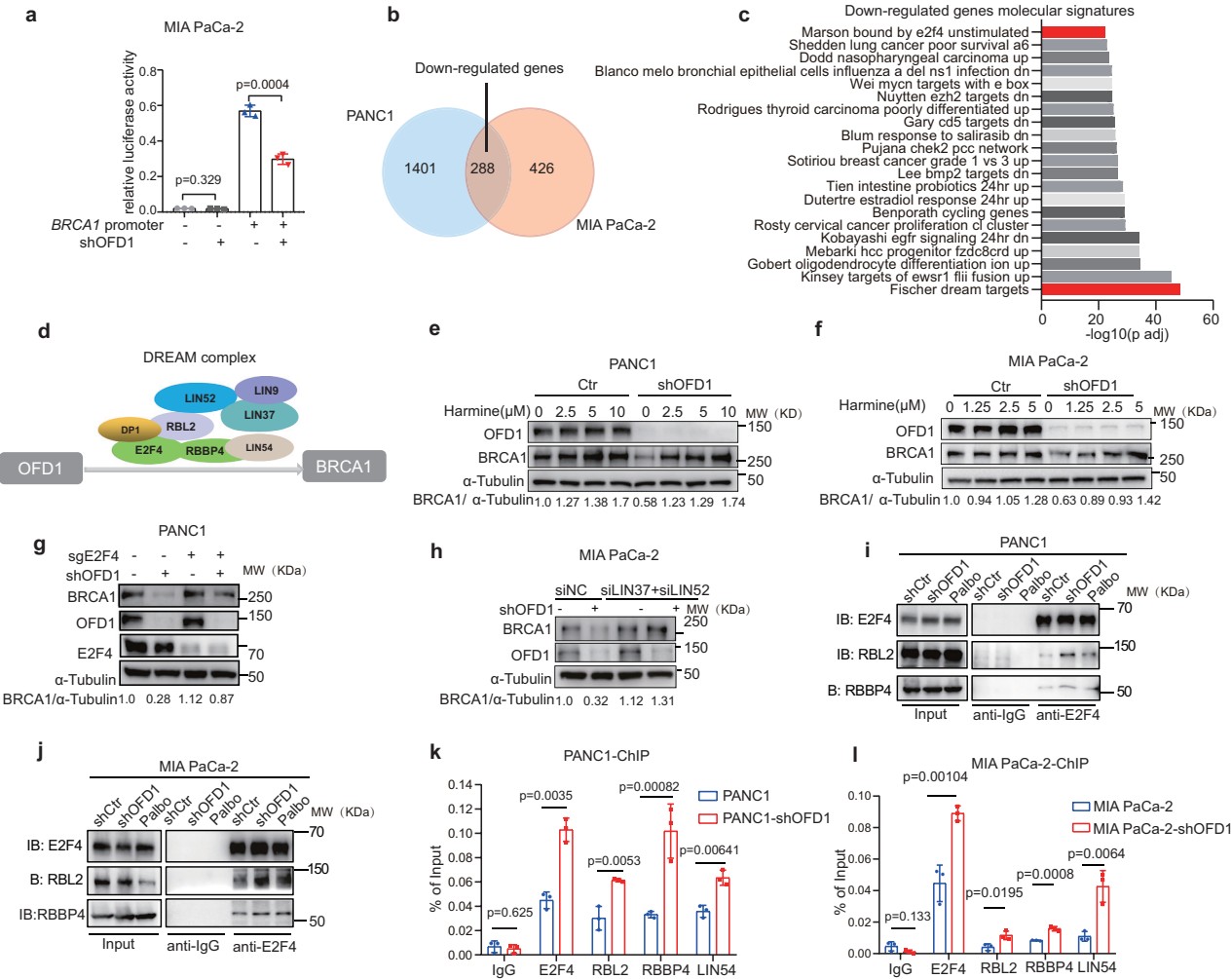

**Fig. 5 | OFD1 modulates BRCA1 expression via the DREAM complex. a** Analysis of BRCA1 promoter activity in MIA PaCa-2 cells after 48 h Dox-induced OFD1 knockdown. Cells transfected with BRCA1 promoter-luciferase or control vector; luciferase activity measured as firefly/Renilla ratio. Data: mean values ± SD (*n* = 3 biologically independent experiments); unpaired two-tailed Student's *t* test. **b** Venn diagram showing the overlap of downregulated genes in PANC1 and MIA PaCa-2 cell lines following OFD1 knockdown. **c** Molecular signature enrichment analysis (GSEA) of downregulated genes in the PANC1 and MIA PaCa-2 cell lines upon OFD1 knockdown using MSigDB. Nominal *P*-values were calculated by permutation test, NES accounted for gene set size and correlation, and FDR *q*- values were adjusted using the Benjamini-Hochberg method. All tests were two-sided. The downregulated gene pathways Fischer_DREAM_TARGETS and BOUND_BY_E2F4 were labeled in red. **d** Schematic model illustrating the hypothesis that *BRCA1* is transcriptionally repressed by the DREAM complex. **e, f** Harmine treatment rescues BRCA1 expression following OFD1 knockdown in pancreatic cancer cell lines. PANC1 (**e**) and MIA PaCa-2 (**f**) cells were treated with varying concentrations of

harmine upon OFD1 knockdown, BRCA1 protein levels were assessed by western blot, and the BRCA1/α-Tubulin ratio was quantified based on the grayscale intensity ratio. Representative images are from three independent biological replicates. **g** E2F4 knockout restores BRCA1 expression following OFD1 knockdown in MIA PaCa-2 cells. Representative images are from three independent biological replicates. **h** Dual knockdown of DREAM components *LIN52* and *LIN37* rescues BRCA1 expression suppressed by OFD1 knockdown. Representative images are from three independent biological replicates. **i, j** Endogenous co-immunoprecipitation (co-IP) indicates increased E2F4-RBL2, E2F4-RBBP4 interaction upon OFD1 knockdown or Palbociclib (0.5 μM for 48 h) in PANC1 (**i**) and MIA PaCa-2 (**j**) cell lines. Representative images are from three independent biological replicates. **k, l** ChIP-qPCR analysis showing increased enrichment of DREAM complex members (E2F4, RBL2, RBBP4, LIN54) at the BRCA1 promoter in shOFD1 PANC1 (**k**) and MIA PaCa-2 (**l**) cells. Data are presented as mean values ± SD (*n* = 3 independent experiments); unpaired two-tailed Student's *t* test. Western blot is representative of three independent biological replicates. Source data are provided as a Source Data file.

## Targeting OFD1 synergizes with olaparib in orthotopic and spontaneous pancreatic tumor models

Given the demonstrated synergistic effect of OFD1 knockdown and PARPi in vitro and xenograft animal models, we further validated this synergy using a luciferase bioluminescent pancreatic orthotopic model. Nude mice bearing luciferase-expressing orthotopic MIA PaCa-2 tumors were treated with vehicle, OFD1 knockdown, olaparib, or the combination of OFD1 knockdown and olaparib. Tumor bioluminescence was measured every two weeks. While the vehicle and olaparib treatment groups exhibited significant tumor burden, the OFD1 knockdown group showed a marked reduction in tumor burden. Notably, mice treated with the combination of OFD1 knockdown and olaparib had tumor

burden reduced to nearly undetectable levels (Fig. 7a–c). In addition, analysis of the relative pancreas mass at the endpoint (day 78) revealed significant tumor regression in the combination of OFD1 knockdown and olaparib group (Fig. 7d). The doxycycline-induced knockdown of OFD1 and downregulation of *BRCA1* in tumor tissues were validated by western blot analysis (Fig. 7f). Furthermore, examination of organ metastasis at the endpoint revealed extensive metastasis in the vehicle and olaparib treatment groups, whereas the combination treatment group exhibited almost undetectable organ metastasis (Fig. 7e, g and Supplementary Fig. 9a). These findings suggest that the combination of OFD1 knockdown and olaparib treatment may provide a promising approach to reduce organ metastasis.

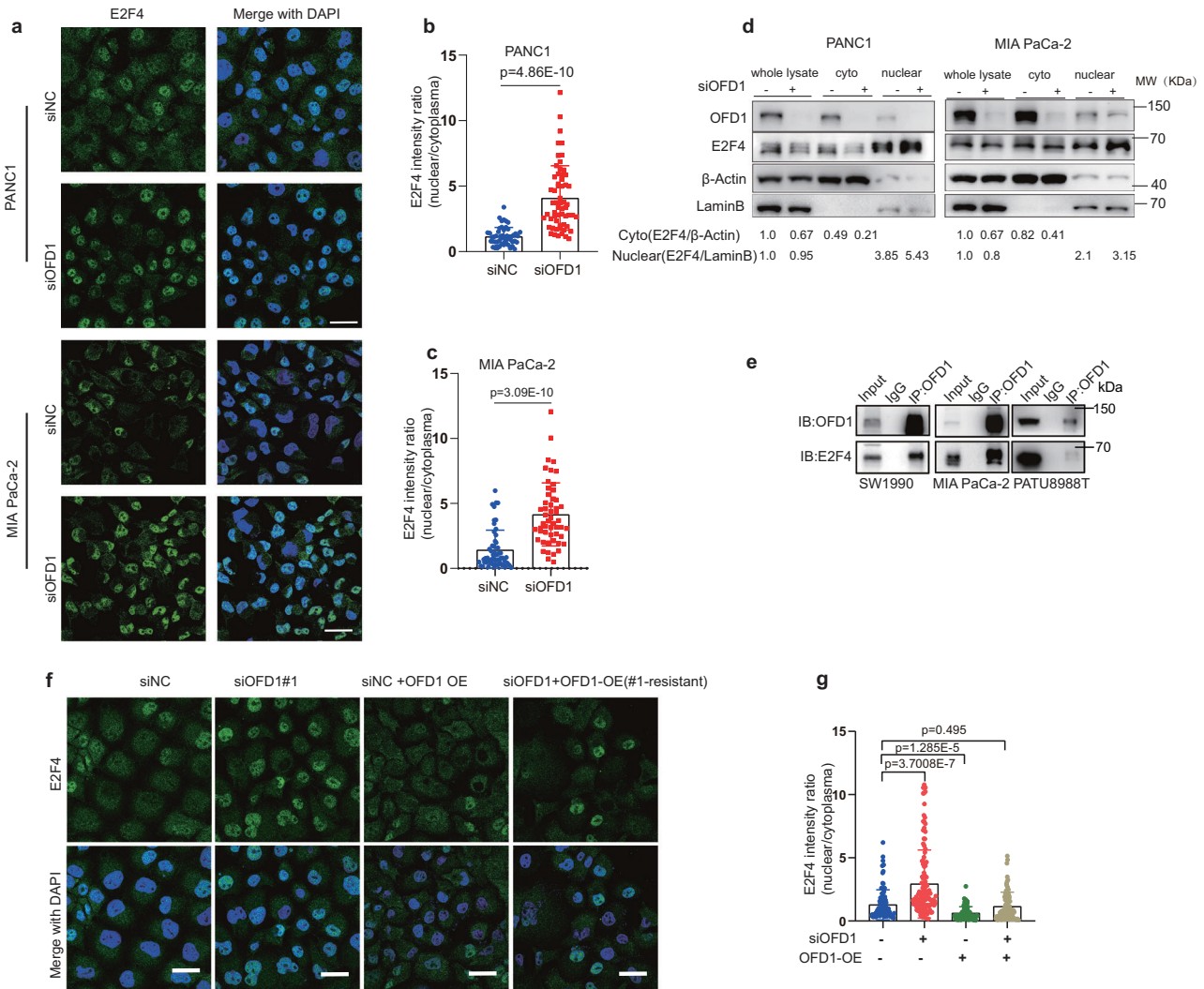

**Fig. 6 | OFD1 regulates the nuclear import of E2F4. a** OFD1 knockdown promotes nuclear translocation of E2F4. Representative immunofluorescence images of E2F4 localization in PANC1 cells following OFD1 knockdown were shown. Scale bar = 20 μm. **b, c** Quantification of Nucleus/Cytosol E2F4 intensity ratio in PANC1 (**b**) and MIA PaCa-2 (**c**). Data are presented as mean values ± SD, unpaired two-tailed Student's *t* test. PANC1: siNC (*n* = 51 cells) vs siOFD1(*n* = 60 cells); MIA PaCa2: siNC (*n* = 54 cells) vs siOFD1 (*n* = 52 cells). **d** Western blot analysis of subcellular E2F4 distribution. Nuclear and cytoplasmic fractions were isolated from PANC1 and MIA PaCa-2 cells with or without OFD1 knockdown. Lamin B1 and β-Actin were used as markers for nuclear and cytoplasmic fractions, respectively. Representative images are from three independent biological replicates. **e** Endogenous co-immunoprecipitation confirms the interaction between OFD1 and E2F4. IP was performed in SW1990, MIA PaCa-2, and PATU8988T pancreatic cancer cell lines. IgG is served as a negative control. Representative images are from three independent biological replicates. **f, g** Rescue experiment validates the role of OFD1 in regulating E2F4 nuclear translocation. Representative immunofluorescence images (**f**) and quantification of the nuclear-to-cytoplasmic E2F4 intensity ratio (**g**) in PANC1 cells treated with siNC, siOFD1, GFP-OFD1 (siRNA#1-resistant), and siOFD1 + GFP-OFD1. Scale bar = 10 μm. Data are shown as mean values ± SD; unpaired two-tailed Student's *t* test. siNC (*n* = 93 cells) vs siOFD1 (*n* = 118 cells); siNC (*n* = 93 cells) vs OFD1-OE (*n* = 80 cells); *p* = 0.495 siNC (*n* = 93 cells) vs siOFD1 + OFD1 OE (*n* = 81 cells). Western blot is representative of three independent biological replicates. Source data are provided as a Source Data file.

Next, we treated MIA PaCa-2 orthotopic tumor-bearing mice with vehicle, shOFD1, olaparib, and the combination of OFD1 knockdown and olaparib over an extended treatment period. The survival of these mice was monitored for one year. Notably, olaparib alone did not significantly extend overall survival, as most mice succumbed to pancreatic cancer within three months. In contrast, inhibition of OFD1 alone prolonged survival to approximately 6-7 months. Remarkably, the combination of olaparib and OFD1 knockdown significantly improved survival, with 66% of the treated mice surviving until the experimental endpoint on Day 365 (Fig. 7h).

A major limitation of the transplant pancreas cancer mouse model is the lack of a competent immune system, a critical feature of human PDAC[53,54]. Genetically engineered mouse models, such as the KPC mouse model, overcome this limitation by recapitulating key aspects of the immune microenvironment observed in human PDAC[55,56]. Loss of *Brca1* in KPC has been reported to confer susceptibility to PARPi[57]. To investigate the role of OFD1 in the *KRAS*-driven spontaneous PDAC model, we generated a stable *Ofd1* knockout cell line derived from KPC1199 cancer cells, originating from pancreatic tumors of KPC mice[58], and reduced *Brca1* mRNA expression was also observed in *Ofd1*-knockout KPC1199 cells (Supplementary Fig. 9b, c). Xenograft tumors derived from KPC1199 cells with *Ofd1* knockout treated with olaparib showed a marked reduction in tumor growth and volume (Supplementary Fig. 9d–f). To further investigate the role of OFD1 in PDAC development, we generated KPCO mice (Fig. 7i). Tumor-bearing KPC or KPCO mice were treated with either vehicle or olaparib. In KPC mice, olaparib treatment alone did not improve survival compared to the vehicle, and *Ofd1* knockout alone slightly prolonged survival. Notably,

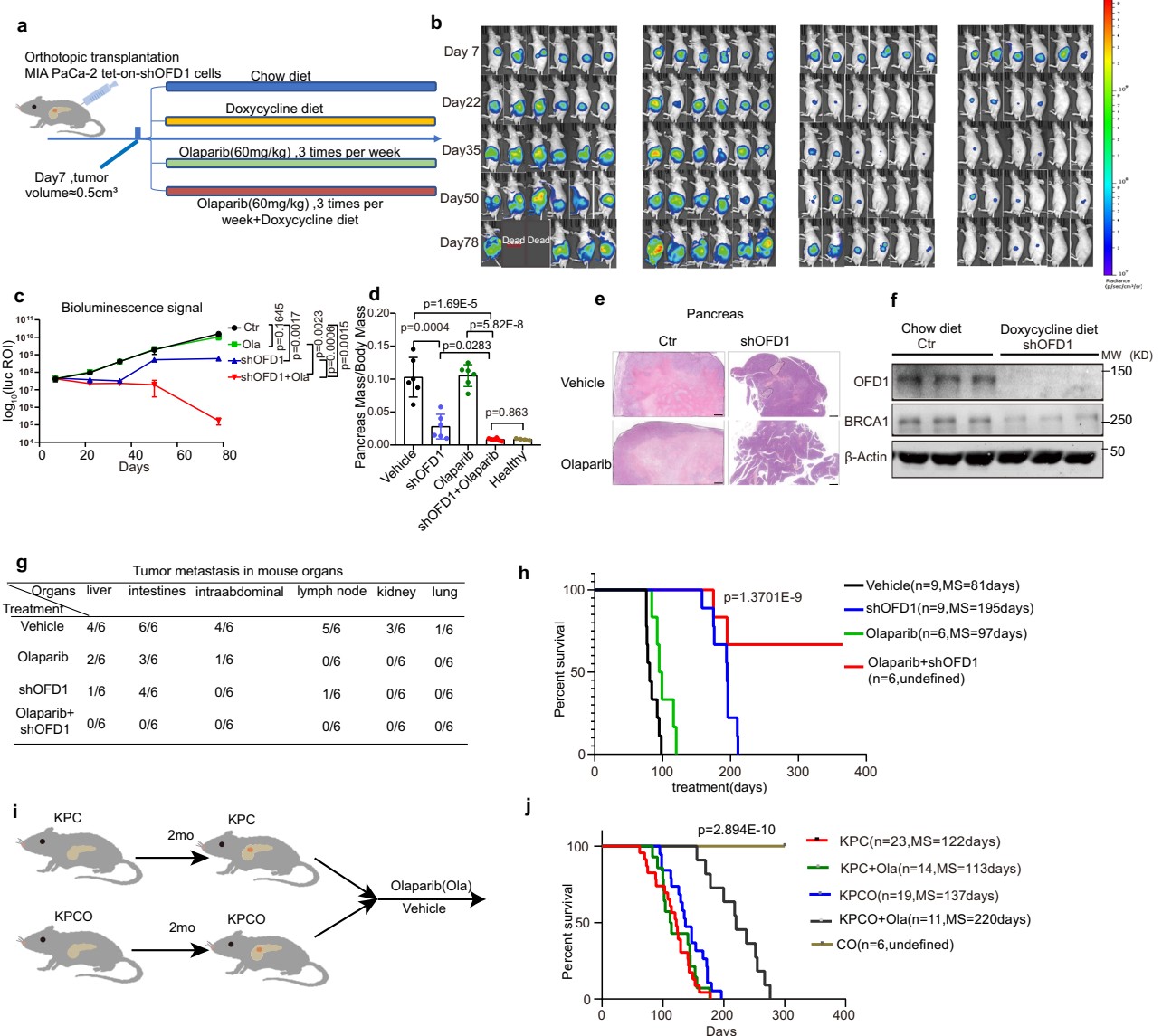

**Fig. 7 | Combined inhibition of OFD1 and PARP causes tumor regression and prolongs survival in spontaneous and orthotopic pancreatic cancer models.**
**a** Experimental design and treatment schedule. **b, c** Tumor growth was monitored by bioluminescence imaging. **b** Luciferase signals were measured on Days 7, 22, 35, 50, and 78. **c** Tumor growth curves were plotted based on bioluminescence intensity (*n* = 6 mice per group). Data are presented as mean values ± SD, Statistical significance at Day 78 was assessed by two-way ANOVA with Tukey's multiple comparisons test. **d** Pancreas-to-body mass ratio at the experimental endpoint. Data are presented as mean values ± SD, unpaired two-tailed *t* tests. Group sizes: vehicle (*n* = 6 mice), shOFD1 (*n* = 6 mice), olaparib (*n* = 6 mice), shOFD1 + olaparib (*n* = 6 mice), Healthy control (*n* = 4 mice). **e** Representative H&E staining of pancreatic tissues from each treatment group. Images were taken at the experimental endpoint. Scale bar = 200 μm. **f** Western blot analysis of OFD1 and BRCA1 protein levels in pancreatic tumors to evaluate the effect of doxycycline-induced OFD1

knockdown (*n* = 3 mice per group). **g** Quantification of metastatic burden. The ratio of tumor metastasis versus major internal organs was calculated in each group after mice were sacrificed at the endpoint. **h** MIA PaCa-2 tumor-bearing mice were treated with vehicle (*n* = 9 mice), shOFD1 (*n* = 9 mice), olaparib (*n* = 6 mice), or shOFD1 + olaparib (*n* = 6 mice) for up to 365 days (1 year). Overall survival was analyzed using Kaplan-Meier survival analysis and the Log-rank (Mantel-Cox) test. **i** Schematic representation of the KPCO mouse model of pancreatic cancer, the model utilizes *LSL-Kras*[G12D/+] (K), *Trp53*[R172H/+] (P), and *Pdx1-Cre* (C) alleles. Pancreas-specific depletion of OFD1 was achieved by crossing KPC with *Ofd1*[flox/flox] mice, generating the KPCO model. **j** Kaplan-Meier survival analysis: Survival comparison of KPC (*n* = 23 mice, MS = 122 days), KPCO (*n* = 19 mice, MS = 137 days), KPC + olaparib (*n* = 14 mice, MS = 113 days), KPCO + olaparib (*n* = 11 mice, MS = 220 days), and CO (*n* = 6 mice). Statistical significance was assessed using the Mantel-Cox test. Source data are provided as a Source Data file.

the combination of *Ofd1* knockout and olaparib significantly extended survival in KPC mice (Fig. 7j). These results demonstrated that the synergistic effect of OFD1 inhibition and PARPi is also applicable in immunocompetent PDAC mouse models.

### Targeting OFD1 synergizes with olaparib in pancreatic PDX models and other BRCA-associated cancers
PARPi are effective only in a small subset of pancreatic cancers that harbor *BRCA1/2* mutations, as the majority of cases remain resistant to

PARPi due to proficient BRCA1 activity. Although OFD1 is not directly involved in DNA damage repair, it regulates *BRCA1* transcription in a stress-inducible manner, making it a promising target for cancer therapy. We evaluated OFD1 expression in human tumor tissues and peri-cancerous tissues surgically resected from 90 PDAC patients using IHC with validated OFD1 and BRCA1 antibodies. The PDAC cohorts were classified into strong, moderate and weak expression groups based on their IHC intensity and area (Fig. 8a), OFD1 and BRCA1 expression showed a strong correlation in these pancreatic cancer

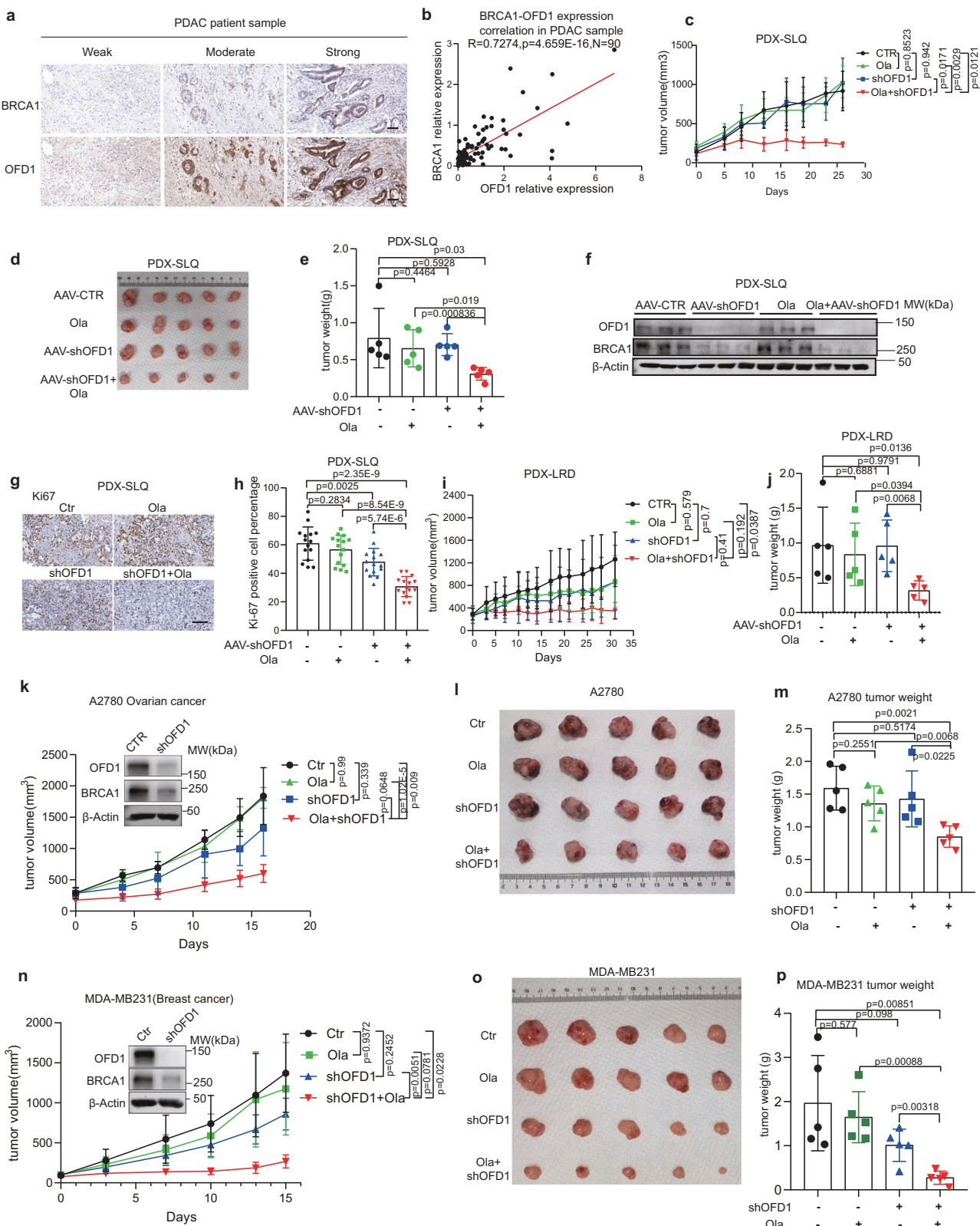

samples (Fig. 8b). PDX models are highly representative preclinical models that closely reflect the phenotypic and therapeutic characteristics of pancreatic cancer[59]. To investigate the potential synergistic effect of OFD1 inhibition and olaparib treatment in pancreatic cancer, we used two pancreatic PDX models derived from patients with *BRCA1/2* wild-type tumors. Once tumors reached approximately 100 mm³, animals were randomly assigned to one of four groups: vehicle control,

intratumoral injection of AAV-shOFD1, oral administration of olaparib, or a combination of AAV-shOFD1 and olaparib. Among these groups, only the combination treatment resulted in significant and rapid tumor regression in both PDX models (Fig. 8c–j and Supplementary Fig. 10a).

Sensitivity to PARPi has been observed in tumor types associated with heritable *BRCA1/2* mutations, including pancreatic, breast, ovarian, and prostate cancers[1]. To explore the applicability of OFD1

**Fig. 8 | The combination of OFD1 knockdown and PARP inhibition effectively suppresses tumor growth in PDAC PDX and other BRCA- associated cancer xenografts. a, b** Representative IHC images and quantification of OFD1 and BRCA1 expression in human PDAC tissues (*N* = 90 patients). Scale bar, 50 µm. Pearson correlation coefficient (two-sided test) was used to assess the relationship between OFD1 and BRCA1 levels. **c**–**e** Effect of olaparib, shOFD1 (AAV-shOFD1 intratumoral injection), and combination treatment on PDAC PDX-SLQ. **c** Tumor growth curves (mean ± SD, *n* = 5 mice), analyzed by two-way ANOVA with Tukey's post hoc test. **d** Representative endpoint tumor images. **e** Tumor weights (mean ± SD, *n* = 5 tumors), analyzed by unpaired two-tailed *t*test. **f** Immunoblot analysis of OFD1 and BRCA1 protein levels in PDX-SLQ tumors. α-Tubulin was used as a loading control. **g**, **h** Ki-67 immunostaining in PDX-SLQ tumors. **g** Representative IHC images of Ki-67 expression. Scale bar = 100 µm. **h** Quantification of Ki-67-positive cells was performed from three random fields per mouse (*n* = 5 mice per group), mean values ± SD, unpaired two-tailed *t* tests. **i, j** Tumor growth and weight in PDAC PDX model (PDX-LRD). **i** Tumor growth curve was plotted and analyzed using two-way ANOVA followed by Tukey's post hoc test (*n* = 5 mice), mean values ± SD. **j** Tumor weight was measured at endpoint and analyzed using unpaired two-tailed *t* tests (*n* = 5 tumors), mean values ± SD. **k**–**m** Efficacy of combination therapy in an A2780 ovarian cancer xenograft model. **k** Tumor volumes were measured over time and analyzed by two-way ANOVA followed by Tukey's post hoc test (*n* = 5), mean values ± SD. **l** Representative images of harvested tumors at endpoint (*n* = 5 tumors). **m** Tumor weights were measured and analyzed using unpaired two-tailed *t*-tests (*n* = 5). mean values ± SD. **n**–**p** Combination therapy efficacy in the MDA-MB-231 TNBC xenograft model. **n** Tumor volumes were measured over time and analyzed by two-way ANOVA followed by Tukey's post hoc test (*n* = 5 mice), mean values ± SD. **o** Representative endpoint tumor images (*n* = 5 tumors). **p** Tumor weights analyzed by unpaired two-tailed *t* test (*n* = 5 tumors). mean values ± SD. Source data are provided as a Source Data file.

inhibition in other BRCA-associated cancer types with *BRCA1/2* wild-type genotypes, we generated OFD1-inducible knockdown cell lines in MDA-MB231 breast cancer cells, PC3 prostate cancer cells, and A2780 ovarian cancer cells. Nude mice subcutaneously injected with these cell lines were treated with either vehicle control, olaparib alone, doxycycline alone (to induce OFD1 shRNA expression), or a combination of doxycycline and olaparib. Across all tumor types, the combination of doxycycline and olaparib showed the most pronounced tumor regression compared to that of the other groups, demonstrating a strong synergistic effect of OFD1 inhibition and PARPi on BRCA-associated tumors (Fig. 8k–p, Supplementary Fig. 10b, c).

## Discussion

PDAC is notorious for its malignancy and refractoriness, with a 5-year overall survival rate less than 10 % across all stages[60]. Current therapeutic strategies, including chemotherapy and surgical resection, are hindered by challenges such as recurrence, drug resistance, and severe side effects[8]. These limitations underscore the urgent demand for novel therapeutic approaches, particularly targeted therapies and immunotherapies. Genetic mutations in DNA damage response genes, especially those involving in the HRR pathway, give rise to an HRR-deficient phenotype, commonly referred to as "BRCAness". This molecular vulnerability has prompted the investigation of PARPi as a potential treatment option for PDAC[7]. A significant breakthrough in this field is the FDA's approval of olaparib as a first-line maintenance therapy for patients with metastatic PDAC harboring germline *BRCA1/2* mutations. However, *BRCA1/2* mutations are relatively uncommon in PDAC, with *BRCA2* mutations being the most frequent (1.4%–7%), followed by mutations in *BRCA1* and *PALB2*[61]. Consequently, only 5–7% of PDAC patients care urrently eligible for and benefit from PARPi therapy[62], emphasizing the importance of extending its applicability to HRR-proficient tumors[20,63]. The concept of 'BRCAness', therefore, provides a critical conceptual framework for developing therapeutic strategies aimed at sensitizing BRCA-proficient tumors to PARPi therapy, representing an area with substantial potential for future translational research.

In this study, we observed a marked overexpression of OFD1 in pancreatic cancer tissues compared with adjacent normal tissues, as confirmed by both tissue microarrays from PDAC patients and TCGA datasets. By screening an FDA-approved drug library, we identified several compounds that exhibit synthetic lethal interactions when combined with OFD1 inhibition. Notably, PARPi, including olaparib and rucaparib, emerged as top candidates. Mechanistically, knockdown of OFD1 induces the nuclear translocation of E2F4 and facilitates the assembly of the DREAM complex on the *BRCA1* promoter, thereby repressing *BRCA1* transcription (model illustrated in Supplementary Fig. 10e). The DREAM complex and E2F4 are critical for the transcriptional repression of *BRCA1*, as previously demonstrated in studies[41,47,48]. and dysregulated *BRCA1/2* expression has been shown to

enhance sensitivity to PARPi[47]. Our findings demonstrate that targeting OFD1 in combination with olaparib leads to a synergistic anti-tumor effect in various PDAC models, including xenografts, orthotopic models, PDX and spontaneous KPC models. This synergy also extends to other BRCA-associated cancers, including breast, prostate, and ovarian cancer. These findings emphasize the therapeutic potential of developing OFD1 inhibitors for combination therapies with PARPi in both PDAC and other BRCA-associated cancers.

Mutations in OFD1 are associated with several human genetic disorders, including oral-facial-digital syndrome type 1, Joubert syndrome (JBS10), and primary ciliary dyskinesia (PCD)[28,64–66]. Genetic ablation of *Ofd1* in mice leads to embryonic lethality[67]. However, conditional knockout of *Ofd1* in adult mouse pancreas had minimal effect on lifespan (Fig. 7j), suggesting a potential therapeutic window for targeting OFD1 in cancer therapy. In this study, we reveal the unanticipated role of OFD1 inhibition in inducing the BRCAness phenotype in cancer. Interestingly, Abramowicz et al. also reported that OFD1 deficiency impairs HRR in fibroblasts derived from patients with oral-facial-digital syndrome type I[68], further reinforcing OFD1's involvement in HRR. In addition, we observed a strong positive correlation between OFD1 and BRCA1 expression in PDAC samples. *BRCA1*, *BRCA2*, and *PALB2* are critical genes involved in DNA maintenance and are targets of the DREAM complex. In pancreatic cancer, mutation frequencies of these genes are as follows: *BRCA2* (3.9%), *BRCA1* (1.2%), and *PALB2* (0.9%)[7]. Mutations in these genes compromise the HRR pathway, which is essential for the therapeutic efficacy of PARPi in PDAC[69]. Notably, among these genes, only *BRCA1* was consistently and significantly downregulated across all tested cell lines, emphasizing OFD1's prominent role in regulating BRCA1 expression in PDAC.

In conclusion, our study demonstrates that OFD1 inhibition triggers the BRCAness phenotype, with significant implications for sensitizing BRCA1/2 proficient cancers to PARPi. Impaired BRCA1 expression, which enhances sensitivity to PARPi, has been observed in multiple cancer types[47,70–72]. Given the unique and robust regulation of *BRCA1* expression by OFD1, targeting OFD1 offers a promising strategy for extending the use of PARP inhibitor therapy to *BRCA1* wide-type cancers. This approach opens emergent therapeutic opportunities with substantial clinical potential, offering hope for more effective treatment options across a broader spectrum of cancers.

## Methods
### Ethical statement
This study complies with all relevant ethical regulations and was approved by the Ethics Committee of Ruijin Hospital and Shanghai Jiao Tong University School of Medicine. All animal study protocols were approved by the Animal Research Committee of Shanghai Jiao Tong University School of Medicine (Reference number: DLAS-MP-ANIM.11). For the patient-derived xenograft study, written informed consent was obtained from all participants, and the study was approved by the

Ethics Committee of Ruijin Hospital (Reference number: 2013-70). Pancreatic cancer and normal tissue microarrays (TMA, HPa-nA180Su03) were prepared by Shanghai Outdo Biotech. The collection and preparation of these tissues were approved by the Scientific Investigation Board of Taizhou Hospital in accordance with the Declaration of Helsinki (Reference number: SHYJS-CP-1901009). Written informed consent was obtained from all human participants prior to sample collection.

## Cell Lines

The cell lines used in this study (PANC1, SW1990, PATU8988T, MIA PaCa-2, PANC1005, HPAC, HPAFII, BXPC3) were all purchased from the American Type Culture Collection (ATCC). The murine-derived KPC1199 cell line originated from a spontaneous PDAC mouse model KPC ($LSL$-$Kras^{G12D/+}$; $Trp53^{R172H/+}$; $Pdx1$-$cre$) on a C57BL/6 genetic background[58], and was generously provided by Dr. Jing Xue (Renji Hospital, Shanghai Jiao Tong University School of Medicine, Shanghai, China). All cell lines were maintained in the recommended medium according to the supplier's protocol, supplemented with 10% fetal bovine serum (FBS) and 1% penicillin/streptomycin.

## Gene silencing by siRNA and shRNA

The small interfering RNA (siRNA) oligos were transfected into the indicated cell lines using RNAi Max (Invitrogen, Cat#13778150). Specific details regarding the siRNA and shRNA oligos used are provided in the key resources table.

For doxycycline-inducible knockdown of OFD1, a Tet-pLKO-shOFD1 construct was generated by inserting OFD1-targeting shRNA sequences (shOFD1#1: 5'-GAACGAAGAGAACTAGAAA-3'; shOFD1 #2: 5'-CGAAAAGGCTATAGTGGTT-3') into the Tet-pLKO-puro vector (Addgene, Cat#21915), which had been digested with AgeI (NEB, Cat#R3552S) and EcoRI (NEB, Cat#R3101S) restriction enzymes and dephosphorylated for 30 minutes at 37°C. The digested plasmid was run on a 1% agarose gel, excised, and purified using the FastPure Gel DNA Extraction Mini Kit (Vazyme, Cat#DC301-01). The oligonucleotides were phosphorylated using T4 PNK with T4 Ligation Buffer (NEB, Cat#M0201S). The samples were annealed in a thermocycler at 37°C for 30 min, then at 95 °C for 5 min, and finally were gradually frozen to 25 °C at 5 °C/min. The annealed oligonucleotides were then diluted 1:200 in RNase/DNase-free water. Ligation of the annealed oligonucleotides into the digested Tet-pLKO-puro vector was performed using Quick Ligase (NEB, Cat#M0202S). To induce OFD1 knockdown, cells stably infected with Tet-shOFD1 virus were treated with 100 ng/mL of Doxycycline (Sigma, Cat#324385).

## PDAC Patient-derived xenograft (PDX) models

The human PDAC tissue samples used in this study were obtained from patients who underwent surgical resections at Ruijin Hospital affiliated to Shanghai Jiao Tong University School of Medicine (Shanghai, China). The study was approved by the Ethics Committee of Ruijin Hospital and Shanghai Jiao Tong University School of Medicine (reference number: 2013-70), and all participants provided informed consent. Freshly resected pancreatic tumor specimens were collected in DMEM medium without FBS and kept on wet ice for engraftment within 24 hours after resection. Approximately 1 mm³ tumor tissue was subcutaneously implanted into the flank region of athymic 6- to 8-week-old male BALB/c nu/nu mice using a trocar. A portion of tumors was subjected to BRCA1/2 exon sequencing analysis. For tumor subculture, once the primary xenograft reached a volume of ~600 mm³, mice were anesthetized and tumor tissues were collected for serial transplantation to the next generation of recipient mice. In the therapeutic experiment, once tumors reached approximately 100 mm³, animals were randomly assigned to four experimental groups: control, shOFD1, olaparib, shOFD1 + olaparib. Mice were treated with olaparib (60 mg/kg) or vehicle three times a week, in combination with intratumoral injection of AAV-shOFD1 or AAV-shCtr. Tumor volume and body weight were measured twice a week. The tumor volume was limited to no more than 3000 mm³. At the experiment endpoint, all tumors were harvested and weighed. Mice number in each group has been indicated in the relevant figure legend. Tumor volumes were calculated using the formula: Tumor volume (mm³) = $0.52 \times$ (length × width²).

## Screening of OFD1-targeted synthetic lethality inhibitors

The FDA- approved compound library was purchased from Selleck Chemicals (Cat# L1300). MIA PaCa-2 and MIA PaCa-2-shOFD1 cells were cultured in DMEM medium with 10% FBS and 1% penicillin/streptomycin. A total of 2000 cells per well were seeded into 96-well plates, after cell attachment, MIA PaCa-2 and MIA PaCa-2-shOFD1 cells were treated with either DMSO or small-molecule inhibitors at varying concentrations (0, 1, 5, 10, 25 and 50 μM). After 72 h, cell viability was assessed using the Cell Counting Kit-8 assay. For compounds exhibiting low IC50 values, additional concentrations around the estimated IC50 were included at lower dose ranges to enhance resolution. Conversely, for compounds with higher IC50 values, supplementary higher concentrations were incorporated to ensure accurate curve fitting and reliable IC50 estimation. Dose-response data were analyzed using GraphPad Prism 8 (nonlinear regression, variable slope, four parameters, bottom constant equal to 0, top constant equal to 1). $LogIC_{50}$ values were statistically compared using the extra sum-of-squares F Test.

## Subcutaneous and orthotopic xenograft model

All animal experiments were conducted in the SPF animal facilities at Shanghai Jiao Tong University School of Medicine. The study was approved by the Ethics Committee of Ruijin Hospital and Shanghai Jiao Tong University School of Medicine (policy number: DLAS-MP-ANIM.11). For the subcutaneous xenograft model, 6- to 8-week-old athymic male nu/nu mice were acclimatized in the mouse room for one week. A total of $5 \times 10^6$ SW1990-Tet-on-shOFD1 or PC3-Tet-on-shOFD1 cells in 100 μL PBS were injected subcutaneously into the right flank of each nude mice. In addition, 6- to 8-week-old athymic female nu/nu mice were used for the subcutaneous xenograft models of A2780-Tet-on-shOFD1 cells or MDA-MB-231-Tet-on-shOFD1 cells with the identical implantation and assignment protocol. Once tumors reached approximately 100 mm³, animals were randomly assigned to four experimental groups: control, shOFD1, olaparib, shOFD1 + olaparib. Mice in the shOFD1 and shOFD1 + olaparib group were fed a chow diet supplemented with 500 mg/kg doxycycline, while mice in the olaparib and shOFD1 + olaprib group were treated with olaparib (60 mg/kg) or vehicle three times a week via oral lavage. Tumor volume was measured twice a week and was limited to no more than 3000 mm³. After 4 weeks of treatments, the mice were sacrificed, and tumors were collected and fixed in 4% paraformaldehyde for subsequent histological analysis. Mice number in each group has been indicated in the relevant figure legend. Tumor volumes were calculated using the formula: Tumor volume (mm³) = $0.52 \times$ (length × width²).

Subcutaneous xenograft model of KPC1199-sgCtr and KPC1199-sgOfd1 cells were generated and treated with the same protocol.

For the orthotopic xenograft study, $10^6$ MIA PaCa-2-Tet-on-shOFD1 cells with stable luciferase expression were transplanted into the pancreatic tail of 6- to 8-week-old athymic male nu/nu mice. One week later, the mice were intraperitoneally injected with 100 mg/kg of luciferin (200 μL of 10 mg/mL luciferin in PBS) 12 to 15 min before imaging, then were anesthetized with isoflurane, and imaged using IVIS (Xenogen Spectrum). For tumor burden analyses, Living Image version 4.4 (Caliper Life Sciences) was used to quantify tumor burden. A circular region of interest (ROI) was set around the pancreas and tumor within each experimental group. Mice were randomly assigned to four experimental groups: control, shOFD1, olaparib, shOFD1 + olaparib.

For the groups treated with doxycycline or olaparib, mice were fed with the dosing regimen consistent with the subcutaneous model. Mice number in each group has been indicated in the relevant figure legend. Bioluminescent imaging was performed every three weeks. The endpoint of survival monitoring was defined as a body weight loss of more than 15% (approximately 4 g) compared to healthy controls, or showed obvious signs of reduced mobility and hunching, or developed ascites. And the survival data were recorded and analyzed using GraphPad Prism 8.

### Transgenic Animal Model

The *Pdx1-Cre*, *LSL-Kras^{G12D/+}*, and *TrpS3^{R172H/+}* genetically engineered mice were obtained from The Jackson Laboratory (Bar Harbor, ME) and were generously provided by Dr. Zhigang Zhang (Renji Hospital, Shanghai Jiao Tong University School of Medicine, Shanghai, China). The *Ofd1(flox/flox)* mice were generated using ES gene-targeting technology, and subsequently hybridized with the KPC mouse model to produce the desired transgenic cohort. Mice were euthanized upon exhibiting more than 15% body weight loss, abdominal distension (ascites), labored breathing, or markedly reduced activity, in accordance with institutional animal care guidelines.

### Cell proliferation assay

Cell proliferation assay was assessed using the Cell Counting kit 8 (CCK8, Dojindo, Japan). Briefly, 1000 cells per well were seeded into 96-well plate and treated with varying concentration of olaparib in the presence of doxycycline for 96 h. Cell viability was assessed using the CCK8 assay according to the manufacturer's instructions.

### Colony formation assay

Cells were seeded in 6-well plates at a density of $2 - 8 \times 10^3$ cells per well, depending on the growth rate of each cell line. Cells were cultured in medium supplemented with the indicated drug treatment for 10 to 14 days, with the medium replaced twice a week. Afterward, cells were fixed with methanol and stained with 0.1% crystal violet diluted in methanol. The extent of cell confluence in each well was quantified using ImageJ software.

### Quantitative real-time PCR analysis

Total RNA from cells was extracted using the RNA Isolation Kit V2 (Vazyme, Cat#RC112-01) and subsequently reverse-transcribed into cDNA by Reverse Transcription Kit (Vazyme, Cat# R233-01) according to the manufacturer's protocols. All experiments were performed in biological triplicate. Quantitative real-time PCR was carried out using SYBR Green qPCR Master Mix (Vazyme, Cat#Q111-02) on the Applied Biosystems Quant Studio5 Real Time PCR System. The relative gene expression values of target genes in each sample were normalized to the corresponding expression of ACTB. The 2-DDCt method was employed to calculate fold changes in gene expression. The gene primer list is provided in the supplementary materials.

### Western blotting assay

Cultured cells were lysed in RIPA buffer (Pierce, Thermo Fisher Scientific, Cat#87787), and protein concentration was determined using the BCA Protein Assay Kit (Pierce, Thermo Fisher Scientific, Cat#23227). Cell lysates were mixed with the 1 × SDS sample buffer and boiled for 10 min at 99 °C. Proteins were separated by SDS-PAGE and transferred to a PVDF membrane. The membrane was blocked with PBST containing 5% milk for 1 h at room temperature and then washed with PBST for three times. Then the membrane was incubated with the appropriate dilution of primary antibody (OFD1, 1:50000; E2F4, 1:2000, Abclonal, Cat#19670, 1:2000; BRCA1, CST, Cat#9010, 1:1000; RBL2, CST, Cat#13610, 1:1000; RBBP4, Abclonal, Cat#A3645, 1:2000; alpha- Tubulin, Proteintech, Cat#66031-1-Ig, 1:10000; Flag, Sigma, Cat#A8592, 1:10000; GFP, Proteintech, Cat#50430-2-AP, 1:3000) at

4 °C overnight. The membrane was subsequently washed three times with PBST before being incubated with HRP-conjugated anti-rabbit or anti-mouse (diluted 1:10000) secondary antibodies for 1 h at room temperature. After three washes with PBST, the immunoblots membrane were visualized by enhanced chemiluminescence in the ChemiDoc Touch Image system (Bio-Rad). The immunoblotting results were quantified using Image J (NIH).

### Immunofluorescence and Quantification

For immunofluorescence analysis, cells were seeded on coverslips and washed three times with PBS. For ionizing radiation experiments, cells were fixed with 4% paraformaldehyde for 30 minutes at room temperature 4 h after treatment with 10 Gy. Permeabilization was performed using Blocking Buffer (0.5% Triton X-100, 2.5% BSA) in PBS for 30 min. Relevant primary antibodies, diluted in blocking buffer, were added and incubated overnight. Cells were washed three times for 10 minutes with PBS buffer and incubated with Alexa Fluor 488-labeled anti-rabbit (Thermo Fisher Scientific, Cat#A-11008) or Alexa Fluor 594-labeled anti-mouse antibodies (Thermo Fisher Scientific, Cat#A-11012) at 1:1000 dilution for 1 h at room temperature. After washing three times for 10 minutes with PBS buffer, cells were stained with DAPI (Thermo Fisher Scientific, Cat# D1306) at a 1:10000 dilution for 5 min, the coverslips were mounted with anti-fade mounting medium (Merck, Cat#345789), and slides were examined by a laser scanning confocal microscope (Olympus IX83).

### Proximity ligation assay

PANC1 cells expressing GFP-OFD1 were seeded onto coverslips. The PLA assay (Sigma-Aldrich, Cat#DUO92007) was performed according to the manufacturer's instructions, using primary antibodies against GFP (Abclonal, Cat#AE012, 1:500) and E2F4 (CST, Cat#40291, 1:500). Samples were visualized using a laser scanning confocal microscope (Olympus IX83). The total intensity of the PLA signal per cell was quantified using Fiji software.

### Metaphase spreads

For quantification of chromosome breaks, metaphase spreads were performed. Cells were exposed to 0.2 µg/mL Colcemid (TargetMol, Cat#T19720) for 2 h at 37 °C. Following treatment, cells were fixed with Carnoy fixative (3:1 methanol:acetic acid) and incubated at 4 °C overnight. Fixed cells were dropped onto uncoated microscope slides and air-dried for 24 h at room temperature. Metaphases were prepared and stained with Giemsa Dye (Sangon Biotech, Cat#A600477). After mounting with Permount (Electron Microscopy Sciences, Cat#17986), metaphase spreads were visualized using a microscope (Olympus IX83).

### Immunoprecipitation

For exogenous protein immunoprecipitation (IP), HEK293T cells were transfected with 6-10 µg of the indicated plasmids and cultured for 48 h. Whole-cell lysates were extracted on ice using TAP lysis buffer (20 mM Tris-HCl pH = 7.5, 150 mM NaCl, 0.5% NP-40, 1 mM EDTA, 1/100 protease inhibitor cocktail (APExBIO, Cat#K1007) for 30 min, followed by centrifugation at $15,000 \times g$ for 30 min. 5−10 µL of anti-Flag M2 Affinity Gel (Sigma, Cat#A2220) was added to the supernatant of each sample, followed by incubation at 4 °C for 3 h. The precipitates were washed five times with TAP lysis buffer and eluted with 0.2 mg/mL Flag peptide diluted in TAP lysis buffer. The eluted samples were analyzed using immunoblotting.

For IP analysis of endogenous protein-protein interactions, whole-cell lysates extracted from stable pancreatic cancer lines (PANC1-Tet-shOFD1, MIA PaCa2-Tet-shOFD1, SW1990-Tet-shOFD1 and PATU8988T-Tet-shOFD1) were incubated overnight with protein G Magnetic beads (Cell Signaling, Cat#D70024) and antibodies against IgG (Abclonal, Cat#AC005), OFD1 (1:1000), or E2F4 (CST, Cat#40291,

1:100). After washed five times with TAP lysis buffer, the precipitates were eluted with SDS loading buffer and analyzed by immunoblotting.

## Recombinant protein purification

OFD1-Flag and E2F4-Strep II-HA recombinant proteins are expressed in HEK293S cells. After 48 hours, cells were collected by centrifugation (500 × g, 15 min, 4 °C), and the pellet was resuspended in lysis buffer (50 mM Tris-HCl, pH 7.5, 150 mM KCl, and 1 mM DTT, 1 mM EDTA, 1.5% Triton X-100, 1/100 protease inhibitor cocktail (APExBIO, Cat#K1007)). The supernatant was separated from cell debris by centrifugation at 14,000 × g for 30 min at 4 °C, and then incubated with Flag agarose (Sigma-Aldrich, Cat#A2220) or StrepTactin XT beads (IBA, Cat#2-5030-025) equilibrated with lysis buffer, for 5 h at 4 °C. After binding, the beads were washed with 10 column volumes of lysis buffer containing 500 mM KCl and 250 mM KCl, followed by elution with 0.2 mg/mL Flag peptide or 50 mM biotin (pH = 8.0). E2F4-Strep II-HA protein eluted by biotin was further incubated with HA agarose (Sigma-Aldrich, Cat#A2095) for a second round of purification and eluted with 0.2 mg/mL HA peptide. Purified proteins were stored in −80 °C in 20% glycerol.

## In vitro pulldown assay

Pull-down assays were performed by incubating 20 μg purified OFD1-Flag and E2F4-Strep II-HA recombinant protein with Flag agarose or HA agarose in TAP lysis buffer in a 1.5 mL microcentrifuge tube for 3-4 h at 4 °C. Then, the beads were washed 3 times with 1 mL of TAP lysis buffer before elution with 1 × SDS loading buffer and analyzed by immunoblotting. For the in vitro peptide competition assay, 5 μg purified OFD1-Flag and E2F4-Strep II-HA recombinant protein were incubated with HA agarose, and a series of candidate peptides were added into the incubation system at a concentration of 200 μM for binding motif screening assay. For the validation assay, the peptide (ELSSLPLGPTTLDTRPLQSS) was supplied at different concentrations (0, 12.5, 25, 50, 100 and 200 μM) into the identical incubation system as mentioned above in a 1.5 mL microcentrifuge tube. An overnight incubation at 4 °C was required for peptide competition assays to disrupt OFD1-E2F4 interaction. After incubation, the beads were washed 3 times with 1 mL of TAP lysis buffer, eluted with 1 × SDS loading buffer, and analyzed by immunoblotting.

## Histology and Immunohistochemistry

Tissues were fixed in 4 % paraformaldehyde and processed for paraffin embedding. Tissue sections, 5 μm in thickness, were cut and stained with hematoxylin and eosin (H&E). For tumor burden quantification, at least six randomly selected fields of view at 200 × magification were analyzed in the stained pancreas and other organs sections[73]. For IHC staining, tissue sections were also subjected to antigen retrieval (15 min in 10 nM citrate buffer at pH = 6, 98 °C) prior to immunostaining, the tissue sections were incubated with blocking buffer for 1 hour before overnight incubation with the primary antibodies (Ki67, Proteintech, Cat#27309-1-AP; γH2AX, CST, Cat#60566). The sections were incubated with biotinylated goat anti-rabbit and streptavidin HRP (Biocare Medical) for 10 min, and counterstained with hematoxylin. DAB positivity was analyzed. Analyses for comparative DAB positivity was performed using ImageJ software by designing a macro to define a positively stained area. At least ten 200 × visual fields were selected for analysis.

Pancreatic cancer sample sections, with corresponding normal tissue microarray (TMA, project number: HPanA180Su03) sections, were prepared by Shanghai Outdo Biotech Co., Ltd. (Shanghai, China). Sample collection and preparation were approved by the Scientific Investigation Board of Taizhou Hospital and were in accordance with the ethical principles originating from the Declaration of Helsinki and Shanghai Outdo Biotech Co., Ltd. (reference number: SHYJS-CP-1901009). These tissue arrays contained tissues from 90 paired pancreatic cancer and normal tissue samples were used to examine the expression profiles of OFD1 and BRCA1 using IHC. For IHC, the tissue sample sections were incubated with anti-OFD1 antibodies at a 1:4000 dilution and mouse anti-BRCA1 monoclonal antibodies (Santa cruz, Cat#sc-6954) a 1:100 dilution. IHC staining were scored by two independent pathologists who were blinded to the clinical characteristics of the patients. Using ImageJ (NIH) IHC scoring system was based on the intensity and extent of staining, as a 3-tier scale (0; negative to weak, 1; moderate, 2; strong). For OFD1 expression and survival analysis, Strong staining was defined as "High", and the rest of the staining was defined as "Low".

## Chromatin immunoprecipitation (ChIP)

ChIP assays were conducted using the EZ-ChIP kit (Millipore, Cat#17-295). Briefly, PANC1 or MIA PaCa-2 cells were cross-linked with 1% formaldehyde for 10 minutes at room temperature, followed by quenching with glycine (final concentration, 0.125 M). These cells were collected using a cell Scraper and washed twice with cold PBS. Cell pellets were lysed with cell lysis buffer supplemented with a protease inhibitor cocktail and incubated on ice for 30 min, after centrifugation at 10,000 × g for 10 min, the pellets were resuspended in nuclear lysis buffer (50 mM Tris, pH = 8.0, 10 mM EDTA, 1% SDS) and subjected to sonication (30 W, 15 s on and 30 s off, 8 times). The supernatant was collected by centrifugation at 10,000 × g for 20 min and 1:10 diluted in dilution buffer (50 mM Tris, pH = 8.0, 150 mM NaCl, 0.5% NP-40, with protease inhibitor cocktail) before immunoprecipitation (IP) with primary antibody against IgG, E2F4 (CST, Cat#40291, 1:100), RBL2 (CST, Cat##13610,1:50-1:100), LIN54 (Bethyl Laboratories, Cat#A303-799A, 1:100) and RBBP4 (GeneTex, Cat#GTX70234, 1:100). The enrichment of 300 - 600 bp of sheared DNA was confirmed by agarose gel electrophoresis. A total of 50 mg of DNA was used for IP. Diluted protein-DNA complexes were precleared and incubated overnight with protein G Dynabeads and antibodies at 4 °C. Immunoprecipitates were washed sequentially with Low salt buffer, High salt buffer, LiCl wash buffer, TE buffer. DNA extraction from immunoprecipitates by incubating with proteinase K at 65 °C for 4 h, followed by inactivation at 95 °C for 10 min. ChIP DNA was purified using a ChIP DNA purification kit (Active Motif, Cat#AM58002) for subsequent ChIP–qPCR (primer sequences are listed).

## RNA-seq

Pancreatic cancer cell lines PANC1-tet-on-shOFD1, MIA PaCa-2-tet-on-shOFD1, and PATU8988T-Tet-on-shOFD1 were treated with DMSO or doxycycline for 72 h to induce OFD1 knockdown. RNA-seq samples were prepared following the methodology described in a previous study[74]. Total RNA was extracted using TRIzol reagent (Invitrogen) following standard protocols. RNA quality and quantity were assessed by NanoDrop 2000 and Agilent 2100 Bioanalyzer. Libraries were prepared using the VAHTS Universal V10 RNA-seq Library Prep Kit and sequenced on an Illumina NovaSeq 6000 platform. Reads were aligned to the reference genome with HISAT2. Gene expression was quantified as FPKM, and read counts were obtained via HTSeq-count. PCA was performed in R to assess sample consistency. Differential expression analysis used DESeq2 with adjusted p-value < 0.05 and fold change > 2 or < 0.5 as significance criteria. Hierarchical clustering and radar plots of top DEGs were generated in R.Functional enrichment (GO, KEGG, Reactome, WikiPathways) was conducted using hypergeometric tests and visualized with various R plots. GSEA was performed to identify enriched gene sets based on ranked gene expression differences. The transcriptome sequencing and analysis were conducted by OE Biotech Co., Ltd. (Shanghai, China).

The analysis of the data was conducted using various software tools and public databases, including STRING for protein-protein interaction (PPI) and GSEA (Gene Set Enrichment Analysis) to identify enriched biological pathways.

## Nuclear and Cytoplasmic fractionation assay

Cell extracts were prepared using the Thermo Scientific NE-PER Nuclear and Cytoplasmic Extraction Reagents cell fractionation kit (Thermo Fisher Scientific, Cat#78833), according to the manufacturer's protocol. For stable PANC1 or MIA PaCa-2 tet-on-shOFD1 cell lines, cells were harvested using trypsin-EDTA and centrifuged at $500 \times g$ for 5 min. The cell pellet was washed with PBS and transferred to a 1.5 mL microcentrifuge tube. After centrifuging at $500 \times g$ for 2-3 min, the supernatant was removed, and the cell pellet was suspended with ice-cold CER I. The tube was vigorously vortexed for 15 seconds and incubated on ice for 10 min. Then, ice-cold CER II was added, vortexed for 5 sec, and incubated on ice for 1 min. After vertexing again for 5 sec, the tube was centrifuged at maximum speed in a microcentrifuge ($-16,000 \times g$) for 5 min. The supernatant (cytoplasmic extract) was transferred to a clean pre-chilled tube and stored on ice until use or storage at $-80\,^{\circ}\mathrm{C}$. The insoluble fraction containing nuclei was suspended in ice-cold NER, vortexed vigorously for 15 sec, and incubated on ice. The tube was vortexed every 10 min for a total of 40 min. Afterward, the tube was centrifuged at $20,000 \times g$ for 10 min, and the supernatant (nuclear extract) was transferred to a clean, pre-chilled tube and stored on ice or stored on ice at $-80\,^{\circ}\mathrm{C}$. All cell extracts were quantified using BCA-based protein assays and subsequently subjected to immunoblotting.

## Comet assay

Neutral comet assays were performed using a commercial comet assay kit (Abcam, Cat#ab238544). Briefly, 75 μL comet agarose was pipetted onto the comet slide to form a base layer, cell samples (~ 8000 cells) were mixed with comet agarose at a 1/10 ratio (v/v) by pipetting and immediately transferred (75 μL/well) onto the top of the comet agarose base layer. The slides were incubated with lysis buffer, followed by alkaline solution. After washing, electrophoresis was performed under neutral conditions, and cells were stained with DNA dye. The slides were visualized using a confocal microscope, and the results were analyzed using ImageJ software.

## HR reporter assay

I-SceI-based reporter assays were employed to measure the efficiency of repair via HR. The pCVL Traffic Light Reporter 1.1 (Sce target) Ef1a Puro and pCVL SFFV d14GFP EF1s HA.NLS.Sce (opt) plasmids were generously provided by Andrew Scharenberg (Addgene plasmid Cat#31482; Cat#31476)[74]. pCVL Traffic Light Reporter was packaged into lentivirus and transduced into the pancreatic cancer cell line. Stable cell lines were selected by puromycin for 1 week, followed by transfection with the I-SceI expression plasmid. After treatment, cells were harvested and analyzed by flow cytometry (FACS) using LSR II (BD Bioscience). At least 100,000 cells were counted.

## Statistics and reproducibility

Statistical significance was determined by specific tests and is presented as means ± SD as indicated in the figure legends. Statistical analyses were performed using GraphPad Prism 8.0 (GraphPad Software). Unpaired or paired two-tailed student's $t$ test were used when comparing data from two groups. Statistical analysis for multiple group comparisons was conducted using two-way ANOVA followed by Tukey's post hoc test. For survival analyses, Kaplan-Meier plots were generated, and the statistical differences evaluated using the Log-rank (Mantel-Cox) test. Statistical parameters and significance ($P$-value) are reported in the Figures or the Figure Legends. A $p$-value < 0.05 was considered statistically significant. All microscopic, biochemical, and biological assays were independently repeated at least three times. For RNA-seq data, a Wilcoxon Rank sum test or Kruskal-Wallis test was applied.

## Reporting summary

Further information on research design is available in the Nature Portfolio Reporting Summary linked to this article.

## Data availability

The raw RNA-seq data generated in this study have been deposited in the NCBI Gene Expression Omnibus (GEO) under accession number GSE282746. To compare OFD1 expression between pancreatic cancer and normal tissues, we used the following publicly available GEO datasets: GSE15471[31] (https://www.ncbi.nlm.nih.gov/geo/query/acc.cgi?acc=GSE15471); GSE71729[34] (https://www.ncbi.nlm.nih.gov/geo/query/acc.cgi?acc=GSE71729); GSE62452[35] (https://www.ncbi.nlm.nih.gov/geo/query/acc.cgi?acc=GSE62452); GSE272362[32] (https://www.ncbi.nlm.nih.gov/geo/query/acc.cgi?acc=GSE272362); GSE71989[36] (https://www.ncbi.nlm.nih.gov/geo/query/acc.cgi?acc=GSE71989); GSE63158[33,37] (https://www.ncbi.nlm.nih.gov/geo/query/acc.cgi?acc=GSE63158). Drug response correlation data with OFD1 mRNA expression were obtained from the Cancer Therapeutics Response Portal (CTRP v2) (https://www.broadinstitute.org/cancer-therapeutics-response-portal). All other data supporting the findings of this study are available within the Article, Supplementary Information, or Source Data files. Source data are provided with this paper.

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

## Acknowledgements

We thank Dr. Zhigang Zhang for providing the *LSL-Kras^{G12D/+}; p53^{R172H/+}; Pdx-1-Cre (KPC)* genetically modified mice. The work was supported, in part, by grants from the National Key Research and Development Program of China (2023YFA0914900) to Q.Z.; the National Natural Science Foundation of China (32361163613, 92254307 and M-1040) to Q.Z.; the National Natural Science Foundation of China (82473225) to Z.T.; the Shanghai Frontiers Science Center of Cellular Homeostasis and Human Diseases; the Fundamental Research Funds for the Central Universities, the innovative research team of high-level local universities in Shanghai (SHSMU-ZDCX20212000); and Shanghai Science and Technology Commission (20JC1410100). The authors declare no competing financial interests.

## Author contributions

P.L., J.Y., and Q.Y. started the project and performed most of the experiments; Y.P., C.L., and X.Z. contributed with cell culture experiments; H.L. and W.N. contributed with animal experiments; L.J. and B.S. helped with the PDX experiments and PDAC tumor sample preparation; P.L., J.Y., Q.Y., Z.T., and Q.Z. edited the manuscript; P.L., J.Y. and Q.Y., Z.T., and Q.Z. conceived the project, designed the experiments, analyzed the data and wrote the manuscript with the help of all authors. All authors discussed the results and commented on the manuscript.

## Competing interests

The authors declare no competing interests.
