## [Transparent Peer Review file · Nature Communications]

OFD1 Restraint Creates Therapeutic Vulnerability for PARP inhibitor in Pancreatic Cancer

Corresponding Author: Professor Qing Zhong

Version 0:

Reviewer comments:

Reviewer #1

(Remarks to the Author)

OFD1 gene is associated with X-chromosome linked orofacial developmental syndrome and is functionally essential for ciliogenesis. Additionally, less studied roles of OFD1 in cell cycle and DNA repair have been described by several groups, including the contributing authors of this study. The manuscript proposes a novel mechanism of regulation of BRCA1 expression of OFD1 that could have implications for the DSB DNA repair and cancer cell sensitivity to PARP inhibitor drugs. The main finding of this study is that depletion of OFD1 in pancreatic cancer cell lines significantly increases sensitivity to PARPi both in vitro and in vivo, resulting in improved survival of the tumor bearing animals. This effect was consistently observed in tumor xenograft models, mouse KPC model of pancreatic cancer, and PDX models, suggesting that the status of OFD1 could have clinical significance for predicting the PARPi response. The authors further propose a mechanism where OFD1 regulates expression of BRCA1 gene at the transcript level and provide convincing evidence of this relationship. The effect of OFD1 depletion on DNA repair is consistent with previously observed defect in the DNA repair in the cells obtained from individuals with OFD1 syndrome (PMID: 27798113). However, the previous study proposed a different mechanism for this phenotype focused on the interaction between OFD1 and TIP-60 histone acetyltransferase complex, which was not explored here. Instead, the authors propose that OFD1 promotes the expression of cell-cycle regulated genes, including BRCA1, by increasing the cytoplasmic retention of E2F4 transcriptional repressor and disrupting the binding of DREAM complex to DNA. Consistent with this model, inhibition of DYRK1A kinase required for the DREAM assembly by harmine, is shown to rescue the BRCA1 expression in the OFD1-depleted cells. However, the evidence to support the interaction between OFD1 and E2F4 is not strong and mostly relies on overexpression. The binding between OFD1 and E2F4 was not detected in multiple proteomic studies referenced in the BioGrid database, therefore it is unlikely that OFD1 is a major interacting protein of E2F4. Indeed, the mechanism of the nuclear import of E2F4 is well studied and involves the interaction with p130, which is inhibited by CDK phosphorylation. The OFD1-loss phenotypes observed in this study appear to be consistent with inhibition of CDK activity, but this possibility was not considered by the authors.

In summary, this manuscript reports a compelling story in support of a novel, potentially significant role of OFD1 in the tumor response to PARPi drugs that could be of interest to cancer researchers and clinicians. However, the proposed mechanistic model is not strongly supported by unconvincing binding data, and some well-known alternative mechanisms are not considered. It is also unclear how OFD1 can be targeted for clinical translation although the authors suggest that it "holds tremendous potential". Therefore, overall impact of this study appears to be moderate.

General comments – the manuscript requires careful editing and proofreading as there are multiple typos and errors – too many to list, as well as some unexplained abbreviations, discrepancies between the figures and text, and some confusing sentences. Some figures contain very small text labels that are unreadable, and typos.

Some specific comments on the figures:

Figures 3f-h: Rescue of BRCA1 expression in the OFD1 KO cell line by inducible OFD1 is not convincing and does not appear to be dose-dependent.

Figures 3i-j: These experiments do not establish causative relationship between OFD1 expression changes and BRCA1, only the fact that both proteins respond to hypoxia and serum starvation. Other treatments appear to have minimal effect on these proteins. Results of the CCLE and TCGA analyses in Fig. 3k-l suggest that perhaps some cell lines and cancer types show a modest correlation between these two genes, but do not strongly support a conclusion that OFD1 can influence

BRCA1 expression.

Figure 5j: Description in the text does not match the labeling in the figure (LIN54 knockdown instead of LIN37).

Figure 5k: This is not a rigorous format to represent immunoprecipitation-western blot data. The control IgG lines should be presented on the same panel as specific IP samples. Also, increased binding between RBBP4 and E2F4 is likely due to upregulation of RBBP4 in the shOFD1 input samples.

Figure 6d: The differences in E2F4 distribution are not obvious and should be quantified.

Figure 6f: Although the text refers to this experiment as interaction assay between the endogenous proteins, the Methods section describes that overexpressed GFP-OFD1 was used in the PLA assay.

Figure 6h: The lanes are mislabeled in the lower panel; 6k-l: regulatory domain is inconsistently marked as 181-336 or 182-336.

Figure 8a-b: There is no description of the tumor samples or scoring system in the Methods.

Reviewer #2

(Remarks to the Author)

The manuscript by Peng Li et al. aims to elucidate the role of OFD1 in the regulation of BRCA1 through a transcriptional mechanism involving the DREAM complex, particularly by binding to E2F4 to regulate its nuclear translocation. The authors provide substantial evidence indicating that the combination treatment of OFD1 deletion/knockdown and PARP inhibitor synergistically inhibits tumor growth, demonstrated through various study models, from in vitro cancer cells to PDX murine models. Their work suggests that reducing BRCA1 levels by depleting OFD1 presents a promising therapeutic potential for treating pancreatic cancers, many of which have a low risk of BRCA mutations or deficiency in cancer patients. The study is of significant interest but requires a considerable number of additional experiments for publication.

Major Concerns:

- 1) The western blot of BRCA1 should be shown for the cells tested in Figure 2 to see if the effect is universal, especially those sensitive to the combination of OFD1 depletion and Olaparib. The cell lines in the subsequent figures are PANC1 and MiaPaca2, which show sensitivity to OFD1 depletion to different extents. Why not use SW1990, PATU8988T, or PANC1005 in the RNA-seq? Particularly for the DNA repair assay, why did the authors use PANC1-DRGFP cells, which are the most sensitive to OFD1 depletion (no proliferation at all)?
- 2) The authors should perform rescue experiments in OFD1-depleted cells by overexpressing OFD1 and BRCA1, especially the latter, to establish the direct link and significance of OFD1-BRCA1 regulation in DNA damage response and repair efficiency.
- 3) The authors state that "BRCA1 was one of the most significantly altered homology-directed repair (HDR) genes in response to OFD1 knockdown." What about BRCA2 and RAD51, as they are also targets of E2F4? The authors should include RNA-seq data in a supplementary table.
- 4) In Figure 3f, despite overexpression levels of OFD1 being over 10 times more than the endogenous OFD1, the restoration of BRCA1 levels is very limited (the quantification in 3g and 3h is not consistent with the blot in 3f), suggesting that BRCA1 might not be the major target. The BRCA1 blot in this figure shows too much degradation and is not convincing. Additionally, BARD1 blot or BRCA1 foci could be included as indicators of BRCA1 restoration.
- 5) The expression of BRCA1 is regulated by the cell cycle (increasing in S/G2M), while it has been reported that OFD1 depletion causes cell cycle arrest before the S phase and leads to robust inhibition of cellular proliferation. Is it possible that the reduced BRCA1 expression in OFD1-depleted cells is indirectly due to cell cycle arrest?

Specific Comments:

- 1) Cell cycle analysis of OFD1 depletion should be provided with and without DNA damage.
- 2) In Figure 3c and 3d, the labeling of the graph is too small to read. The normalized enrichment score (NES) and false discovery rate (FDR) should be provided alongside each GSEA plot.
- 3) Figure 4b: The percentage in the reporter assay is too low to be reliable (2-3-fold less than the reference paper). The knockdown efficacy in the western blot should be shown. Is the increase in NHEJ with siBRCA1 significant? Why does siOFD1 not have the same phenotype, considering siBRCA1 and siOFD1 have similar HR defects?
- 4) Figure 4d: There is no difference in γ H2AX foci between shCtrl and shOFD1, but Figure 2i shows more DNA damage (γ H2AX and comet tail) in shOFD1 than shCtrl. Is this due to different knockdown efficiencies? A western blot of OFD1 and γ H2AX is needed. Or is it due to different cells used? The basal percentage of cells with γ H2AX foci is vastly different between the two experiments; why?
- 5) Figure 4i: The scale bar length should be provided.
- 6) LIN54 and LIN52 were tested, but the figure mentions LIN37 and LIN52 in Figure 5j.
- 7) Figure 6k: The domain label is 181-336, but in Figure 6i, it is 182-336.
- 8) Figure 7a: Have the authors performed western blotting to show that doxycycline treatment reliably knocks down OFD1 expression in the orthotopic model?
- 9) Figure 7c: The authors should provide statistical significance values for the comparison of mice groups at day 77 and apply this to all figures that present data in this manner.
- 10) Figure 7d: Expressing the pancreatic mass to body mass ratio may be skewed by cachexia in mice with large tumor

burdens. The authors should show that these mice have non-significant differences in body mass throughout the experiment. Additionally, if this is the case, they should quantify whether pancreatic mass/body mass is statistically different among groups, particularly comparing shOFD1 vs shOFD1+OLA.

11) Figure 7g: The rows in the table should be aligned for clarity.

12) Figures 7h and 7j: Indicate the number of mice included in the analysis for each treatment group in the figure legends.

13) Figure 8f: The OFD1 and BRCA1 blots are inconsistent with other figures. The authors should repeat and provide convincing data.

Errors and Typos:

- Correct errors such as "scar bar" (line 1063) and "Immunopredicipation" (line 1037).

Reviewer #3

(Remarks to the Author)

Li et al report the functional interaction between OFD1 and PARP inhibition in pancreatic cancer. They report BRCA1 as a critical gene down-regulated upon OFD1 knockdown and propose that the DREAM complex is involved in this effect. They subsequently go on to show that genetic OFD1 inhibition cooperates with Olaparib to reduce tumor growth in mice, both using human PDAC xenografts and mouse KPC cells in an immunocompetent model. The work is generally well performed and is novel, although I have some issues with their conclusions, as per the comments below.

Major comments

1. Figure 1 should show the data from multiple datasets available for PDAC rather than a selected one. Consistency of findings in the datasets is of utmost important (unless there is an explanation for discrepancies).
2. In several instances, antibody specificity should be demonstrated. This is the case particularly for antibodies detecting OFD1, DREAM complex members LIN52 and LIN54 (which are highly conserved proteins and most antibodies are inadequate).
3. In line 1234: the authors should provide information on whether the effects are associated with the genetic features of the cell lines used.
4. Are the effects of Olaparib also observable (even if at lower doses) in the lines that are sensitive to OFD1 knockdown?
5. There should be a genetic validation of the synthetic lethal effect between OFD1 knockdown and PARP inhibition. This applies to both in vitro and in vivo studies.
6. Can the authors comment on the potential relevance of BRCA2 and/or PALB2? BRCA is notoriously less important in PDAC than the latter two genes.
7. Figure 6 does not provide convincing evidence that the effects are mediated by regulation of E2F4 nuclear translocation.
8. Co-IP of endogenous proteins is important. The authors should provide this evidence.
9. Rescue experiments are required in some of the in vivo experiments.

Minor comments

1. There are multiple typos and grammar errors in the text. English editing should be applied.
2. Gene names should be properly applied (i.e. use of capital letters and/or italics).
3. There are multiple overstatements in the text (e.g., line 116 the word "critical" is used when the effects are modest). The authors should be critical in their statement.

Reviewer #4

(Remarks to the Author)

Version 1:

Reviewer comments:

Reviewer #1

(Remarks to the Author)

In the revised version, the authors made considerable improvements to the manuscript and addressed most of this reviewers' concerns. However, the proposed mechanism of the direct binding between the OFD1 and E2F4 is still not fully confirmed, although the authors now demonstrated the interaction between these proteins at the endogenous levels. The red (interaction) signal in the PLA assay shown in Figure 6f is not visible and does not match the graph, and the GFP-OFD1 signal is not visible in the siE2F4 panel. The experiment showing peptide competition in Figure 6k is not convincing and shows variable levels of many bands, including an unexplained dramatic increase of the E2F4-OFD1 interaction in the presence of peptide #1. The assay shown in support of the direct binding between the two proteins is not adequate because the recombinant proteins were purified from the mammalian cell line and the presence of additional factors can't be ruled out (in fact, several additional bands are clearly visible on the gels shown in Extended data Figure 6f,g). The binding

competition assay shown in Figure 6i is also not convincing because of the apparently unreasonably high concentration of the peptide #6 necessary to slightly attenuate the binding (although some details were missing in the Methods section to fully evaluate this experiment). Such weak activity of this peptide in the in vitro competition assays does not align with more robust effects observed in the cell-based assays. Therefore, it is recommended that the conclusion of the direct binding mechanism between E4F4 and OFD1 is downplayed and the panels 6f,g, and 6i-p are either removed, or shown as Extended data with appropriate comment of their limitations. Finally, there are still many typos in the text (although significantly improved).

Reviewer #2

(Remarks to the Author)

I appreciate that the authors have addressed most of the comments, and I find their responses acceptable overall. However, I still have some concerns.

On rebuttal page 20, despite the significantly reduced BRCA1 levels across the three pancreatic cancer cell lines, the BARD1 levels remain unchanged. This raises questions about the specificity of antibodies used in the experiments.

BRCA1 reduction upon OFD1 depletion in various experiment are very different.

Additionally, some p-values are still missing.

In summary, while most of their responses are satisfactory, the remaining issues regarding data consistency and missing p-values should be clarified.

Reviewer #4

(Remarks to the Author)

Reviewer #5

(Remarks to the Author)

This study identifies OFD1 as a regulator of BRCA1 expression through E2F4 and the DREAM complex. Mechanistically, the authors demonstrate that OFD1 physically interacts with E2F4, sequestering it in the cytoplasm. OFD1 depletion releases E2F4, allowing it to form the DREAM complex on the BRCA1 promoter, thereby reducing BRCA1 expression. This work provides a novel link between ciliopathy-related protein function (OFD1) and DNA repair regulation, an underexplored intersection of ciliopathy and tumor biology that enhances the study's significance.

The authors present extensive in vitro validation using multiple pancreatic cancer cell lines, synergy assays with PARP inhibitors, and mechanistic analyses, including ChIP, PLA, and rescue experiments. In vivo, they employ xenografts, orthotopic tumor models, patient-derived xenografts (PDXs), and an immunocompetent KPC transgenic model. The inclusion of both PDX and KPC models strengthens the translational relevance of their findings.

The authors have provided detailed, point-by-point responses to Reviewer 3's comments, addressing concerns with additional data, including expanded GEO analyses, endogenous co-immunoprecipitations, and both in vitro and in vivo rescue experiments. Specifically, they have:

1. Supplemented Figure 1 with multiple PDAC datasets to validate OFD1 overexpression.
2. Provided antibody specificity data for key proteins.
3. Clarified cell line selection and extended RNA-seq and functional analyses across multiple pancreatic cancer cell lines.
4. Demonstrated synergy between low-dose Olaparib and OFD1 depletion in sensitive cells.
4. Genetically validated the synthetic lethal effect of combined OFD1 and PARP1 knockdown.
5. Addressed the potential role of BRCA2/PALB2, showing BRCA1 as the most consistently affected target.
6. Confirmed that E2F4 nuclear translocation mediates BRCA1 downregulation.
7. Supplied endogenous co-IP data supporting the OFD1–E2F4 interaction.
8. Included additional rescue experiments in in vivo models to further validate their conclusions.

Overall, the authors have satisfactorily addressed Reviewer 3's concerns, significantly strengthening both the mechanistic insights and translational impact of their work. The revised manuscript is substantially improved and appears suitable for publication, pending final editorial review.

While the text is generally clear, a final editorial sweep for style and grammar could further enhance clarity.

Version 2:

Reviewer comments:

Reviewer #1

(Remarks to the Author)

The authors have addressed all the concerns and provided additional data to support their conclusions.

Reviewer #2

(Remarks to the Author)

The authors have satisfactorily addressed the majority of my concerns. The revised manuscript is significantly improved and clarifies key points previously raised. I find the current version acceptable and support its publication.

Reviewer #5

(Remarks to the Author)

The revised manuscript is suitable for publication in the journal Nature Communications.

Reviewer's comments:

Reviewer #1 - DREAM complex (Remarks to the Author):

OFD1 gene is associated with X-chromosome linked orofacial developmental syndrome and is functionally essential for ciliogenesis. Additionally, less studied roles of OFD1 in cell cycle and DNA repair have been described by several groups, including the contributing authors of this study. The manuscript proposes a novel mechanism of regulation of BRCA1 expression of OFD1 that could have implications for the DSB DNA repair and cancer cell sensitivity to PARP inhibitor drugs. The main finding of this study is that depletion of OFD1 in pancreatic cancer cell lines significantly increases sensitivity to PARPi both in vitro and in vivo, resulting in improved survival of the tumor bearing animals. This effect was consistently observed in tumor xenograft models, mouse KPC model of pancreatic cancer, and PDX models, suggesting that the status of OFD1 could have clinical significance for predicting the PARPi response. The authors further propose a mechanism where OFD1 regulates expression of BRCA1 gene at the transcript level and provide convincing evidence of this relationship. The effect of OFD1 depletion on DNA repair is consistent with previously observed defect in the DNA repair in the cells obtained from individuals with OFD I syndrome (PMID: 27798113). However, the previous study proposed a different mechanism for this phenotype focused on the interaction between OFD1 and TIP-60 histone acetyltransferase complex, which was not explored here. Instead, the authors propose that OFD1 promotes the expression of cell-cycle regulated genes, including BRCA1, by increasing the cytoplasmic retention of E2F4 transcriptional repressor and disrupting the binding of DREAM complex to DNA. Consistent with this model, inhibition of DYRK1A kinase required for the DREAM assembly by harmine, is shown to rescue the BRCA1 expression in the OFD1-depleted cells. However, the evidence to support the interaction between OFD1 and E2F4 is not strong and mostly relies on overexpression. The binding between OFD1 and E2F4 was not detected in multiple proteomic studies referenced in the BioGrid database, therefore it is unlikely that OFD1 is a major interacting protein of E2F4. Indeed, the mechanism of the nuclear import of E2F4 is well studied and involves the interaction with p130, which is inhibited by CDK phosphorylation. The OFD1-loss phenotypes observed in this study appear to be consistent with inhibition of CDK activity, but this possibility was not considered by the authors.

In summary, this manuscript reports a compelling story in support of a novel, potentially significant role of OFD1 in the tumor response to PARPi drugs that could be of interest to cancer researchers and clinicians. However, the proposed mechanistic model is not strongly supported by unconvincing binding data, and some well-known alternative mechanisms are not considered. It is also unclear how OFD1 can be targeted for clinical translation although the authors suggest that it "holds tremendous potential". Therefore, overall impact of this study appears to be moderate.

General comments – the manuscript requires careful editing and proofreading as there are multiple typos and errors – too many to list, as well as some unexplained abbreviations, discrepancies between the figures and text, and some confusing sentences. Some figures contain very small text labels that are unreadable, and typos.

Some specific comments on the figures:

Figures 3f-h: Rescue of BRCA1 expression in the OFD1 KO cell line by inducible OFD1 is not

convincing and does not appear to be dose-dependent.

Figures 3i-j: These experiments do not establish causative relationship between OFD1 expression changes and BRCA1, only the fact that both proteins respond to hypoxia and serum starvation. Other treatments appear to have minimal effect on these proteins. Results of the CCLE and TCGA analyses in Fig. 3k-l suggest that perhaps some cell lines and cancer types show a modest correlation between these two genes, but do not strongly support a conclusion that OFD1 can influence BRCA1 expression.

Figure 5j: Description in the text does not match the labeling in the figure (LIN54 knockdown instead of LIN37).

Figure 5k: This is not a rigorous format to represent immunoprecipitation-western blot data. The control IgG lines should be presented on the same panel as specific IP samples. Also, increased binding between RBBP4 and E2F4 is likely due to upregulation of RBBP4 in the shOFD1 input samples.

Figure 6d: The differences in E2F4 distribution are not obvious and should be quantified.

Figure 6f: Although the text refers to this experiment as interaction assay between the endogenous proteins, the Methods section describes that overexpressed GFP-OFD1 was used in the PLA assay.

Figure 6h: The lanes are mislabeled in the lower panel; 6k-l: regulatory domain is inconsistently marked as 181-336 or 182-336.

Figure 8a-b: There is no description of the tumor samples or scoring system in the Methods.

We sincerely appreciate your thorough review and detailed comments on our manuscript. As suggested, we have provided a point-by-point response to Reviewer #1's comments below.

Point-by-point response to Reviewer#1:

Major points:

1. The effect of OFD1 depletion on DNA repair is consistent with previously observed defect in the DNA repair in the cells obtained from individuals with OFD I syndrome (PMID: 27798113). However, the previous study proposed a different mechanism for this phenotype focused on the interaction between OFD1 and TIP-60 histone acetyltransferase complex, which was not explored here.

Response: Thank you for raising a good point. Indeed, we propose a different mechanism from the previous report about the function of OFD1 in DNA repair (PMID: 27798113). The published report (PMID: 27798113) suggested that OFD1 localizes to chromatin to interact with TIP60, through which OFD1 controls DNA double-strand break repair. In contrast, we propose that OFD1 regulates DNA repair through transcriptional regulation, by interacting with E2F4 at centriolar

satellites, a component of the DREAM transcriptional repressor that is known to be crucial for the expression of DNA repair genes (PMID: 31092693; PMID: 34477552). It is likely that both mechanisms could contribute to the DNA repair regulation in a cell type specific manner. Previous studies have shown that OFD1 localizes to nuclei in OFD Type I patient cells (PMID: 17761535; PMID: 27798113), which is rarely found in pancreatic cancer cells in our study. Nevertheless, we investigated whether OFD1 interacts with TIP60 in pancreatic cancer cells, as the reviewer suggested. In pancreatic cancer cell line MIA PaCa-2, OFD1 mainly localizes at centriolar satellites and had no detectable overlap with nuclear localized TIP60 (**Response1 Fig. 1a**). Neither OFD1 depletion by siRNA nor its overexpression affects the nuclear/cytoplasmic distribution (**Response1 Fig. 1a, b**) and abundance of TIP60 (**Response1 Fig. 1c, d**). We examined the interaction between OFD1 and TIP60 in pancreatic cancer cells using endogenous immunoprecipitation assays, OFD1 co-immunoprecipitated with E2F4, but not with TIP60 (**Response1 Fig. 1e**). **These results suggest that OFD1-TIP60 interaction is likely cell type-specific and largely absent in pancreatic cancer cells.**

Response1 Figure 1

Response1 Figure 1. Interplay between OFD1 and TIP60 in pancreatic cancer cells.

a, Immunofluorescence showing the effect of OFD1 knockdown and overexpression on the nuclear and cytoplasmic localization of TIP60 in the pancreatic cancer cell line MIA PaCa-2. Scale bar = 10 μ m. **b**, Statistical analysis of the nuclear and cytoplasmic localization of TIP60 in MIA PaCa-2 TetON-GFP-OFD1 cells transfected with siNC (control), siOFD1, or inducibly overexpressing GFP-OFD1 upon doxycycline treatment. Unpaired t-test. **c**, Western blot analysis examining the effects of si-OFD1 on TIP60 expression in MIA PaCa-2 cells. **d**, Western blot analysis of OFD1 and TIP60 expression in MIA PaCa-2 cells following doxycycline-induced OFD1 overexpression. **e**, Endogenous immunoprecipitation showing the interaction between OFD1 and E2F4, but not TIP60, in SW1990 pancreatic cancer cells. IgG was used as a negative control.

2: The evidence to support the interaction between OFD1 and E2F4 is not strong and mostly relies on overexpression. The binding between OFD1 and E2F4 was not detected in multiple proteomic studies referenced in the BioGrid database, therefore it is unlikely that OFD1 is a major interacting protein of E2F4.

Response: The BioGRID interaction database is an open-access repository of biological interactions, manually curated from published literature (PMID: 30476227). According to the BioGRID database, the primary binding partners of E2F4 are predominantly components of the DREAM complex, including RBL1, RBL2, LIN9, LIN37, LIN54, and others, identified under a stringent threshold of the top 2% of total binding proteins. These interactions have been well-documented in numerous studies and we have also validated these core interactions. Many important protein complexes are finetuned by regulatory proteins that control their assembly, localization and degradation, and we believe OFD1 is one such regulatory protein for E2F4 localization. We found that, by multiple means, E2F4 interacts with OFD1 in cytoplasm and the disruption of this interaction enables the translocation of E2F4 from cytoplasm to nuclei. As the reviewer pointed out, it is very important to present evidence for endogenous protein interaction. In the revised manuscript, we provided new data to show the endogenous immunoprecipitation between OFD1 and E2F4 (**Response1 Fig. 1e, Fig. 2a; Manuscript-Fig. 6h, Page 9, lines 257-260**). Furthermore, we provided *in vitro* pull-down assay to demonstrate a direct binding between recombinant OFD1 and E2F4 protein (**Response1 Fig. 2b-c; Manuscript-Extended Data Fig. 6f-6h**). Also, we performed the PLA (Proximity Ligation Assay) assay to demonstrate the co-localization of OFD1 and E2F4 (**Response1 Fig. 2d-f; Manuscript-Fig. 6e-6g, Page 9, lines 253-257**). Finally, we performed domain mapping assay to elucidate the precise binding domain between OFD1 and E2F4 (**Response1 Fig. 2g-h; Manuscript-Extended Data Fig. 6i-6j**), which was followed by *in vitro* peptide competition screening to identify the specific OFD1 binding motif of E2F4 (**Response1 Fig. 2i-j; Manuscript-Fig. 6k-6l, Page 10, lines 267-270**). The peptide containing the binding motif was sufficient to disrupt the OFD1-E2F4 interaction, to promote the nuclear translocation of E2F4 and to down-regulate BRCA1 expression (**Response1 Fig. 2j-l; Manuscript-Fig. 6m-6o, Page 10, lines 271-280**). These evidences prove the interaction of OFD1-E2F4 and its role in regulating E2F4 subcellular localization.

Response1 Figure 2

Response1 Figure 2. OFD1 interacts with E2F4 to regulate E2F4 translocation from the cytoplasm to the nucleus.

a, Endogenous immunoprecipitation (IP) showing the interaction between OFD1 and E2F4 in pancreatic cancer cell lines including MIA PaCa-2, SW1990, and PATU898T. **b-c**, *In vitro* pulldown assay demonstrating direct binding between recombinant OFD1 and E2F4 proteins. **d-f**, Co-localization of OFD1 and E2F4 in PANC1 cells

visualized by proximity ligation assay (PLA), and E2F4 knock-down was performed as a negative control group, which was validated by western blot (**d**). Representative images of co-localization of OFD1 and E2F4 in PANC1 cells (**e**). PLA signal intensity in each cell was quantified by ImageJ and presented (**f**). Scale bar = 5 μm . **g**, Schematic diagram showing full-length and truncated E2F4 based on functional domains. **h**, Immunoprecipitation assay showing the E2F4 binding region (amino acids 182-336) with OFD1. **i**, *In vitro* peptide competition screening of the specific OFD1 binding motif on E2F4. E2F4 bind to OFD1 mainly via 283-302 amino acids. The detailed amino acid sequence of candidate peptides derived from E2F4 283-302 amino acids is shown, with the competitive peptide labeled in red. **j**, OFD1-E2F4 binding *in vitro* was disrupted by the competitive peptide (283-302 amino acids) at a concentration of 250 μM . **k**, Representative images showing E2F4 nuclear/cytoplasmic localization in MIA PaCa-2 cells treated with control peptide and peptide (283-302 aas) for 48 hours. **l**, Western blot analysis showing the effect of the competitive peptide (283-302 aas) on BRCA1 protein expression in MIA PaCa-2 and SW1990 pancreatic cancer cells. The numbers under the gel lanes represent the ratio of BRCA1 band intensity to α -tubulin band intensity, which were normalized relative to the line 1 sample.

3: Indeed, the mechanism of the nuclear import of E2F4 is well studied and involves the interaction with p130, which is inhibited by CDK phosphorylation. The OFD1-loss phenotypes observed in this study appear to be consistent with inhibition of CDK activity, but this possibility was not considered by the authors.

Response: We thank the reviewer for the insightful comment. It is well-established that CDK4 phosphorylates p130 at Ser672, thereby reducing its interaction with E2F4 (PMID: 30833638). In our study, we assessed the phosphorylation status of p130/RBL2 at Ser672 in three pancreatic cancer cell lines before and after OFD1 knockdown, we found that OFD1 knockdown did not significantly affect p130 S672 phosphorylation levels (**Response1 Fig. 3a**). To further exclude the effect of OFD1 on CDK4 kinase activity, we performed an IP-Kinase assay (CDK4 Assay Kit, BPS Bioscience, Catalog #79674) to evaluate the effect of OFD1 knockdown on CDK4 kinase activity. This assay measures ATP consumption following the co-incubation of CDK4 with its substrate peptide. We found that OFD1 knockdown did not significantly affect CDK4 kinase activity in pancreatic cancer cells. However, treatment with palbociclib, a selective CDK4/6 inhibitor, resulted in marked inhibition of CDK4 activity across all treatment groups (**Response1 Fig. 3b, c**). Besides, cell cycle analysis revealed no detectable G1/G0 phase arrest following OFD1 knockdown. In contrast, palbociclib treatment (0.5 μM , 48 hours) induced significant G1/G0 phase arrest (**Response1 Fig. 3d**). All these data indicate that OFD1 regulates E2F4 in a CDK-independent pathway.

Response1 Figure 3

Response1 Figure 3. OFD1 knockdown does not affect CDK4 activity or cell cycle progression in pancreatic cancer cells.

a, the phosphorylation of p130 (RBL2) at Ser672 and the total RBL2 protein levels were assessed by Western blotting (WB) following OFD1 knockdown in MIA PaCa2, SW1990, and PATU8988T cells. **b**, MIA PaCa-2 and MIA PaCa-2-shOFD1 cells were lysed using a protein lysis buffer. Part of the lysate was retained for quantifying protein input (whole lysate), while CDK4 protein was captured using anti-CDK4 antibodies from both MIA PaCa-2 and MIA PaCa-2-shOFD1 cell lysates. Western blotting (WB) was performed to evaluate CDK4 protein levels in both the whole lysate and the CDK4 IP fractions. GST-CDK4 (200 ng, 100 ng, 50 ng) from the CDK4 assay kit was included as a reference. **c**, Endogenous CDK4 from MIA PaCa-2 and MIA PaCa-2-shOFD1 cells exhibited similar kinase activity, as indicated by ATP consumption. The CDK4/6 inhibitor Palbociclib (PAL, 0.1 μM) was used as a positive control to inhibit CDK4 activity. GST-CDK4 (200 ng) was co-incubated with the CDK4 substrate peptide, and ATP consumption was measured to assess kinase activity. Similarly, CDK4 immunoprecipitated from MIA PaCa-2 and MIA PaCa-2-shOFD1 cells was co-incubated with the substrate peptide to quantify kinase activity. Four independent replicates were performed, and statistical analysis was conducted using an unpaired t-test. **d**, the impact of OFD1 knockdown and palbociclib (0.5 μM) treatment on cell cycle progression was evaluated in PATU8988T, SW1990, and MIA PaCa-2 cells. Flow cytometric analysis revealed alterations in cell cycle distribution following OFD1 knockdown or palbociclib treatment.

4: It is also unclear how OFD1 can be targeted for clinical translation although the authors suggest that it” holds tremendous potential”.

Response: We appreciate the question raised by the reviewer. It might be difficult to target OFD1 given the nature that it is a scaffolding protein. However, this study provides a unique angle for clinical translation. We found that both depletion of OFD1 by shRNA or delivery of a small peptide to disrupt the interaction between OFD1-E2F4 could be used for anti-tumor effect. In a patient-derived xenograft (PDX) pancreatic cancer models, we depleted OFD1 by shRNA through

the delivery of an adeno-associated virus (AAV), and observed the reduced sizes of tumors in the presence of shRNA against OFD1 (**Response1 Fig. 4a-4d; Manuscript-Fig. 8c-8j, Pages 12, lines 334-341**). In a newly added result, we showed that application of a small peptide disrupting the interaction between OFD1 and E2F4 can promote E2F4 nuclear import, reduce BRCA1 expression and induce a synthetic lethality with Olaparib (**Response1 Fig. 4e-h; Manuscript-Fig. 6k-6p, Pages 10, lines 267-280**). These two approaches hold potential for clinical translation. We have also toned down the statement by removing “tremendous” (**Manuscript, Page14, lines 402-404**).

Response1 Figure 4

Response Figure 4. Targeting OFD1 by AAV-delivered shRNA or peptide interfering OFD1-E2F4 interaction suppresses tumor cell growth.

a, the effect of Olaparib, shOFD1 (AAV-shOFD1 intratumoral injection), and their combination on tumor growth in a PDAC patient-derived xenograft (PDX-SLQ) model is shown. **b**, Representative images of tumors harvested from each treatment group in PDX-SLQ model. **c**, immunoblot analysis of OFD1 and BRCA1 in tumors harvested from the PDX-SLQ model treated with the indicated drugs or vehicle at the end of the experiment. β-Actin was used as a loading control. **d**, Tumor growth curve for each group from PDX-LRD model were analyzed by unpaired t-test. **e**, *In vitro* peptide competition screening of the specific OFD1 binding motif on E2F4. E2F4 binds to OFD1 mainly via 283-302 amino acids. The detailed amino acid sequences of candidate peptides derived from the E2F4 (283-302) amino acids are shown, with the competitive peptide labeled in red. **f**, Representative immunofluorescence images showing the nuclear translocation of E2F4 in MIA PaCa-2 cells treated with the peptide (283-302) and control peptide. Scale bar = 10 μm. **g**, Western blot results of BRCA1 in MIA PaCa-2 cells treated with peptide (283-302) and control peptide. The ratio of BRCA1 band intensity to α-tubulin band intensity, normalized to the sample in line 1, is shown. **h**, Colony-formation assay indicating the effect of peptide (283-302) combined with various concentrations of Olaparib (0, 1.25, 2.5, 5, 10, 20 μM) on MIA PaCa-2 cells.

5: General comments – the manuscript requires careful editing and proofreading as there are multiple typos and errors – too many to list, as well as some unexplained abbreviations, discrepancies between the figures and text, and some confusing sentences. Some figures contain very small text labels that are unreadable, and typos.

Response: We appreciate this critical comment and sincerely apologize for the typographical errors and inconsistencies present in the manuscript. We have thoroughly revised the manuscript to correct these issues, including addressing the unexplained abbreviations, discrepancies between figures and text, and improving the clarity of sentences. Additionally, we have ensured that all figure labels are legible and the text is readable. We believe that these revisions have significantly enhanced the manuscript's quality and readability. Thank you again for your valuable feedback.

Minor points:

Some specific comments on the figures:

6: Figures 3f-h: Rescue of BRCA1 expression in the OFD1 KO cell line by inducible OFD1 is not convincing and does not appear to be dose-dependent.

Response: We are very grateful for the reviewer's question. The quality of this Western Blot (WB) indeed needs improvement. We have repeated this experiment and provided a dose-dependent rescue result in our revised manuscripts (**Response1 Fig. 6; Manuscript-Fig. 3k, Page 6, lines 165-167**).

Response1 Figure 6

Response Figure 6. Exogenous expression of OFD1 rescues BRCA1 expression in a dose-dependent manner.

A doxycycline-inducible, OFD1 siRNA-resistant GFP-OFD1 stable cell line was established in MIA PaCa-2 cells. The cells were treated with various concentrations of doxycycline (0 ng/mL, 50 ng/mL, 100 ng/mL, 200 ng/mL) to induce OFD1 expression. Western blot analysis was performed to assess the expression levels of OFD1 and BRCA1. The numbers under the gel lanes represent the ratio of BRCA1 band intensity to α -tubulin band intensity, which were normalized relative to the sample in lane 1.

7: Figures 3i-j: These experiments do not establish causative relationship between OFD1 expression changes and BRCA1, only the fact that both proteins respond to hypoxia and serum starvation. Other treatments appear to have minimal effect on these proteins. Results of the CCLE and TCGA analyses in Fig. 3k-l suggest that perhaps some cell lines and cancer types show a

modest correlation between these two genes, but do not strongly support a conclusion that OFD1 can influence BRCA1 expression.

Response: We agree with the reviewer that data presented in original Figure 3i-j is correlative since we aim to present the trend of OFD1 and BRCA1 expression upon different treatment in these experiments. To avoid confusion, we have removed these data from the revised manuscript. As suggested by the reviewer and the other two reviewers, we have now provided substantial evidence to prove the causative relationship between OFD1 expression change and BRCA1 in OFD1 knockdown and rescue experiments (**Response1 Fig. 7a-7e; Manuscript-Fig. 3, Page 6, lines 155-170**). These data demonstrate a critical role of OFD1 in regulating BRCA1 expression.

We also agree with the reviewer that our bioinformatic analysis presents a relatively modest correlation in pan-cancers, therefore we have removed these results as well. Interestingly, we observed a significant positive correlation ($R = 0.7356$) between OFD1 and BRCA1 expression during treatment with DNA damage-inducing agents and some kinase inhibitors, this data is included in the revised manuscript (**Response1-Fig. 7f; Manuscript-Extended Data Fig. 3f**).

Response1 Figure 7

Response Figure 7. OFD1 regulates BRCA1 expression

a, Validation of BRCA1 downregulated by Western blot in control versus shOFD1 in MIA PaCa-2, SW1990, PATU8988T and PANC1005 pancreatic cancer cell lines. two independent shRNA targeting OFD1 were used. **b**, Western blot showing that the Tet-on inducible re-expressed GFP-OFD1 (resistant to siOFD1) rescued BRCA1 decrease caused by siOFD1 in MIA PaCa-2 cells. MIA PaCa-2-tet-on-GFP-OFD1 cells were transfected with siOFD1 and treated with various concentration doxycycline (0, 25, 50, 100, 200 ng/mL), OFD1 and BRCA1 protein levels were assessed by western blot. **c**, Western blot showing that the stable re-expression of Flag-OFD1 (resistant to siOFD1) rescued BRCA1 reduction caused by siOFD1 in MIA PaCa-2 cells. **d**, Western blot showing that the stable re-expression of GFP-OFD1 (resistant to siOFD1) rescued the BRCA1 decrease caused by siOFD1 in SW1990 cells. **e**, Western blot showing that the stable re-expressed GFP-OFD1 (resistant to siOFD1) rescued BRCA1 decrease caused by siOFD1 in MIA PaCa-2 cells. **f**, Correlation between OFD1 and BRCA1 mRNA expression in MIA PaCa-2 treated with DNA damage-inducing agents and some kinase inhibitors.

8: Figure 5j: Description in the text does not match the labeling in the figure (LIN54 knockdown instead of LIN37).

Response: Thanks for the reviewer's careful examination. This was our typo. We have now corrected this error in **Manuscript Page 8, lines 225-227**.

9: Figure 5k: This is not a rigorous format to represent immunoprecipitation-western blot data. The control IgG lines should be presented on the same panel as specific IP samples. Also, increased binding between RBBP4 and E2F4 is likely due to upregulation of RBBP4 in the shOFD1 input samples.

Response: We agree with the reviewer that the original presentation is not appropriate due to the unnecessary processing. We have now included our original immunoblots images with all the control IgG panels presented on the same panels as specific IP samples in this revised manuscript (**Response1 Fig. 9**). The up-regulation of RBBP4 is likely due to strong exposure since we didn't detect much difference of RBBP4 levels in short exposure of these blots (**Response1 Fig. 9, panel d; Manuscript-Fig. 5i-j, Page 8, lines 227-230**).

Response1 Figure 9

Response1 Figure 9. OFD1 knockdown promotes the DREAM complex assembly.

a, Endogenous immunoprecipitation (IP) assay showed that silencing OFD1 expression enhanced the interaction between the DREAM complex subunits E2F4, RBL2 and RBBP4 in MIA PaCa-2 cells. The CDK4/6 inhibitor palbociclib treatment was included as a positive control. **b-c**, Display of the original immunoblotting images for E2F4, RBL2 and RBBP4 with short and long exposure time, and the cropped pictures in the primary manuscript was framed with red rectangles. **e**, Endogenous IP assay revealed that silencing OFD1 expression strengthened the interaction between the DREAM complex subunits E2F4, RBL2 and RBBP4 in PANC1 cells. Palbociclib treatment was included as a positive control. **f-h**, Original immunoblotting images for E2F4, RBL2 and RBBP4 are displayed with short and long exposure times, and the cropped pictures in the primary manuscript were framed with red rectangles.

10: Figure 6d: The differences in E2F4 distribution are not obvious and should be quantified.

Response: The immunoblots were quantified as shown below (**Response1 Fig. 10; Manuscript-Fig. 6d**). We observed that cytosolic E2F4 was decreased and nuclear E2F4 was increased upon OFD1 depletion in two pancreatic cancer cell lines.

Response1 Figure 10

Response1 Figure 10. OFD1 knockdown promotes nuclear import of E2F4.

Quantification of E2F4 protein level in whole cell lysate, cytosol and nucleus before and after OFD1 knockdown reveal that the nuclear fraction of E2F4 got elevated in response to OFD1 silence in both PANC1 and MIA PaCa-2 cells. The numbers under the gel lanes represent the ratio of E2F4 band intensity to α-tubulin or Lamin B band intensity, Cytosolic E2F4 levels are normalized to α-tubulin while nuclear E2F4 levels are normalized to Lamin B. which were normalized relative to the sample in lane 1.

11: Figure 6f: Although the text refers to this experiment as interaction assay between the endogenous proteins, the Methods section describes that overexpressed GFP-OFD1 was used in the PLA assay.

Response: Sorry for wrong statement. We have now included a new data of endogenous IP as shown in **Response1 Fig. 11a (Manuscript-Fig. 6h)**. For the PLA assay, indeed we used overexpressed GFP-OFD1 (RNAi resistant form) since we can't find the appropriate antibodies (the PLA assay requires antibodies from different species). Nevertheless, we have ensured that ectopically expressed GFP-OFD1 colocalizes with endogenous OFD1, and the expression level of GFP-OFD1 was comparable to that of endogenous OFD1 to mimics endogenous OFD1 (**Response1 Fig. 11b, c**).

Response1 Figure 11

Response Figure 11. Ectopic expression of GFP-OFD1 and colocalization with Endogenous OFD1.

a, Endogenous immunoprecipitation (IP) was performed to confirm the interaction of OFD1 and E2F4 in SW1990, MIA PaCa-2, and PATU8988T pancreatic cancer cell lines. IgG was used as a negative control. **b**, Co-localization of GFP signal and endogenous OFD1 in cells overexpressing GFP-OFD1 using immunofluorescence staining with OFD1 antibody; **c**, Western blot (WB) analysis confirmed the induction of GFP-OFD1 overexpression in PANC1 cells following doxycycline treatment, along with OFD1 knockdown using siRNA to deplete endogenous OFD1.

12: Figure 6h: The lanes are mislabeled in the lower panel; 6k-l: regulatory domain in inconsistently marked as 181-336 or 182-336.

Response: We thank the reviewer for pointing out the error. This was a typographical mistake, and we have corrected the labeling in the revised manuscript.

13: Figure 8a-b: There is no description of the tumor samples or scoring system in the Methods.

Response: We thank the reviewer for pointing out the missing information. We have now provided the detailed clinical information from pancreatic cancer patient samples in the revised manuscript and include the IHC (Immunohistochemistry) scoring for OFD1 and BRCA1 in the Materials and Methods section, description in the manuscript (**Pages 21. Lines 609-620**) and **Table S4**.

Reviewer's comments:

Reviewer #2 - PARPi, BRCA (Remarks to the Author):

The manuscript by Peng Li et al. aims to elucidate the role of OFD1 in the regulation of BRCA1 through a transcriptional mechanism involving the DREAM complex, particularly by binding to E2F4 to regulate its nuclear translocation. The authors provide substantial evidence indicating that the combination treatment of OFD1 deletion/knockdown and PARP inhibitor synergistically inhibits tumor growth, demonstrated through various study models, from in vitro cancer cells to PDX murine models. Their work suggests that reducing BRCA1 levels by depleting OFD1 presents a promising therapeutic potential for treating pancreatic cancers, many of which have a low risk of BRCA mutations or deficiency in cancer patients. The study is of significant interest but requires a considerable number of additional experiments for publication.

Major Concerns:

- 1) The western blot of BRCA1 should be shown for the cells tested in Figure 2 to see if the effect is universal, especially those sensitive to the combination of OFD1 depletion and Olaparib. The cell lines in the subsequent figures are PANC1 and MiaPaca2, which show sensitivity to OFD1 depletion to different extents. Why not use SW1990, PATU8988T, or PANC1005 in the RNA-seq? Particularly for the DNA repair assay, why did the authors use PANC1-DRGFP cells, which are the most sensitive to OFD1 depletion (no proliferation at all)?
- 2) The authors should perform rescue experiments in OFD1-depleted cells by overexpressing OFD1 and BRCA1, especially the latter, to establish the direct link and significance of OFD1-BRCA1 regulation in DNA damage response and repair efficiency.
- 3) The authors state that "BRCA1 was one of the most significantly altered homology-directed repair (HDR) genes in response to OFD1 knockdown." What about BRCA2 and RAD51, as they are also targets of E2F4? The authors should include RNA-seq data in a supplementary table.
- 4) In Figure 3f, despite overexpression levels of OFD1 being over 10 times more than the endogenous OFD1, the restoration of BRCA1 levels is very limited (the quantification in 3g and 3h is not consistent with the blot in 3f), suggesting that BRCA1 might not be the major target. The BRCA1 blot in this figure shows too much degradation and is not convincing. Additionally, BARD1 blot or BRCA1 foci could be included as indicators of BRCA1 restoration.
- 5) The expression of BRCA1 is regulated by the cell cycle (increasing in S/G2M), while it has been reported that OFD1 depletion causes cell cycle arrest before the S phase and leads to robust inhibition of cellular proliferation. Is it possible that the reduced BRCA1 expression in OFD1-depleted cells is indirectly due to cell cycle arrest?

Specific Comments:

- 1) Cell cycle analysis of OFD1 depletion should be provided with and without DNA damage.
- 2) In Figure 3c and 3d, the labeling of the graph is too small to read. The normalized enrichment score (NES) and false discovery rate (FDR) should be provided alongside each GSEA plot.
- 3) Figure 4b: The percentage in the reporter assay is too low to be reliable (2-3-fold less than the reference paper). The knockdown efficacy in the western blot should be shown. Is the increase in NHEJ with siBRCA1 significant? Why does siOFD1 not have the same phenotype, considering siBRCA1 and siOFD1 have similar HR defects?
- 4) Figure 4d: There is no difference in γ H2AX foci between shCtrl and shOFD1, but Figure 2i shows more DNA damage (γ H2AX and comet tail) in shOFD1 than shCtrl. Is this due to different knockdown efficiencies? A western blot of OFD1 and γ H2AX is needed. Or is it due to different cells used? The basal percentage of cells with γ H2AX foci is vastly different between the two experiments; why?
- 5) Figure 4i: The scale bar length should be provided.
- 6) LIN54 and LIN52 were tested, but the figure mentions LIN37 and LIN52 in Figure 5j.
- 7) Figure 6k: The domain label is 181-336, but in Figure 6i, it is 182-336.
- 8) Figure 7a: Have the authors performed western blotting to show that doxycycline treatment reliably knocks down OFD1 expression in the orthotopic model?
- 9) Figure 7c: The authors should provide statistical significance values for the comparison of mice groups at day 77 and apply this to all figures that present data in this manner.
- 10) Figure 7d: Expressing the pancreatic mass to body mass ratio may be skewed by cachexia in

mice with large tumor burdens. The authors should show that these mice have non-significant differences in body mass throughout the experiment. Additionally, if this is the case, they should quantify whether pancreatic mass/body mass is statistically different among groups, particularly comparing shOFD1 vs shOFD1+OLA.

11) Figure 7g: The rows in the table should be aligned for clarity.

12) Figures 7h and 7j: Indicate the number of mice included in the analysis for each treatment group in the figure legends.

13) Figure 8f: The OFD1 and BRCA1 blots are inconsistent with other figures. The authors should repeat and provide convincing data.

Errors and Typos:

- Correct errors such as "scar bar" (line 1063) and "Immunopredicipation" (line 1037).

We sincerely appreciate your thorough review and detailed comments on our manuscript. As suggested, we have provided a point-by-point response to Reviewer #2's comments below.

Point-by-point response to Reviewer#2:

Major points:

1. The western blot of BRCA1 should be shown for the cells tested in Figure 2 to see if the effect is universal, especially those sensitive to the combination of OFD1 depletion and Olaparib. The cell lines in the subsequent figures are PANC1 and MIA PaCa2, which show sensitivity to OFD1 depletion to different extents. Why not use SW1990, PATU8988T, or PANC1005 in the RNA-seq? Particularly for the DNA repair assay, why did the authors use PANC1-DRGFP cells, which are the most sensitive to OFD1 depletion (no proliferation at all)?

Response: We thank the reviewer for the critical comments. As the reviewer suggested, we have added the WB results showing the expression of BRCA1 upon OFD1 knockdown in all cell lines tested in Figure 2 in the revised manuscript (**Response2 Fig. 1a; Manuscript-Fig. 3h-3j**). OFD1 knockdown led to the downregulation of BRCA1 in all tested cell lines. Also, as the reviewer suggested, we have provided RNA-seq analysis after OFD1 knockdown in PANC-1, MIA PaCa2 and PATU8988T cell lines, BRCA1 downregulation was found in all these analysis as marked (**Response2 Fig. 1b; Manuscript-Fig. 3a-3c**). Besides, we have provided DNA repair assay in MIA PaCa2 and rescue experiments (**Response2 Fig. 1c; Manuscript-Fig. 4a-4b**). All these data indicate that OFD1 regulates BRCA1 expression levels in different pancreatic cancer cells. Although the cell death response varies in different pancreatic cancer cell lines, even in the most sensitive PANC1 cells, we have observed a synergy between OFD1 depletion and PARPi treatment when lower doses were applied (**Response2 Figure 1-2**).

Response2 Figure 1

Response2 Figure 1. Downregulation of BRCA1 and impairment of Homology-Directed Repair (HDR) upon OFD1 knockdown. **a**, Western blot analysis of OFD1 and BRCA1 expression in MIA PaCa-2, SW1990, PATU8988T, and PANC1005 cells following OFD1 knockdown. Two distinct shRNA OFD1 knockdown sequences targeting OFD1 were used; **b**, Volcano plots from RNA seq data showing changes in BRCA1 and other differentially expressed genes in PANC1, MIA PaCa-2, and PATU8988T cells after OFD1 knockdown, with BRCA1 highlighted; **c**, Evaluation of the effect of OFD1 knockdown on homology-directed repair (HDR) capacity in MIA PaCa-2 cells using the Trafficking Light Reporter system, and BRCA1 siRNA was involved as a positive control of HR defect.

Response2 Figure 1-2

Response2 Figure 1-2. Sensitivity of OFD1 WT and knockdown PANC1 cells to Olaparib. **a**, PANC1 wild-type cells were treated with 0, 12.5, 25, 50, 100, 200, 300 μ M Olaparib for 96 hours. Cell viability was assessed using the CCK8 assay. **b**, PANC1 and PANC1-shOFD1 cells were treated with 0, 5 μ M, 10 μ M, or 25 μ M Olaparib for 72 hours. Cell viability was measured by the CCK8 assay, and statistical analysis was performed using two-way ANOVA. The initial cell density was 2,000 cells per well in a 96-well plate.

2. The authors should perform rescue experiments in OFD1-depleted cells by overexpressing OFD1 and BRCA1, especially the latter, to establish the direct link and significance of OFD1-BRCA1 regulation in DNA damage response and repair efficiency.

Response: We thank the reviewer for the insightful comments. As the reviewer suggested, we have conducted rescue experiments in OFD1-deficient cells by ectopically expressing OFD1 (**Response2 Fig. 2; Manuscript-Fig. 4**). This rescue experiment demonstrates that OFD1 is required for BRCA1 expression regulation, DNA damage response and the synergy between OFD1 and PARPi treatments (**Response2 Fig. 2; Manuscript-Fig. 4, Page 7-8, lines 182-205; Manuscript-Fig. 2e-2i, Page 6, lines 143-148**). We have encountered difficulties when we tried to express full-length BRCA1 in rescue experiments probably due to its large size, and several concerns had been raised about ectopically expression of full-length BRCA1, such as achieving proper BRCA1 cellular localization and the potential production of biologically toxic substances (PMID: 12833136, PMID: 9010228). Furthermore, the association between the downregulation of BRCA1 and impaired DNA homologous recombination (HR) repair efficiency has been extensively documented (PMID: 21195000, PMID: 22915752, PMID: 39389065). In this study, we aimed to emphasize the function of OFD1 in regulating the DREAM complex through its interaction with E2F4, and BRCA1 expression is one of the important readouts of this transcriptional regulation.

Response2 Figure 2

Response2 Figure 2. Re-expression of OFD1 in OFD1 depleted cells rescues the cellular response to DNA damage and DNA damage repair.

a-b, Evaluation of the impact of OFD1 knockdown on homology-directed repair (HDR) capacity in MIA PaCa-2 cells using the Trafficking Light Reporter system, and BRCA1 knockdown was included as a positive control of

HR defect; **c**, Impact of OFD1 knockdown on the metaphase chromosome spread in MIA PaCa-2 cell line. Chromosomal abnormalities were quantified for the indicated groups (n >20 metaphase spreads). Arrows indicate examples of chromosomes with missing arms or fusion chromosomes. Scale bar, 50 μ m. **d**, Representative confocal images showing γ H2AX and BRCA1 foci upon OFD1 knockdown and re-expression exogenous OFD1 in MIA PaCa-2 cells treated with no ionizing radiation (IR) or 10 Gy IR. Scale bar = 10 μ m. **e**, Quantification of chromosome abnormality chances in each MIA PaCa-2 cell; **f, g**, Quantification of γ H2AX (**f**) and BRCA1 (**g**) positive foci per cell in MIA PaCa-2, at least 30 cells were analyzed for each datum point. **h**, Representative images of the Comet assay for MIA PaCa-2, MIA PaCa-2-shOFD1, and re-expression exogenous OFD1 after treatment with 0 or 5 μ M Olaparib for 48 hours. Scale bar = 25 μ m **i**, Quantification of tail moment using the Image J (Open-Comet) software. Data are presented as mean \pm SD (n = 100, unpaired t test).

3. The authors state that "BRCA1 was one of the most significantly altered homology-directed repair (HDR) genes in response to OFD1 knockdown." What about BRCA2 and RAD51, as they are also targets of E2F4? The authors should include RNA-seq data in a supplementary table.

Response: We thank the reviewer for raising a good point. As the reviewer suggested, we have also checked the expression of BRCA2 and Rad51, as well as BARD1 and PALB2, in our experiments. Both BRCA2 and RAD51 were down-regulated following OFD1 knockdown in the pancreatic cancer cell line PANC1. However, in other pancreatic cancer cell lines, the down-regulation of these genes was less pronounced, while BRCA1 was consistently and significantly down-regulated across all the tested cell lines (**Response2 Fig. 3**). We have included the RNA-seq data in a supplementary table as requested (**Table S3**).

Response2 Figure 3

Response2 Figure 3. Effect of OFD1 knockdown on HR genes in pancreatic cancer cells.

a-c, western blot analyses were performed to assess the expression levels of homologous recombination (HR)-related proteins, including BRCA1, BARD1, BRCA2, PALB2, and RAD51, following OFD1 knockdown in three pancreatic cancer cell lines PANC1 (**a**), MIA-Paca2 (**b**), SW1990 (**c**).

d, Volcano plot of RNA sequencing results from PATU8988T cells following OFD1 knockdown, highlighting the expression of BRCA1, BRCA2, PALB2, and RAD51 on the plot.

4. In Figure 3f, despite overexpression levels of OFD1 being over 10 times more than the endogenous OFD1, the restoration of BRCA1 levels is very limited (the quantification in 3g and 3h is not consistent with the blot in 3f), suggesting that BRCA1 might not be the major target. The BRCA1 blot in this figure shows too much degradation and is not convincing. Additionally, BARD1 blot or BRCA1 foci could be included as indicators of BRCA1 restoration.

Response: We sincerely thank the reviewer for the constructive feedback. The limited restoration of BRCA1 observed in Figure 3f may be due to protein degradation caused by suboptimal sample processing. To address this issue, we have thoroughly optimized the experimental conditions and repeated the overexpression experiments (**Response2 Fig. 4a; Manuscript-Fig. 3k**). Notably, after overexpressing Flag-tagged OFD1 or GFP-tagged OFD1, we observed a significant restoration of BRCA1 levels (**Response2 Fig. 4b-4d; Manuscript-Fig. 3l-3m**), which was further supported by the reformation of BRCA1 foci under irradiation conditions, demonstrating the functional recovery of BRCA1 (**Response2 Fig. 4e-4h; Manuscript-Fig. 4e-4g**). These results provide compelling evidence that OFD1 overexpression plays a crucial role in restoring both the protein levels and the DNA repair function of BRCA1.

Response2 Figure 4

Response2 Figure 4. Overexpression of OFD1 restores BRCA1 expression and the formation of BRCA1 foci under irradiation (IR) conditions. **a**, Western blot showing that the Tet-on inducible system re-expressed GFP-OFD1 (resistant to siOFD1) rescued BRCA1 decrease caused by siOFD1 in MIA PaCa-2 cells. MIA PaCa-2-tet-on- GFP -OFD1 cells were transfected with siOFD1 and treated with different concentration doxycycline (0, 25, 50, 100, 200 ng/mL), OFD1 and BRCA1 protein levels were examined by western blot. **b**, Western blot showing that the stable re-expressed Flag-OFD1 (resistant to siOFD1) rescued BRCA1 decrease caused by siOFD1 in MIA PaCa-2 cells. **c**, Western blot showing that the stable re-expressed GFP-OFD1 (resistant to siOFD1) rescued BRCA1 decrease caused by siOFD1 in MIA PaCa-2 cells. **d**, Western blot showing that the stable re-expressed GFP-OFD1 (resistant to siOFD1) rescued BRCA1 decrease caused by siOFD1 in SW1990 cells. **e**, Representative confocal images showing γ H2AX and BRCA1 foci upon OFD1 knockdown and re-expression exogenous OFD1 in MIA PaCa-2 cells treated with no ionizing radiation (IR) or 10 Gy IR. Scale bar = 10 μ m. **f**, Quantification of chromosome abnormality chances in each MIA PaCa-2 cell; **g-h**. Quantification of γ H2AX (**g**) and BRCA1 (**h**) positive foci per cell in MIA PaCa-2, at least 30 cells were analyzed for each datum point.

5. The expression of BRCA1 is regulated by the cell cycle (increasing in S/G₂M), while it has been reported that OFD1 depletion causes cell cycle arrest before the S phase and leads to robust inhibition of cellular proliferation. Is it possible that the reduced BRCA1 expression in OFD1-depleted cells is indirectly due to cell cycle arrest?

Response: We thank the reviewer for the helpful comments. In our previously published study (PMID: 36973243, corresponding to Fig. 5c), we demonstrated that OFD1 knockdown induces cell cycle arrest and inhibits cellular proliferation in normal human cells (hTERT-BJ1, IMR-90) as assessed by Ki-67 staining. However, in pancreatic cancer cell lines (PANC1 and MIA PaCa-2), we observed that OFD1 depletion did not affect the proportion of Ki-67 positive cells, and cellular proliferation was not significantly impacted. Further analysis of cell cycle progression in a series of pancreatic cancer cell lines following OFD1 knockdown revealed no significant cell cycle arrest (**Response2 Fig. 5**). These results suggest that the reduced BRCA1 expression observed in OFD1-depleted pancreatic cancer cells is unlikely to be due to cell cycle arrest.

(Muqing Cao, et al. Nat Commun. 2023.Fig.5c)

Response2 Figure 5

Response Figure 5. Cell cycle analysis of OFD1 depletion combined with DNA Damage agents in pancreatic cancer cells. MIA PaCa-2, PATU 8988T, and SW1990 pancreatic cancer cells, along with their corresponding

OFD1 knockdown variants, were treated with 0 or 5 μ M cisplatin for 48 hours. Cell cycle distribution was analyzed by propidium iodide (PI) staining followed by fluorescence-activated cell sorting (FACS).

Specific Comments:

6.(1) Cell cycle analysis of OFD1 depletion should be provided with and without DNA damage.

Response: As the reviewer suggested, we have now included the cell cycle analysis results from flow cytometry following OFD1 knockdown in MIA PaCa-2, PATU 8988T, and SW1990 cells, with and without cisplatin treatment. The data indicate that OFD1 knockdown does not lead to obvious alterations in the cell cycle, either in the presence or absence of DNA damage induced by cisplatin (same as **Response2 Fig. 5**).

7.(2) In Figure 3c and 3d, the labeling of the graph is too small to read. The normalized enrichment score (NES) and false discovery rate (FDR) should be provided alongside each GSEA plot.

Response: As the reviewer suggested, we have revised the image as requested, by increasing the font size for better readability, and adding the normalized enrichment score (NES) and false discovery rate (FDR) alongside each GSEA plot (**Manuscript-Fig. 3d-3g**). Additionally, detailed descriptions have been provided in the text of the revised manuscript (**Page 6, lines 160-163**).

8.(3) Figure 4b: The percentage in the reporter assay is too low to be reliable (2-3-fold less than the reference paper). The knockdown efficacy in the western blot should be shown. Is the increase in NHEJ with siBRCA1 significant? Why does siOFD1 not have the same phenotype, considering siBRCA1 and siOFD1 have similar HR defects?

Response: We thank the reviewer for the comments. Regarding the low proportion of reporter events in Figure 4b, we attribute this issue to the insufficient amount of I-SceI donor plasmid transfected. To address this concern, we have optimized the experimental conditions and improved the transfection in MIA PaCa-2 cells and obtained more satisfactory results which is now similar to the reference paper (**Response2 Fig. 8a-8b**). Additionally, the knockdown efficacy of OFD1 in these experiments is now shown in the Western blot (**Response2 Fig. 8c, Manuscript-Extended Data Fig. 4b**). Furthermore, we would like to clarify that si-BRCA1 treatment did not enhance NHEJ activity statistically in the PANC1 cells in our previous experiments in the initial version of manuscript, since no significant difference ($p = 0.098$) was observed (**Response2 Fig. 8d**).

Response2 Figure 8

Response2 Figure 8. Impact of OFD1 Knockdown on Homologous Recombination (HR) Repair Efficiency using the HR-GFP Reporter System. **a**, Schematic diagram of the HR-GFP reporter system. **b**, Effect of OFD1 knockdown on HR repair efficiency. MIA PaCa-2 DR-GFP cells were transfected with OFD1 siRNA, siNC, or BRCA1 siRNA, along with OFD1 overexpression (OE) plasmids and I-SceI donor plasmids. The percentage of GFP-positive cells was assessed by fluorescence-activated cell sorting (FACS) 48 hours post-transfection. Si-BRCA1 was used as a positive control to inhibit HR repair. Data are presented as mean \pm SD ($n = 3$ independent experiments; unpaired t-test). **c**, Western blot analysis confirming the knockdown of OFD1 and BRCA1 siRNA and re-expression of OFD1. **d**, Effect of OFD1 knockdown on HR repair efficiency in PANC1-DR-GFP cells (unpaired t-test).

9.4: Figure 4d: There is no difference in γ H2AX foci between shCtrl and shOFD1, but Figure 2i shows more DNA damage (γ H2AX and comet tail) in shOFD1 than shCtrl. Is this due to different knockdown efficiencies? A western blot of OFD1 and γ H2AX is needed. Or is it due to different cells used? The basal percentage of cells with γ H2AX foci is vastly different between the two experiments; why?

Response: We thank the reviewer for the critical comments. We carefully checked whether “there is no difference in γ H2AX foci between shCtrl and shOFD1” as the reviewer mentioned, and we found that it is probably a false impression due to light exposure of shCtrl and shOFD1 images. After we increased the exposure time of these images, we found that OFD1 knockdown leads to an increase in the number of γ H2AX foci (**Response2 Fig. 9a-9c; Manuscript-Fig. 4e-4g**), which is similar to increased DNA damage (γ H2AX and comet tail) in shOFD1 than shCtrl as shown in original Figure 2i. Additionally, we have included γ H2AX Western blot (WB) analysis and quantification for the various treatment groups, and this result also showed that OFD1 knockdown leads to an increase in γ H2AX protein level, consistently with γ H2AX foci immunofluorescence

results (Response2 Fig. 9d; Manuscript-Extended Data Fig. 4i).

Response2 Figure 9

Response2 Figure 9. OFD1 is important for DNA damage response in MIA PaCa-2 Cells. **a**, Representative confocal images showing γ H2AX and BRCA1 foci upon OFD1 knockdown and re-expression exogenous OFD1 in MIA PaCa-2 cells treated with no ionizing radiation (IR) or 10 Gy IR. **b-c**, Quantification of γ H2AX (**b**) and BRCA1 (**c**) positive foci per cell in MIA PaCa-2, at least 30 cells were analyzed for each datum point. **d**, Representative Western blot results showing γ H2AX expression in OFD1 knockdown and re-expressing MIA PaCa-2 cells, with and without 10 Gy ionizing radiation (IR) treatment. The numbers under the gel lanes represent the ratio of γ H2AX band intensity to β -actin band intensity, which were normalized relative to the line 1 sample. The grayscale ratio of γ H2AX/ β -actin is quantified and presented below the blot.

10.(5) Figure 4i: The scale bar length should be provided.

Response: Thanks for the reminder. We have now provided this detail in the revised manuscript (Scale bar = 25 μ m) (Manuscript-Extended Data Fig. 4e), detailed descriptions have been provided in Manuscript (Page 54, lines 1350-1352).

11.(6) LIN54 and LIN52 were tested, but the figure mentions LIN37 and LIN52 in Figure 5j.

Response: Apology for the typo. We have now made the correction to dual knockdown of LIN37 and LIN52, detailed descriptions have been provided in Manuscript (Page 8, lines 225-227).

12. (7) Figure 6k: The domain label is 181-336, but in Figure 6i, it is 182-336.

Response: Corrected.

13. (8) Figure 7a: Have the authors performed western blotting to show that doxycycline treatment reliably knocks down OFD1 expression in the orthotopic model?

Response: As the reviewer suggested, we have now performed Western blot analysis to assess the protein levels of OFD1 and BRCA1 in the tissue samples from each treatment group (Response2 Fig. 13; Manuscript-Fig. 7f)

Response2 Figure 13

Response2 Figure 13. Efficiency of doxycycline-induced OFD1 knockdown on protein expression in MIA PaCa-2 pancreatic tumors. Western blot analysis was performed to assess the protein expression levels of OFD1 and BRCA1 in MIA PaCa-2 pancreatic tumors following doxycycline-induced OFD1 knockdown (N = 3).

14. (9): Figure 7c: The authors should provide statistical significance values for the comparison of mice groups at day 77 and apply this to all figures that present data in this manner.

Response: Good point. In the revised manuscript, we have conducted two-way ANOVA statistical analysis to compare the different treatment groups (Response2 Fig. 14; Manuscript-Fig. 7c).

Response2 Figure 14

Response2 Figure 14. Assessment of MIA PaCa-2 pancreatic orthotopic tumor growth via bioluminescence imaging. The growth curve of MIA Paca2 pancreatic orthotopic tumors was monitored using bioluminescence signals. Bioluminescence intensity in each experimental group was quantified (N = 6), and the signal intensity on Day 78 was statistically analyzed using two-way ANOVA.

15.(10) Figure 7d: Expressing the pancreatic mass to body mass ratio may be skewed by cachexia in mice with large tumor burdens. The authors should show that these mice have non-significant differences in body mass throughout the experiment. Additionally, if this is the case, they should quantify whether pancreatic mass/body mass is statistically different among groups, particularly comparing shOFD1 vs shOFD1+OLA.

Response: We thank the reviewer for the helpful comments. Cancer cachexia typically results in the loss of skeletal muscle mass and a decrease in overall body mass (PMID: 21296615). We have provided data on the body weight of mice throughout the experiments, which show no significant differences between the groups (**Response2 Fig. 15a**). Additionally, data on the pancreatic mass/body weight ratios across the different groups, along with the corresponding statistical analysis, also strongly support our conclusion (**Response2 Fig. 15b; Manuscript-Fig. 7d**).

Response2 Figure 15

Response2 Figure 15. The body weight and pancreas mass ratio in MIA PaCa-2 orthotopic tumor-bearing mice. a, Body weight growth curve of MIA PaCa-2-tet-on-shOFD1 tumor-bearing mice from day 0 to day 78. **b,** Pancreas mass/body mass ratio in five treatment groups: Vehicle, shOFD1, Olaparib, shOFD1 + Olaparib, and Healthy mice. unpaired t-test

16.(11) Figure 7g: The rows in the table should be aligned for clarity.

Response: We thank the reviewer for the helpful comments, we have improved the layout of this table to meet the requirement (**Manuscript-Fig. 7g**).

17.(12) Figures 7h and 7j: Indicate the number of mice included in the analysis for each treatment group in the figure legends.

Response: We have now included the number of mice analyzed for each treatment group in the figure legends of the revised manuscript as follows:

Manuscript-Fig. 7h, MIA PaCa-2 tumor-bearing mice were treated with vehicle (**n = 6**), shOFD1 (**n = 6**), Olaparib (**n = 6**), and shOFD1+Olaparib (**n = 6**) for up to 365 days (1 year). The overall survival was calculated using Kaplan-Meier survival analysis for the entire duration of 365 days.

Manuscript-Fig. 7j, Kaplan-Meier analysis comparing survival of KPC (**n = 23**), KPCO (**n = 19**), KPC+Olaparib (**n = 14**), KPCO+Olaparib (**n = 11**), CO (**n = 6**). Mantel-Cox test.

18.(13) Figure 8f: The OFD1 and BRCA1 blots are inconsistent with other figures. The authors should repeat and provide convincing data.

Response: As the reviewer suggested, we have repeated and provided convincing data about OFD1 and BRCA1 blots in the revised manuscript (**Response2 Fig. 18; Manuscript-Fig. 8f**).

Response2 Figure 18

Response2 Figure 18. Genetic validation of OFD1 knockdown by AAV and downregulated BRCA1 protein levels in PDX.

Immunoblot analysis of OFD1 and BRCA1 protein levels in tumors harvested from PDX models treated with either Olaparib or vehicle at the end of the experiment, with β -Actin as a loading control.

19.(14)) : Errors and Typos: Correct errors such as "scar bar" (line 1063) and "Immunopredicipation" (line 1037).

Response: Corrected.

Reviewer's comments:

Reviewer #3 - Pancreatic cancer (Remarks to the Author):

Li et al report the functional interaction between OFD1 and PARP inhibition in pancreatic cancer. They report BRCA1 as a critical gene down-regulated upon OFD1 knockdown and propose that the DREAM complex is involved in this effect. They subsequently go on to show that genetic OFD1 inhibition cooperates with Olaparib to reduce tumor growth in mice, both using human PDAC xenografts and mouse KPC cells in an immunocompetent model. The work is generally well performed and is novel, although I have some issues with their conclusions, as per the comments below.

Major comments

1. Figure 1 should show the data from multiple datasets available for PDAC rather than a selected one. Consistency of findings in the datasets is of utmost important (unless there is an explanation for discrepancies).
2. In several instances, antibody specificity should be demonstrated. This is the case particularly for antibodies detecting OFD1, DREAM complex members LIN52 and LIN54 (which are highly conserved proteins and most antibodies are inadequate).
3. In line 1234: the authors should provide information on whether the effects are associated with the genetic features of the cell lines used.
4. Are the effects of Olaparib also observable (even if at lower doses) in the lines that are sensitive to OFD1 knockdown?
5. There should be a genetic validation of the synthetic lethal effect between OFD1 knockdown and PARP inhibition. This applies to both in vitro and in vivo studies.
6. Can the authors comment on the potential relevance of BRCA2 and/or PALB2? BRCA is notoriously less important in PDAC than the latter two genes.
7. Figure 6 does not provide convincing evidence that the effects are mediated by regulation of E2F4 nuclear translocation.
8. Co-IP of endogenous proteins is important. The authors should provide this evidence.
9. Rescue experiments are required in some of the in vivo experiments.

Minor comments

1. There are multiple typos and grammar errors in the text. English editing should be applied.
2. Gene names should be properly applied (i.e. use of capital letters and/or italics).
3. There are multiple overstatements in the text (e.g., line 116 the word "critical" is used when the effects are modest). The authors should be critical in their statement.

We sincerely appreciate your thorough review and detailed comments on our manuscript. As suggested, we have provided a point-by-point response to Reviewer #3's comments below.

Point-by-point response to Reviewer#3:

Major comments

1. Figure 1 should show the data from multiple datasets available for PDAC rather than a selected one. Consistency of findings in the datasets is of utmost important (unless there is an explanation for discrepancies).

Response: We thank the reviewer for the critical comments. We agree that multiple datasets are essential in supporting the conclusion of elevated OFD1 expression in pancreatic cancer. We have conducted an additional search of the GEO database and validated the elevated expression of OFD1 in pancreatic cancer tissues compared to normal tissues across several additional clinical pancreatic cancer datasets (**Response3 Fig. 1; Manuscript-Fig. 1a, Manuscript-Extended Data Fig. 1a**). These datasets are identified as GSE15471, GSE71729, GSE62452, GSE272362, GSE71989, and GSE63158.

Response3 Figure 1

Response3 Figure 1. Upregulation of OFD1 in human pancreatic cancer tissues. Gene expression levels of OFD1 in tumor (T) versus normal (N) tissues were analyzed using publicly available GEO datasets (GSE15471, GSE71729, GSE62452, GSE272362, GSE71989, and GSE63158). Statistical analysis was performed using paired t-test for GSE15471, and unpaired t-test for GSE71729, GSE62452, GSE272362, GSE71989, and GSE63158.

2. In several instances, antibody specificity should be demonstrated. This is the case particularly for antibodies detecting OFD1, DREAM complex members LIN52 and LIN54 (which are highly conserved proteins and most antibodies are inadequate).

Response: We thank the reviewer for raising the concerns about antibodies. The specificity of the OFD1 antibody has been extensively validated in several previous publications from our laboratory (Tang et al., Nature, 2013, PMID: 24089205; Cao et al., Nature Communications, PMID: 36973243). Regarding components of the DREAM complex, such as LIN54 and RBL2, we performed Western blotting (WB) to validate the antibody specificity after gene expression was silenced using siRNA sequences (note that we did not use an antibody for LIN52 in this study) (**Response3 Fig. 2**). Moreover, the antibodies employed in our experiments have been previously validated for specificity, as reported in the literature (PMID: 33626321).

Response3 Figure 2

Response3 Figure 2. Validation of RBL2 and LIN54 antibodies using corresponding siRNA-mediated knockdown. The specificity of RBL2 and LIN54 antibodies was assessed through siRNA-mediated knockdown. Cells were transfected with siRNA targeting either RBL2 or LIN54, and protein levels of both RBL2 and LIN54 were analyzed by Western blotting.

3. In line 1234: the authors should provide information on whether the effects are associated with the genetic features of the cell lines used.

Response: The reviewer refers to the HEK293T cells for mammalian protein expression and purification. HEK293T is human embryonic kidney cell line that is widely used as a tool cell line for protein expression and purification with ease of use and scalability. We used HEK293T cells here to express and purify recombinant human OFD1 and E2F4 proteins and test their direct protein-protein interaction. Additionally, to demonstrate the functional relevance of the OFD1-E2F4 interaction in pancreatic cancer, we performed endogenous immunoprecipitation (IP) in a panel of pancreatic cancer cell lines, including SW1990, MIA PaCa-2, and PATU8988T, to further validate the widespread occurrence of the OFD1-E2F4 interaction (**Response3 Fig. 3; Manuscript-Fig. 6h**).

Response3 Figure 3

Response3 Figure 3. Endogenous IP demonstrates the interaction between OFD1 and E2F4. Endogenous immunoprecipitation (IP) was performed to confirm the interaction of OFD1 and E2F4 in SW1990, MIA PaCa2, and PATU8988T pancreatic cancer cell lines. IgG was used as a negative control.

4. Are the effects of Olaparib also observable (even if at lower doses) in the lines that are sensitive to OFD1 knockdown?

Response: We thank the reviewer for the inspiring comment, we have indeed observed that in PANC1 pancreatic cancer cells, which are sensitive to OFD1 knockdown, Olaparib exhibits a synergistic effect when combined with OFD1 knockdown, even at lower doses. In our previous experiments, we treated PANC1 cells with Olaparib for 96 hours, and the IC50 was determined to be 179.2 μ M (**Response3 Fig. 4a**), as measured by the CCK8 assay. Here, we treated the PANC1

cell line with low doses of Olaparib (ranging from 0 to 25 μM) for 72 hours and found that, when combined with OFD1 knockdown, Olaparib synergistically inhibited cell proliferation, even at these lower concentrations (**Response3 Fig. 4b**).

Response3 Figure 4

Response3 Figure 4. Sensitivity of OFD1 WT and knockdown PANC1 cells to Olaparib. **a**, PANC1 wild-type cells were treated with 0, 12.5, 25, 50, 100, 200, 300 μM Olaparib for 96 hours. Cell viability was assessed using the CCK8 assay. **b**, PANC1 and PANC1-shOFD1 cells were treated with 0, 5 μM , 10 μM , or 25 μM Olaparib for 72 hours. Cell viability was measured by the CCK8 assay, and statistical analysis was performed using two-way ANOVA. The initial cell density was 2,000 cells per well in a 96-well plate.

5. There should be a genetic validation of the synthetic lethal effect between OFD1 knockdown and PARP inhibition. This applies to both in vitro and in vivo studies.

Response: We sincerely thank the reviewer for the insightful suggestion. PARP1, the primary target of PARP inhibitors (PARPi), exert two main effects: (i) catalytic inhibition of PARP1, thereby preventing PARylation, and (ii) "locking" or "trapping" PARP1 on damaged DNA (PMID: 23118055, 24356813, 27797957). However, there is a possibility that PARPi might also affect other members of the PARP family (e.g., PARP2) and may involve additional mechanisms (PMID: 31218365). In this study, we employed a genetic validation approach by knocking down both PARP1 and OFD1 in two pancreatic cancer cell lines (MIA PaCa2 and SW1990). Combined knockdown of both OFD1 and PARP1 displayed a synthetic lethal effect on pancreatic cancer cells (**Response 3 Fig. 5**), which is similar to the observed synergy between OFD1 knockdown and PARPi treatment.

Response 3 Figure 5

Response 3 Figure 5. Validation of the synthetic lethal effect of the combination of OFD1 and PARP1 knockdown

a-b, Western blot validation of PARP1 knockdown by siRNA and OFD1 knockdown by shRNA in MIA PaCa2 (a) and SW1990 (b) cells. **c**, MIA PaCa2 cells were treated with the combined knockdown of both OFD1 (shRNA) and PARP1 (siRNA) for 72 hours, followed by assessment of cell viability. Statistical differences between groups were analyzed using an unpaired t-test. **d**, SW1990 cells were treated with the combined knockdown of both OFD1 (shRNA) and PARP1 (siRNA) for 72 hours, followed by cell viability assessment. Statistical differences between groups were analyzed using an unpaired t-test.

6. Can the authors comment on the potential relevance of BRCA2 and/or PALB2? BRCA1 is notoriously less important in PDAC than the latter two genes.

Response: We appreciate the reviewer's comments about the relevance of these repair genes in PDAC. BRCA1, BRCA2, and PALB2 are key DNA maintenance genes and targets of the DREAM complex. In pancreatic cancer, the mutation frequencies of these genes are as follows: BRCA2 (3.9%), BRCA1 (1.2%), and PALB2 (0.9%) (PMID: 32855305). Mutations in these genes compromise the function of the homologous recombination (HR) repair pathway and are clinically relevant for the therapeutic use of PARP inhibitors in pancreatic cancer (PMID: 37798442). We examined the impact of OFD1 knockdown on the expression of BRCA1, BRCA2, and PALB2 in various pancreatic cancer cell lines. Notably, BRCA1 was consistently and significantly downregulated across all tested cell lines, whereas OFD1 knockdown significantly reduced BRCA2 expression only in PANC1 cells, however, it had limited effects on BRCA2 and PALB2 expression in the other cell lines (**Response3 Fig. 6**). The underlying mechanism of this discrepancy is currently unclear, which will be explored in the future studies.

Response3 Figure 6

Response3 Figure 6. Effect of OFD1 knockdown on HR genes in pancreatic cancer cells.

a-c, After OFD1 knockdown in three pancreatic cancer cell lines, western blot analysis was conducted to assess the expression levels of HR-related proteins, including BRCA1, BARD1, BRCA2, PALB2, and RAD51.

d, Volcano plot of RNA sequencing results from PATU8988T cells after OFD1 knockdown, highlighting the expression of BRCA1, BRCA2, PALB2, and RAD51 on the plot.

7. Figure 6 does not provide convincing evidence that the effects are mediated by regulation of E2F4 nuclear translocation.

Response: To address the reviewer's concern whether the effects (BRCA1 downregulation etc.) are mediated by regulation of E2F4 nuclear translocation, we designed and performed an experiment by fusing a nuclear localization signal (NLS) to the C-terminus of E2F4 and transfecting E2F4-NLS into SW1990 cells to illustrate the importance of E2F4 nuclear import to BRCA1 regulation. Our results showed that forced E2F4 nuclear import by expressing E2F4-NLS has a similar effect on BRCA1 downregulation as OFD1 knockdown (**Response3 Fig. 7a-7b**), suggesting that E2F4 nuclear import is important for BRCA1 downregulation. Moreover, peptide

disrupting the OFD1-E2F4 interaction also caused BRCA1 downregulation and enhanced Olaparib sensitivity (**Response3 Fig. 7c-7h**). Additionally, we have cited relevant literature that supports E2F4's role in the DREAM complex and regulation of BRCA1 (PMID: 31092693, PMID: 23108140), reinforcing the critical role of E2F4 nuclear translocation in BRCA1 downregulation.

Response3 Figure 7

Response Figure 7. E2F4 nuclear import is important in regulating BRCA1 expression.

a, Immunofluorescence imaging of E2F4 in SW1990 cells transiently transfected with wild-type (WT) GFP-E2F4 and GFP-E2F4 with a C-terminal nuclear localization signal (NLS). After transfection with E2F4-NLS, E2F4 is predominantly localized to the cell nucleus. **b**, Detection of OFD1 and BRCA1 expression in SW1990 cells transfected with either WT GFP-E2F4 or GFP-E2F4 with NLS. **c**, *In vitro* peptide competition screening of the precise OFD1 binding motif of E2F4. E2F4 relied its 283-302 amino acids to bind OFD1. The detailed amino acid sequence of candidate peptides derived from the E2F4 “regulatory” domain were shown and the competitive peptide was labeled in red. **d**, The OFD1-E2F4 binding *in vitro* was interrupted by giving the competitive peptide (283-302) at concentration 250 μ M. **e-f**, Representative immunofluorescence (IF) images illustrating the nuclear

translocation of E2F4 in MIA PaCa-2 cells treated with the peptide (283-302) and control peptide and quantification of E2F4 nuclear /cytoplasm ratio(f). **g**, Western blot results of BRCA1 in MIA PaCa-2 cells treated with peptide (283-302) and control peptide. The numbers under the gel lanes represent the ratio of BRCA1 band intensity to α -tubulin band intensity, which were normalized relative to the line 1 sample. **h**, Colony-formation assay indicating the effect of peptide (283-302) combined with various concentrations of Olaparib (0, 1.25, 2.5, 5, 10, 20 μ M) on MIA PaCa-2 cells.

8. Co-IP of endogenous proteins is important. The authors should provide this evidence.

Response: We concur with the reviewer for the importance of endogenous Co-IP. We have performed endogenous co-immunoprecipitation (Co-IP) experiments to investigate endogenous protein interactions as the reviewer suggested. Specifically, we performed endogenous Co-IP assays to assess the interaction between OFD1 and E2F4 in three pancreatic cancer cell lines: SW1990, MIA PaCa2, and PATU8988T (**Response3 Fig. 8; Manuscript-Fig. 6h**). The results demonstrate a strong and consistent interaction between OFD1 and E2F4 across all these cell lines. We believe this new data have addressed your concerns and further strengthen the conclusions of our study.

Response3 Figure 8

Response Figure 8. Endogenous IP demonstrates the interaction between OFD1 and E2F4. Endogenous immunoprecipitation (IP) was performed to confirm the interaction of OFD1 and E2F4 in SW1990, MIA PaCa2, and PATU8988T pancreatic cancer cell lines. IgG was used as a negative control.

9. Rescue experiments are required in some of the in vivo experiments.

Response: Agreed. We have performed rescue experiments in the *in vivo* tumor xenograft assays. Specifically, we used the pancreatic cancer cell line SW1990 and ectopically expressed an OFD1 shRNA resistant form. In this rescue experiment, re-expression of OFD1 can abolish the synthetic tumor suppression effect caused by the combination of OFD1 knockdown and Olaparib (**Response3 Fig. 9; Manuscript-Fig. 2e-2i, Page 6, lines 143-148**)

Response3 Figure 9

Response Figure 9. Knockdown of OFD1 and Olaparib imposes a synthetic lethality effect in suppressing SW1990 subcutaneous tumors.

a, SW1990 Xenograft Tumor Mouse Model: 5×10^6 sh-Ctr, shOFD1, and OFD1 re-expressing (shOFD1-OFD1 OE) SW1990 cells were injected subcutaneously into mice. When tumors reached approximately 100 mm³, mice were randomly assigned to treatment with either vehicle or Olaparib. Tumor volume was measured twice a week, and group differences in tumor volume were calculated on the final day of the experiment using two-way ANOVA. **b**, Representative images of tumors from each group of SW1990 xenografts at the end of the treatment. **c**, Tumor weights were measured at the endpoint of the experiment. **d**, Western blot analysis was performed to assess the expression levels of OFD1 and BRCA1 in tumors from each treatment group. β-Actin was used as a loading control. **e**, Representative images of Ki-67 and γH2AX immunohistochemistry (IHC) staining in tumors harvested from each group. Scale bar = 50 μm. **f**, Quantification of Ki-67 and γH2AX IHC staining images. Results are presented as the mean values from a representative experiment, including three randomly selected fields of view from three different mice.

Minor comments

1. There are multiple typos and grammar errors in the text. English editing should be applied.

Response: Thanks for the reminder. We have carefully reviewed the entire text and have corrected all typos and grammar errors.

2. Gene names should be properly applied (i.e. use of capital letters and/or italics).

Response: We have carefully revised our manuscript.

3. There are multiple overstatements in the text (e.g., line 116 the word "critical" is used when the effects are modest). The authors should be critical in their statement.

Response: We appreciate the reviewer's suggestion and have revised the relevant statements accordingly to ensure more accurate wording (e.g., the word "critical" in **line 123-124** has been changed to "important").

Reviewer's comments:

Reviewer #4 - ECR (Remarks to the Author):

Response: We sincerely appreciate the reviewer's efforts and valuable suggestions. We have carefully addressed all comments and provided a point-by-point response. We hope this revision meets the standards for publication in *Nature Communications*.

Point-by-point response

Reviewer #1 (Remarks to the Author):

In the revised version, the authors made considerable improvements to the manuscript and addressed most of this reviewer's concerns. However, the proposed mechanism of the direct binding between the OFD1 and E2F4 is still not fully confirmed, although the authors now demonstrated the interaction between these proteins at the endogenous levels. The red (interaction) signal in the PLA assay shown in Figure 6f is not visible and does not match the graph, and the GFP-OFD1 signal is not visible in the siE2F4 panel. The experiment showing peptide competition in Figure 6k is not convincing and shows variable levels of many bands, including an unexplained dramatic increase of the E2F4-OFD1 interaction in the presence of peptide #1. The assay shown in support of the direct binding between the two proteins is not adequate because the recombinant proteins were purified from the mammalian cell line and the presence of additional factors can't be ruled out (in fact, several additional bands are clearly visible on the gels shown in Extended data Figure 6f, g). The binding competition assay shown in Figure 6i is also not convincing because of the apparently unreasonably high concentration of the peptide #6 necessary to slightly attenuate the binding (although some details were missing in the Methods section to fully evaluate this experiment). Such weak activity of this peptide in the in vitro competition assays does not align with more robust effects observed in the cell-based assays. Therefore, it is recommended that the conclusion of the direct binding mechanism between E2F4 and OFD1 is downplayed and the panels 6f, g, and 6i-p are either removed, or shown as Extended data with appropriate comment of their limitations. Finally, there are still many typos in the text (although significantly improved).

Response:

We thank the reviewer raising critical and insightful comments about direct binding which are very helpful. We have improved our data quality and revised our statement. With all these improvements, we believe our conclusion about the OFD1-E2F4 direct interaction sustained. Please see below for our detailed description. However, we would like to follow the reviewer's suggestion to tone down this statement, we have now shown these data in Extended data (**Manuscript-Extended Figures 6-7**). Lastly, we have corrected typos by careful proofreading and editing. Thanks again for the reviewer's kind help.

1. The red (interaction) signal in the PLA assay shown in Figure 6f is not visible and does not match the graph, and the GFP-OFD1 signal is not visible in the siE2F4 panel.

Response: We thank the reviewer for this insightful comment. We have re-acquired and reprocessed the images using optimized exposure parameters and have carefully selected representative regions that more accurately reflect the experimental results. The updated images have been included in the revised manuscript (**Response1-Fig. 1a-1d; Manuscript-Extended Fig. 6c-6h**). Furthermore, to strengthen our conclusion, we conducted an additional proximity ligation assay (PLA), which demonstrated that treatment with the 283–302 peptide at 25 μ M significantly impaired the OFD1–E2F4 interaction in MIA PaCa-2 cells (**Response1-Fig. 1e-1f**).

Response1-Figure 1. The PLA assay for the interaction between OFD1 and E2F4.

a. Co-localization of OFD1 and E2F4 in PANC1 cells visualized by proximity ligation assay (PLA) assay. Representative images and zoomed-in pictures showing the co-localization of OFD1 and E2F4 in PANC1 cells with or without E2F4 knockdown, Scale bar = 10 μ m, Zoom in Scale bar = 2.5 μ m. **b.** The PLA signal intensity in each cell was quantified by ImageJ and presented. $n=33-41$, unpaired two tailed student's t-test **c.** Co-localization of OFD1 and E2F4 in PANC1 cells visualized by proximity ligation assay (PLA) assay. Representative images and zoomed-in pictures showing the co-localization of GFP-OFD1 and E2F4 in PANC1 cells with or without GFP knockdown. **d.** Quantification of PLA signal intensity in each cell using ImageJ. Scale bar = 10 μ m. Zoom in Scale bar = 2.5 μ m. Error bars, $n=65-68$, mean \pm SD, unpaired student's t-test (two-tailed). **e-f.** Representative images and zoomed-in pictures showing the co-localization of OFD1 and E2F4 in MIA PaCa2 cells, with or without 25 μ M peptide (283-302) treatment for 48 hours, visualized by the PLA assay (**e**). Scale bar = 10 μ m. Quantification of PLA signal intensity in each group was performed using ImageJ (**f**).

2.The experiment showing peptide competition in Figure 6k is not convincing and shows variable levels of many bands, including an unexplained dramatic increase of the E2F4-OFD1 interaction in the presence of peptide #1.

Response: We appreciate the reviewer's critical comments regarding the peptide competition assay in Figure 6k. We have optimized the experimental conditions and repeated these experiments, as detailed in the Methods section, and presented the new data in the revised manuscript (**Response1-Fig. 2a; Manuscript-Extended Figure. 7a**).

Response1-Figure 2. *In vitro* peptide competition screening for the OFD1 binding motif of E2F4.

a. Candidate peptides were systematically designed based on the amino acid sequence of the 182-336 amino acid residues of E2F4 protein, which presented OFD1 binding capacity, and their schematic representation is shown in the diagram. **b.** E2F4 binds to OFD1 through its amino acid residues 283–302 (peptide #7). Eight peptides labeled #1 to #8 encompassing amino acids 182-336 of E2F4 were utilized for competitively disrupting OFD-Flag and E2F4-Strep-HA interaction in an *in vitro* pull-down assay.

3. The assay shown in support of the direct binding between the two proteins is not adequate because the recombinant proteins were purified from the mammalian cell line and the presence of additional factors can't be ruled out (in fact, several additional bands are clearly visible on the gels shown in Extended data Figure 6f, g).

Response: We appreciate the reviewer's comments regarding the direct binding assay. To further substantiate the direct interaction between OFD1 and E2F4, we conducted an *in vitro* pull-down assay using E2F4 purified recombinant protein from *E. coli* and OFD1 purified recombinant protein from 293S cells. Both proteins are purified to more than 90% homogeneity. This result support the direct interaction between these proteins rather than non-specific binding (**Response1-Figure 3a-3c**).

Response1-Figure 3. Recombinant purification of E2F4 and OFD1 and assessment of their interaction using a pull-down assay

a. Protein purification of recombinant C-terminal Flag-tagged OFD1 from 293S cells. **b.** Protein purification of recombinant N-terminal His-tagged E2F4 from *E. coli*. **c.** *In vitro* pull-down assay

demonstrates a direct binding between recombinant OFD1-Flag and His-E2F4 protein.

4. The binding competition assay shown in Figure 6i is also not convincing because of the apparently unreasonably high concentration of the peptide #6 necessary to slightly attenuate the binding (although some details were missing in the Methods section to fully evaluate this experiment). Such weak activity of this peptide in the *in vitro* competition assays does not align with more robust effects observed in the cell-based assays.

Response: We thank the reviewer for the helpful comment. The initially weak inhibitory effect of peptide #6 observed in the binding competition assay was likely due to suboptimal assay conditions and the use of an overly broad concentration gradient, which may have masked the peptide's true activity. To address this concern, we first conducted a screening using a broad concentration range (0-200 μ M) to identify the active window. Based on these results, we then applied a more refined and physiologically relevant concentration gradient (0-50 μ M), consistent with the doses used in our cell-based assays. Under these optimized conditions, we observed a clear, dose-dependent attenuation of the OFD1-E2F4 interaction when peptides were present (**Response1 Fig. 4a-4b; Manuscript-Extended Fig. 8a-8c**). Moreover, we performed surface plasmon resonance (SPR) analysis to examine the interaction between the peptide (residues 283–302) and OFD1, with a binding affinity of $K_D = 8.27 \times 10^{-8}$ M (**Response1 Fig. 4c**), further supporting the peptide's functional role in modulating the OFD1-E2F4 interaction. Detailed experimental procedures have been added to the revised Methods section to allow full evaluation and ensure reproducibility of the assay.

Response1-Figure 4. The peptide (283-302) attenuates the OFD1-E2F4 interaction in a dose-dependent manner. a. The *in vitro* OFD1-E2F4 binding was evaluated in the presence of the competitive peptide (283-302) at titrated concentrations ranging from 0 to 200 μ M. **b.** The *in*

vitro OFD1-E2F4 binding was evaluated in the presence of the competitive peptide (283-302) at titrated concentrations ranging from 0 to 50 μ M. c. Direct binding validation between recombinant OFD1 protein and peptide (283-302 aas) through surface plasmon resonance (SPR) assay.

Reviewer #2 (Remarks to the Author):

I appreciate that the authors have addressed most of the comments, and I find their responses acceptable overall. However, I still have some concerns.

On rebuttal page 20, despite the significantly reduced BRCA1 levels across the three pancreatic cancer cell lines, the BARD1 levels remain unchanged. This raises questions about the specificity of antibodies used in the experiments.

BRCA1 reduction upon OFD1 depletion in various experiment are very different.

Additionally, some p-values are still missing.

In summary, while most of their responses are satisfactory, the remaining issues regarding data consistency and missing p-values should be clarified.

1. On rebuttal page 20, despite the significantly reduced BRCA1 levels across the three pancreatic cancer cell lines, the BARD1 levels remain unchanged. This raises questions about the specificity of antibodies used in the experiments.

Response: We appreciate the reviewer's insightful comment. Given that BRCA1 and BARD1 form a stable heterodimer and are mutually dependent for protein stability (PMID: 28976962; PMID: 32094664), the initially observed discrepancy raised valid concerns regarding antibody specificity. To address this, we replaced the original BARD1 antibody with a reported BARD1 antibody (Abcam, ab245434). We first validated the specificity of BRCA1 and BARD1 antibodies in cells depleted with BRCA1 or BARD1 by siRNAs. Knockdown efficiency was confirmed at both the mRNA (qPCR) and protein (Western blot) levels. Consistent with previous reports, BRCA1 knockdown reduced BARD1 protein levels and vice versa (**Response2 Fig. 1a-1d**), confirming the specificity of both antibodies. Using these validated antibodies, we re-examined BRCA1 and BARD1 expression following OFD1 knockdown and observed consistent downregulation of both proteins across all three pancreatic cancer cell lines (**Response2 Fig. 1e**), supporting the model of BRCA1-BARD1 co-regulation.

Response2-Figure 1. Impact of OFD1 knockdown on BRCA1 and BARD1 protein levels in pancreatic cancer cells.

a. qPCR analysis of BARD1 knockdown efficiency by siRNA and the impact of OFD1 knockdown on BARD1 expression in PANC1 cells. Data are presented as mean \pm SD (n = 3). Statistical significance was determined using an unpaired two-tailed Student's t-test. **b.** qPCR analysis of BRCA1 knockdown efficiency by siRNA and the impact of OFD1 knockdown on BRCA1 expression in PANC1 cells. Data are presented as mean \pm SD (n = 3). Statistical significance was determined using an unpaired two-tailed Student's t-test. **c.** Western blot analysis of BRCA1 knockdown efficiency by siRNA and its effect on BARD1 protein levels. **d.** Western blot analysis of BARD1 knockdown efficiency by siRNA and its effect on BRCA1 protein levels. **e.** Western blot analyses were performed to assess the expression levels of BRCA1 and BARD1 following OFD1 knockdown in three pancreatic cancer cell lines: PANC-1, MIA PaCa-2, and SW1990.

2. BRCA1 reduction upon OFD1 depletion in various experiment are very different.

Response: We thank the reviewer for this thoughtful comment. We note that the extent of BRCA1 reduction following OFD1 depletion exhibited some degree of variability across the pancreatic cancer cell lines. This variability likely stems from differences in shOFD1 knockdown efficiency and variation in Western blot exposure settings. To address these issues, we optimized experimental conditions and standardized exposure times, resulting in more consistent and reproducible BRCA1 downregulation (**Response2 Fig. 2a-2d; Manuscript Fig. 3h-j; Extended Fig. 3c**). In the previously submitted version of the manuscript, the data from BRCA1 rescue experiments under OFD1 overexpression were suboptimal due to insufficient doxycycline induction and non-standardized blot exposure settings. These technical limitations contributed to the inconsistency in BRCA1 expression. In the revised version, we have optimized the doxycycline concentration and imaging parameters, which resulted in clearer and more consistent BRCA1 downregulation (**Response2 Fig. 2e; Manuscript Fig. 3k**). These improvements have enhanced the robustness and reliability of our conclusions.

Response2-Figure 2. BRCA1 is downregulated by OFD1 knockdown and restored by OFD1

overexpression in pancreatic cancer cell lines.

a-d. Validation of BRCA1 downregulation by western blotting. Western blot analysis of BRCA1 protein levels in control versus shOFD1-transfected MIA PaCa-2 (a), SW1990 (b), PATU8988T (c), and PANC1005 (d) pancreatic cancer cell lines. Two independent shRNAs targeting OFD1 were used. **e.** Rescue of BRCA1 expression in MIA PaCa-2 Cells by Tet-on inducible GFP-OFD1. Western blot showing that the Tet-on inducible system re-expressed GFP-OFD1 (resistant to siOFD1) restored BRCA1 protein levels reduced by siOFD1 in MIA PaCa-2 cells. MIA PaCa-2-Tet-on-GFP-OFD1 cells were transfected with siOFD1 and treated with varying concentrations of doxycycline (0, 25, 50 ng/mL), OFD1 and BRCA1 protein levels were assessed by western blot.

3. Additionally, some p-values are still missing.

Response: We appreciate the reviewer's comment regarding the missing p-values. We have thoroughly re-examined the manuscript and added the missing p-values to the relevant figures. The updated p-values are now provided for all statistical comparisons, ensuring the results are fully transparent and reproducible. These changes have been incorporated into the revised manuscript.

Reviewer #4 (Remarks to the Author):

Response: We sincerely thank the co-reviewer for their contribution as part of the Nature Communications initiative to support peer review training and the recognition of Early Career Researchers. We greatly appreciate their time and constructive input, which have helped improve the quality of our manuscript.

Reviewer #5 - replacement for Reviewer #3 (Remarks to the Author):

This study identifies OFD1 as a regulator of BRCA1 expression through E2F4 and the DREAM complex. Mechanistically, the authors demonstrate that OFD1 physically interacts with E2F4, sequestering it in the cytoplasm. OFD1 depletion releases E2F4, allowing it to form the DREAM complex on the BRCA1 promoter, thereby reducing BRCA1 expression. This work provides a novel link between ciliopathy-related protein function (OFD1) and DNA repair regulation, an underexplored intersection of ciliopathy and tumor biology that enhances the study's significance. The authors present extensive in vitro validation using multiple pancreatic cancer cell lines, synergy assays with PARP inhibitors, and mechanistic analyses, including ChIP, PLA, and rescue experiments. In vivo, they employ xenografts, orthotopic tumor models, patient-derived xenografts (PDXs), and an immunocompetent KPC transgenic model. The inclusion of both PDX and KPC models strengthens the translational relevance of their findings.

The authors have provided detailed, point-by-point responses to Reviewer 3's comments, addressing concerns with additional data, including expanded GEO analyses, endogenous co-immunoprecipitations, and both in vitro and in vivo rescue experiments. Specifically, they have:

1. Supplemented Figure 1 with multiple PDAC datasets to validate OFD1 overexpression.
2. Provided antibody specificity data for key proteins.
3. Clarified cell line selection and extended RNA-seq and functional analyses across multiple pancreatic cancer cell lines.
4. Demonstrated synergy between low-dose Olaparib and OFD1 depletion in sensitive cells.
4. Genetically validated the synthetic lethal effect of combined OFD1 and PARP1 knockdown.
5. Addressed the potential role of BRCA2/PALB2, showing BRCA1 as the most consistently affected target.
6. Confirmed that E2F4 nuclear translocation mediates BRCA1 downregulation.
7. Supplied endogenous co-IP data supporting the OFD1–E2F4 interaction.
8. Included additional rescue experiments in in vivo models to further validate their conclusions.

Overall, the authors have satisfactorily addressed Reviewer 3's concerns, significantly strengthening both the mechanistic insights and translational impact of their work. The revised manuscript is substantially improved and appears suitable for publication, pending final editorial review.

While the text is generally clear, a final editorial sweep for style and grammar could further enhance clarity.

Response: We sincerely thank the reviewer for the positive and encouraging feedback. In response to the suggestion, we have carefully re-examined the entire manuscript and performed a thorough revision to further improve the style, grammar, and clarity of the text. We hope the current version meets the journal's standards for publication.

REVIEWERS' COMMENTS

Reviewer #1 (Remarks to the Author):

The authors have addressed all the concerns and provided additional data to support their conclusions.

Response: Thank you for the encouraging comment. We are pleased that the revised manuscript meets the expectations.

Reviewer #2 (Remarks to the Author):

The authors have satisfactorily addressed the majority of my concerns. The revised manuscript is significantly improved and clarifies key points previously raised. I find the current version acceptable and support its publication.

Response: We sincerely thank the reviewer for the positive assessment. We are glad that the revised manuscript has addressed the concerns and improved the clarity of the key points.

Reviewer #5 (Remarks to the Author):

The revised manuscript is suitable for publication in the journal Nature Communications.

Response: We sincerely thank the reviewer for the positive endorsement and support for publication.